# COUNTERING THE ATTACK-DEFENSE COMPLEXITY GAP FOR ROBUST CLASSIFIERS

## ABSTRACT

We consider the decision version of defending and attacking Machine Learning classifiers. We provide a rationale for known difficulties in building robust models by proving that, under broad assumptions, attacking a polynomial-time classifier is $NP$-complete in the worst case; conversely, training a polynomial-time model that is robust on even a single input is $\Sigma_2^P$-complete, barring collapse of the Polynomial Hierarchy. We also provide more general bounds for non-polynomial classifiers. We point out an alternative take on adversarial defenses that can sidestep such a complexity gap, by introducing Counter-Attack (CA), a system that computes on-the-fly robustness certificates for a given input up to an arbitrary distance bound $\varepsilon$. Finally, we empirically investigate how heuristic attacks can approximate the true decision boundary distance, which has implications for a heuristic version of CA. As part of our work, we introduce UG100, a dataset obtained by applying both heuristic and provably optimal attacks to limited-scale networks for MNIST and for CIFAR10. We hope our contributions can provide guidance for future research.

## 1 INTRODUCTION

Adversarial attacks, i.e. algorithms designed to fool machine learning models, represent a significant threat to the applicability of such models in real-world contexts (Brendel et al., 2019; Brown et al., 2017; Wu et al., 2020). Despite years of research effort, countermeasures (i.e. "defenses") to adversarial attacks are frequently fooled by applying small tweaks to existing techniques (Carlini & Wagner, 2016; 2017a; Croce et al., 2022; He et al., 2017; Hosseini et al., 2019; Tramer et al., 2020).

We argue that this pattern is due to differences between the fundamental mathematical problems that defenses and attacks need to tackle. Specifically, we prove that while attacking a polynomial-time classifier is $NP$-complete in the worst case, **training a polynomial-time model that is robust even on a single input is $\Sigma_2^P$-complete**. We also provide more general bounds for non-polynomial classifiers, showing that a $A$-time classifier can be attacked in $NP^A$ time. We then give an informal intuition for our theoretical results, which also applies to heuristic attacks and defenses. Our result highlights that, **unless the Polynomial Hierarchy collapses, there exists a potential, structural, difficulty for defense approaches that focus on building robust classifiers at training time**.

We then show that the asymmetry can be **sidestepped** by an alternative perspective on adversarial defenses. As an exemplification, we introduce a new technique, named Counter-Attack (CA) that, instead of training a robust model, evaluates robstness on the fly for a specific input by running an adversarial attack. This simple approach, while very simple, provides robustness guarantees against perturbations of an arbitrary magnitude $\varepsilon$. Additionally, we prove that while generating a certificate is $NP$-complete in the worst case, **attacking CA using perturbations of magnitude $\varepsilon' > \varepsilon$ is $\Sigma_2^P$-complete**, which represents a form of computational robustness – weaker than the one by (Garg et al., 2020), but holding under much more general assumptions. CA can be applied in any setting where at least one untargeted attack is known, while also allowing one to capitalize on future algorithmic improvements: **as adversarial attacks become stronger, so does CA**.

Finally, we **investigate the empirical performance of an approximate version of CA where a heuristic attack is used instead of an exact one**. This version achieves reduced computational time, at the cost of providing only approximate guarantees. We found heuristic attacks to be high-quality approximators for exact decision boundary distances, in experiments over a subsample of MNIST and CIFAR10 and small-scale Neural Networks. In particular, a pool of seven heuristic attacks

provided an accurate (average over-estimate between 2.04% and 4.65%) and predictable (average $R^2 > 0.99$) approximation of the true optimum. We compiled our benchmarks and generated adversarial examples (both exact and heuristic) in **a new dataset, named UG100**, and made it publicly available[1]. Overall, we hope our contributions can **support future research** by highlighting potential structural challenges, pointing out key sources of complexity, inspiring research on heuristics and tractable classes, and suggesting alternative perspectives on how to build robust classifiers.

## 2  RELATED WORK

Robustness bounds for NNs were first provided in (Szegedy et al., 2013), followed by (Hein & Andriushchenko, 2017) and (Weng et al., 2018b). One major breakthrough was the introduction of automatic verification tools, such as the Reluplex solver (Katz et al., 2017). However, the same work also showed that proving properties of a ReLU network is $NP$-complete. Researchers tried to address this issue by working in three directions. The first is building more efficient solvers based on alternative formulations (Dvijotham et al., 2018; Singh et al., 2018; Tjeng et al., 2019). The second involves training models that can be verified with less computational effort (Leino et al., 2021; Xiao et al., 2019) or provide inherent robustness bounds (Sinha et al., 2018). The third focuses on guaranteeing robustness under specific threat models (Han et al., 2021) or input distribution assumptions (Dan et al., 2020; Sinha et al., 2018). Since all these approaches have limitations that reduce their applicability (Silva & Najafirad, 2020), heuristic defenses tend to be more common in practice. Exact approaches can also be used to compute provably optimal adversarial examples (Carlini et al., 2017; Tjeng et al., 2019), although generating them requires a non-trivial amount of computational resources. Refer to Appendix M for a more in-depth overview of certified defenses.

Another line of research has focused on understanding the nature of robustness and adversarial attacks. Frameworks such as (Dreossi et al., 2019), (Pinot et al., 2019) and (Pydi & Jog, 2021) focused on formalizing the concept of adversarial robustness. Some studies have highlighted trade-offs between robustness (under specific definitions) and properties such as accuracy (Dobriban et al., 2020; Zhang et al., 2019), generalization (Min et al., 2021) and invariance (Tramèr et al., 2020). However, some of these results have been recently questioned, suggesting that these trade-offs might not be inherent in considered approaches (Yang et al., 2020; Zhang et al., 2020). Adversarial attacks have also been studied from the point of view of Bayesian learning to derive robustness bounds and provide insight into the role of uncertainty (Rawat et al., 2017; Richardson & Weiss, 2021; Vidot et al., 2021). Adversarial attacks have also been studied in the context of game theory (Ren et al., 2021), identifying Nash equilibria between attacker and defender (Pal & Vidal, 2020; Zhou et al., 2019).

Finally, some works have also focused on the computational complexity of specific adversarial attacks and defenses. In particular, Mahloujifar & Mahmoody (2019) showed that there exist exact polynomial-time attacks against classifiers trained on product distributions. Similarly, Awasthi et al. (2019) showed that for degree-2 polynomial threshold functions there exists a polynomial-time algorithm that either proves that the model is robust or finds an adversarial example. Other works have also provided hardness results; Degwekar et al. (2019) showed that there exist certain classification tasks such that learning a robust model is as hard as solving the Learning Parity with Noise problem (which is $NP$-hard); Song et al. (2021) showed that learning a single periodic neuron over noisy isotropic Gaussian distributions in polynomial time would imply that the Shortest Vector Problem (conjectured to be $NP$-hard) can be solved in polynomial time. Finally, Garg et al. (2020) showed that, by requiring attackers to provide a valid cryptographic signature for inputs, it is possible to prevent attacks with limited computational resources from fooling the model in polynomial time.

## 3  BACKGROUND AND FORMALIZATION

Extensive literature in the field of adversarial attacks suggests that generating adversarial examples is comparatively easier than building robust classifiers (Carlini & Wagner, 2016; 2017a; Croce et al., 2022; He et al., 2017; Hosseini et al., 2019; Tramer et al., 2020). In this section, we introduce some key definitions that we will employ to provide a theoretically grounded, potential, motivation for such

---

[1]All our code, datasets, pretrained weights and results are available anonymously under MIT license at https://anonymous.4open.science/r/counter-attack.

discrepancy. We aim at capturing the key traits shared by most of the literature on adversarial attacks, so as to identify properties that are valid under broad assumptions.

We start by defining the concept of *adversarial example*, which intuitively represents a modification of a legitimate input that is so limited as to be inconsequential from a practical perspective, but classified erroneously by a target model. Formally, let $f : X \to \{1, \ldots, N\}$ be a discrete classifier. Let $B_p(\boldsymbol{x}, \varepsilon) = \{\boldsymbol{x}' \in X \mid \|\boldsymbol{x} - \boldsymbol{x}'\|_p \leq \varepsilon\}$ be a $L^p$ ball of radius $\varepsilon$ and center $\boldsymbol{x}$. Then we have:

**Definition 1** (Adversarial Example). *Given an input $\boldsymbol{x}$, a threshold $\varepsilon$, and a $L^p$ norm[2], an adversarial example is an input $\boldsymbol{x}' \in B_p(\boldsymbol{x}, \varepsilon)$ such that $f(\boldsymbol{x}') \in C(\boldsymbol{x})$, where $C(\boldsymbol{x}) \subseteq \{1, \ldots, N\} \setminus \{f(\boldsymbol{x})\}$.*

This definition is a simplification compared to human perception, but it is adequate for a sufficiently small $\varepsilon$, and it is adopted in most of the relevant literature. An *adversarial attack* can then be viewed as an optimization procedure that attempts to find an adversarial example. We define an adversarial attack for a classifier $f$ as a function $a_{f,p} : X \to X$ that solves the following optimization problem:

$$\underset{\boldsymbol{x}' \in X}{\arg\min}\{\|\boldsymbol{x}' - \boldsymbol{x}\|_p \mid f(\boldsymbol{x}') \in C(\boldsymbol{x})\} \tag{1}$$

The attack is considered successful if the returned solution $\boldsymbol{x}' = a_{f,p}(\boldsymbol{x})$ also satisfies $\|\boldsymbol{x}' - \boldsymbol{x}\|_p \leq \varepsilon$. We say that an attack is *exact* if it solves Equation (1) to optimality; otherwise, we say that the attack is *heuristic*. An attack is said to be *targeted* if $C(\boldsymbol{x}) = C_{t,y'}(\boldsymbol{x}) = \{y'\}$ with $y' \neq f(\boldsymbol{x})$; it is instead *untargeted* if $C_u(\boldsymbol{x}) = \{1, \ldots, N\} \setminus \{f(\boldsymbol{x})\}$. We define the *decision boundary distance* $d_p^*(\boldsymbol{x})$ of a given input $\boldsymbol{x}$ as the minimum $L^p$ distance between $\boldsymbol{x}$ and another input $\boldsymbol{x}'$ such that $f(\boldsymbol{x}) \neq f(\boldsymbol{x}')$. Note that this is also the value of $\|a_{f,p}(\boldsymbol{x}) - \boldsymbol{x}\|_p$ for an exact, untargeted, attack.

Intuitively, a classifier is *robust w.r.t. an example $\boldsymbol{x}$* iff $\boldsymbol{x}$ cannot be successfully attacked. Formally:

**Definition 2** (($\varepsilon, p$)-Local Robustness). *A discrete classifier $f$ is ($\varepsilon, p$)-locally robust w.r.t. an example $\boldsymbol{x} \in X$ iff $\forall \boldsymbol{x}' \in B_p(\boldsymbol{x}, \varepsilon)$ we have $f(\boldsymbol{x}') = f(\boldsymbol{x})$.*

We then provide some additional definitions that are needed for our results, namely ReLU networks and FSFP spaces. ReLU networks are defined as follows:

**Definition 3** (ReLU network). *A ReLU network is a composition of sum, multiplication by a constant, and ReLU activation, where $ReLU : \mathbb{R} \to \mathbb{R}_0^+$ is defined as $ReLU(x) = max(x, 0)$.*

Note that any hardness result for ReLU classifiers also applies to the more general class of classifiers. Fixed-Size Fixed-Precision (FSFP) spaces, on the other hand, capture two common assumptions about real-world input spaces: all inputs can be represented with the same number of bits and there exists a positive minorant of the distance between inputs.

**Definition 4** (Fixed-Size Fixed-Precision space). *Given a real $p > 0$, a space $X \subseteq \mathbb{R}^n$ is FSFP if there exists a $\nu \in \mathbb{R}$ such that $\forall \boldsymbol{x}.|r(\boldsymbol{x}')| \leq \nu$ (where $|r(\boldsymbol{x})|$ is the size of the representation of $\boldsymbol{x}$) and there exists a $\mu \in \mathbb{R}$ such that $\mu > 0$ and $\forall \boldsymbol{x}, \boldsymbol{x}' \in X. (\|\boldsymbol{x}' - \boldsymbol{x}\|_p < \mu \implies \boldsymbol{x} = \boldsymbol{x}')$.*

Examples of FSFP spaces include most image encodings, as well as 32-bit and 64-bit IEE754 tensors. Examples of non-FSFP spaces include the set of all rational numbers in an interval. Similarly to ReLU networks, hardness results for FSFP spaces also apply to more general spaces.

## 4 AN ASYMMETRICAL SETTING

In this section, we provide a theoretically sound result that is a viable explanation for why attacks seem to outperform defenses. The core of our analysis is proving that attacks are less computationally expensive than defenses in the worst case, unless the Polynomial Hierarchy collapses. Specifically, we prove that the decision version of attacking a ReLU classifier is $NP$-complete:

**Theorem 1**[3] (Untargeted $L^\infty$ attacks against ReLU classifiers are $NP$-complete). *Let $U\text{-}ATT_p$ be the set of all tuples $\langle \boldsymbol{x}, \varepsilon, f \rangle$ such that:*

$$\exists \boldsymbol{x}' \in B_p(\boldsymbol{x}, \varepsilon).f(\boldsymbol{x}') \neq f(\boldsymbol{x}) \tag{2}$$

*where $\boldsymbol{x} \in X$, $X$ is a FSFP space and $f$ is a ReLU classifier. Then $U\text{-}ATT_\infty$ is $NP$-complete.*

---

[2]We use the term "norm" for $0 < p < 1$ even if in such cases the $L^p$ function is not subadditive.

[3]The proofs of all our theorems and corollaries can be found in the appendices.

**Corollary 1.1.** *For every $0 < p \leq \infty$, $U\text{-}ATT_p$ is $NP$-complete.*

**Corollary 1.2.** *Targeted $L^p$ attacks (for $0 < p \leq \infty$) against ReLU classifiers are $NP$-complete.*

**Corollary 1.3.** *Theorem 1 holds even if we consider the more general set of polynomial-time classifiers w.r.t. the size of the tuple.*

Theorem 1 represents a minor generalization of existing results in the literature (Katz et al., 2017). However, together with the following more general bound for non-polynomial-time classifiers, it lays the groundwork for our main result.

**Theorem 2.** *Let $A$ be a complexity class, let $f$ be a classifier, let $Z_f = \{\langle \boldsymbol{x}, y \rangle \mid y = f(\boldsymbol{x}), \boldsymbol{x} \in X\}$ and let $U\text{-}ATT_p(f) = \{\langle \boldsymbol{x}, \varepsilon, g \rangle \in U\text{-}ATT_p' \mid g = f\}$, where $U\text{-}ATT_p'$ is the same as $U\text{-}ATT_p$ but without the ReLU classifier restriction. If $Z_f \in A$, then for every $0 < p \leq \infty$, $U\text{-}ATT_p(f) \in NP^A$.*

**Corollary 2.1.** *For every $0 < p \leq \infty$, if $Z_f \in \Sigma_n^P$, then $U\text{-}ATT_p(f) \in \Sigma_{n+1}^P$.*

The latter result implies that, if $Z_f \in P$, then $U\text{-}ATT_p(f) \in NP$. Informally, Corollary 2.1 establishes that, under broad assumptions, evaluating and attacking a classifier are in complexity classes that are strongly conjectured to be distinct, with the attack problem being the harder one.

The decision version of *training a robust model* is even more complex (again in the worst case):

**Theorem 3** (Finding a set of parameters that make a ReLU network $(\varepsilon, p)$-locally robust on an input is $\Sigma_2^P$-complete). *Let $PL\text{-}ROB_p$ be the set of tuples $\langle \boldsymbol{x}, \varepsilon, f_{\boldsymbol{\theta}}, v \rangle$ such that:*

$$\exists \boldsymbol{\theta}'. (v_f(\boldsymbol{\theta}') = 1 \implies \forall \boldsymbol{x}' \in B_p(\boldsymbol{x}, \varepsilon). f_{\boldsymbol{\theta}'}(\boldsymbol{x}') = f_{\boldsymbol{\theta}'}(\boldsymbol{x})) \tag{3}$$

*where $\boldsymbol{x} \in X$, $X$ is a FSFP space and $v_f$ is a polynomial-time function that is 1 iff the input is a valid parameter set for $f$. Then $PL\text{-}ROB_\infty$ is $\Sigma_2^P$-complete.*

**Corollary 3.1.** *$PL\text{-}ROB_p$ is $\Sigma_2^P$-complete for all $0 < p \leq \infty$.*

**Corollary 3.2.** *Theorem 3 holds even if, instead of ReLU classifiers, we consider the more general set of polynomial-time classifiers w.r.t. the size of the tuple.*

Our results rely on worst-case constructions and assume that the Polynomial Hierarchy does not collapse; moreover, both $NP$ and $\Sigma_2^P$ can be solved via super-polynomial algorithms. That said, we believe our theorems to have a strong practical relevance. First, the Polynomial Hierarchy collapse is strongly conjectured to be false; second, even super-polynomial algorithms can have dramatically different run times (e.g. SAT vs Quantified Boolean Formula solvers). Finally, generic classifiers can learn (and are known to learn) complex input-output mappings with many local optima. Intuitively, this is the core of the scenario captured by our worst-case construction, and also what makes Equation (1) difficult to solve. Again intuitively, robustness requires solving a nested optimization problem with universal quantification (since we need to guarantee the same prediction on all neighboring points), thus motivating the higher complexity class. For this reason, we think **our results provide a plausible explanation for the hardness gap that is routinely observed in the relevant literature**. Of course, there are definitely sub-cases where the problem is simple enough for exact attacks to run in polynomial time (e.g. (Awasthi et al., 2019)); this suggests that, *under specific circumstances*, guaranteed robustness could be achieved at reasonable effort. By this argument, our proof also provides additional motivation for research on tractable classes of robust classifiers.

**Additional Sources of Asymmetry** There are additional, complementary, factors that may provide an advantage to the attacker. We review them informally, since they can support efforts to build more robust defenses. First, the attacker can gather information about the target model, e.g. by using genuine queries (Papernot et al., 2017), while the defender has not such advantage. As a result, the defender often needs to either make assumptions about adversarial examples (Hendrycks & Gimpel, 2017; Roth et al., 2019) or train models to identify common properties (Feinman et al., 2017; Grosse et al., 2017). These assumptions can be exploited, such as in the case of Carlini & Wagner (2017a), who generated adversarial examples that did not have the expected properties.Second, the attacker can focus on one input at the time, while the defender has to guarantee robustness on a large subset of the input space. This weakness can be exploited: for example, MagNet (Meng & Chen, 2017) relies on a model of the entire genuine distribution, which can be sometimes inaccurate. Carlini & Wagner (2017b) broke MagNet by searching for examples that were both classified differently and mistakenly considered genuine. Finally, defenses cannot significantly compromise the accuracy of a model. Adversarial training, for example, often reduces the clean accuracy of the model (Madry et al., 2018), leading to a trade-off between accuracy and robustness.

## 5 SIDE-STEPPING THE COMPUTATIONAL ASYMMETRY

The limitations imposed by Theorem 3 cannot be addressed directly in the general case (barring collapse of the Polynomial Hierarchy). However, they can be *sidestepped* by changing perspective: we exemplify this by introducing an alternative approach to provide robust classification, which also allows us to take advantage of existing defenses. Instead of obtaining a robust model from scratch, we propose to evaluate the robustness of the classifier on a case-by-case basis, flagging the input if a robust answer cannot be provided. Specifically, given a norm-order $p$ and threshold $\varepsilon$, we propose to:

- Design a model that is as robust as possible using available and practically viable defenses;
- For every input received, determine if the model is $(\varepsilon, p)$-locally robust on the input by running an adversarial attack on the input;
- If the attack succeeds, flag the input.

We name this technique *Counter-Attack* (CA). Instead of attempting to build a robust model, CA ensures that answers from a partially robust model are flagged as unreliable when they could be the result of an attack. This approach, while very simple, can take advantage of existing defenses, provides robustness guarantees, and is considerably hard to fool, as we will later prove.

The behavior in case an input is flagged depends on the context. Examples include relying on a slower but more robust model (e.g. a human), or rejecting the input altogether. This kind of approach is viable in all cases where the goal is to support (rather than replace) human decision-making.

Note that the flagging rate of CA is heavily dependent on the robustness of the model: a model that is robust on the entire input distribution will have a flagging rate of zero. Therefore, any improvement in the field of adversarial defenses also decreases the flagging rate of CA. Moreover, if there are known robustness bounds, they can be exploited to simplify the attack: for example, if the model is known to be $(\varepsilon_{cert}, p)$-robust on $\boldsymbol{x}$, with $\varepsilon_{cert} < \varepsilon$, the attack can focus on searching adversarial examples in $B_p(\boldsymbol{x}, \varepsilon) \setminus B_p(\boldsymbol{x}, \varepsilon_{cert})$. At the same time, developing stronger and faster attacks also benefits CA, since better attacks can find adversarial examples more quickly.

The major drawback of CA is that it requires running an exact adversarial attack on every input. We will investigate a possible mitigation for this phenomenon based on employing heuristic attacks, which still provide a significant degree of robustness (see Section 6). Finally, we stress that CA is just one of potentially several alternative paradigms that could circumvent the computational asymmetry. We hope our contribution will encourage other researchers to investigate this direction.

### 5.1 FORMAL PROPERTIES

When used with an exact attack, CA provides formal robustness guarantees for an arbitrary $p$ and $\varepsilon$:

**Theorem 4.** *Let $0 < p \leq \infty$ and let $\varepsilon > 0$. Let $f : X \to \{1, \ldots, N\}$ be a classifier and let $a$ be an exact attack. Let $f_{CA}^a : X \to \{1, \ldots, N\} \cup \{\star\}$ be defined as:*

$$f_{CA}^a(\boldsymbol{x}) = \begin{cases} f(\boldsymbol{x}) & \|a_{f,p}(\boldsymbol{x}) - \boldsymbol{x}\|_p > \varepsilon \\ \star & otherwise \end{cases} \tag{4}$$

*Then $\forall \boldsymbol{x} \in X$ an $L^p$ attack on $\boldsymbol{x}$ with radius greater than or equal to $\varepsilon$ and with $\star \notin C(\boldsymbol{x})$ fails.*

The notation $f_{CA}^a(\boldsymbol{x})$ refers to the classifier $f$ combined with CA, relying on attack $a$. The condition $\star \notin C(\boldsymbol{x})$ requires that the input generated by the attack should not be flagged by CA.

**Corollary 4.1.** *Let $1 \leq p \leq \infty$ and let $\varepsilon > 0$. Let $f$ be a classifier on inputs with $n$ elements that uses CA with norm $p$ and radius $\varepsilon$. Then for all inputs and for all $1 \leq r < p$, $L^r$ attacks of radius greater than or equal to $\varepsilon$ and with and $\star \notin C(\boldsymbol{x})$ will fail. Similarly, for all inputs and for all $r > p$, $L^r$ attacks of radius greater than or equal to $n^{\frac{1}{r} - \frac{1}{p}} \varepsilon$ and with $\star \notin C(\boldsymbol{x})$ will fail (treating $\frac{1}{\infty}$ as 0).*

Since the only expensive step in CA consists in applying an adversarial attack to an input, the complexity is the same as that of a regular attack. CA can therefore represent a more feasible task compared to training a robust model.

**Attacking with a Higher Radius** In addition to robustness guarantees for a chosen $\varepsilon$, CA provides a form of computational robustness even beyond its intended radius. To prove this statement, we first

formalize the task of attacking CA (referred to as Counter-CA, or CCA).This involves finding, given a starting point $\boldsymbol{x}$, an input $\boldsymbol{x}' \in B_p(\boldsymbol{x}, \varepsilon')$ that is adversarial but not flagged by CA, i.e. such that $f(\boldsymbol{x}') \in C(\boldsymbol{x}) \wedge \forall \boldsymbol{x}'' \in B_p(\boldsymbol{x}', \varepsilon).f(\boldsymbol{x}'') = f(\boldsymbol{x}')$. Note that, *for $\varepsilon' \leq \varepsilon$, no solution exists*, since $\boldsymbol{x} \in B_p(\boldsymbol{x}', \varepsilon)$ and $f(\boldsymbol{x}) \neq f(\boldsymbol{x}')$.

**Theorem 5** (Attacking CA with a higher radius is $\Sigma_2^P$-complete). *Let $CCA_p$ be the set of all tuples $\langle \boldsymbol{x}, \varepsilon, \varepsilon', C, f \rangle$ such that:*

$$\exists \boldsymbol{x}' \in B_p(\boldsymbol{x}, \varepsilon'). \left( f(\boldsymbol{x}') \in C(\boldsymbol{x}) \wedge \forall \boldsymbol{x}'' \in B_p(\boldsymbol{x}', \varepsilon).f(\boldsymbol{x}'') = f(\boldsymbol{x}') \right) \tag{5}$$

*where $\boldsymbol{x} \in X$, $X$ is a FSFP space, $\varepsilon' > \varepsilon$, $f(\boldsymbol{x}) \notin C(\boldsymbol{x})$ $f$ is a ReLU classifier and whether an output is in $C(\boldsymbol{x}^*)$ for some $\boldsymbol{x}^*$ can be decided in polynomial time. Then $CCA_\infty$ is $\Sigma_2^P$-complete.*

**Corollary 5.1.** *$CCA_p$ is $\Sigma_2^P$-complete for all $0 < p \leq \infty$.*

**Corollary 5.2.** *Theorem 5 holds even if, instead of ReLU classifiers, we consider the more general set of polynomial-time classifiers w.r.t. the size of the tuple.*

In other words, under our assumptions, fooling CA is harder than running it. This phenomenon represents a form of computational robustness, a term introduced by Garg et al. (2020) in a very different setting where genuine examples can be cryptographically signed. Corollary 2.1 also implies that, unless the Polynomial Hierarchy collapses, it is impossible to obtain a better gap between running the model and attacking (e.g. a $P$-time model that is $\Sigma_2^P$-hard to attack). Note that while Theorem 5 shows that fooling CA is $\Sigma_2^P$-complete in general, attacking can be expected to be easy in practice when $\varepsilon' \gg \varepsilon$: this is however a very extreme case, where the threshold may have been poorly chosen or the adversarial examples might be visually distinguishable from genuine examples.

## 5.2 Using Heuristic Attacks with CA

CA in its exact form has limited scalability due to Theorem 1. This could be addressed by using approaches with guaranteed bounds, as suggested in Section 5, or by simply relying on heuristic attacks. In this second scenario, to compensate for the heuristic nature of the employed attacks, we can flag the input $\boldsymbol{x}'$ if the attack fails to find an adversarial example in a radius of $\varepsilon + b(\boldsymbol{x}')$, where $b : X \to \mathbb{R}_0^+$ is a buffer model. The idea behind $b$ is that if a heuristic attack can identify an adversarial example within a radius of $\varepsilon + b(\boldsymbol{x}')$, an exact attack would be able to find an adversarial example within a radius of $\varepsilon$.

The effectiveness of this approach depends on how well heuristic attacks approximate the decision boundary distance, which is an interesting topic for investigation by itself. Note that consistency of the estimation is in fact more important than its accuracy: if a heuristic attack overestimates $d_p^*(\boldsymbol{x})$ in a predictable manner, we can train a buffer model to accurately correct the error.

With this approach, if the heuristic attack finds an adversarial example with distance less than $\varepsilon$, we can confidently flag the input (i.e. false positives are guaranteed to be impossible). However, if the distance is above $\varepsilon$, it is possible to have a false negative. Note that using approaches with guaranteed bounds would lead to a complementary situation.

**Fooling the Heuristic Attack-Based CA** The fact that the heuristic relaxation of CA can overestimate the decision boundary distance means that it is possible to generate adversarial examples with $\varepsilon' \leq \varepsilon$. Specifically, if an adversarial example $\boldsymbol{x}^{adv}$ for an input $\boldsymbol{x}$ is such that $d_p^*(\boldsymbol{x}^{adv}) \leq \varepsilon$ and $f(\boldsymbol{x}^{adv}) \neq f(\boldsymbol{x})$ but $\|a_{f,p}(\boldsymbol{x}^{adv}) - \boldsymbol{x}^{adv}\|_p > \varepsilon + b(\boldsymbol{x}^{adv})$, CA will incorrectly accept $\boldsymbol{x}^{adv}$. However, there are several informal considerations suggesting that fooling CA might be harder than running it. Such considerations are backed by empirical evidence in Section 6. First, both CA and CCA need to attack the same model, but CCA has at most as much information regarding the target model as CA, thus making the attacker at most as sample efficient as the defender. Second, fooling CA involves solving a nested optimization problem, while CA only needs to solve one; specifically, verifying the feasibility of a CCA solution involves running CA on the solution. Finally, as better attacks are developed the chances of CA being fooled become slimmer, since these attacks will be less likely to find sub-optimal adversarial examples.

## 6 EMPIRICAL INVESTIGATION OF HEURISTIC ATTACKS

Section 5.2 introduced the problem of investigating how accurately heuristic attacks can approximate the true decision boundary distance, which is needed for the heuristic version of CA to work, but also an interesting topic per se. In this section, we test whether $\|x - x_h\|_p$, where $x_h$ is an adversarial example found by a heuristic attack, is predictably close to the true decision boundary distance (i.e. $d_p^*(x)$). Consistently with Athalye et al. (2018) and Weng et al. (2018a), we focus on the $L^\infty$ norm. Additionally, we focus on *pools* of heuristic attacks. The underlying rationale is that different adversarial attacks should be able to cover for their reciprocal blind spots, providing a more reliable estimate. Since this evaluation is empirical, it requires sampling from a chosen distribution, in our case specific classifiers and the MNIST (LeCun et al., 1998) and CIFAR10 (Krizhevsky et al., 2009) datasets. This means that the results are not guaranteed for other distributions, or for other defended models: studying how adversarial attacks fare in these cases is an important topic for future work.

**Experimental Setup**   We randomly selected ~2.3k samples each from the test set of two datasets, MNIST and CIFAR10. We used three architectures per dataset (named A, B and C), each trained in three settings, namely standard training, PGD adversarial training (Madry et al., 2018) and PGD adversarial training with ReLU loss and pruning (Xiao et al., 2019) (from now on referred to as ReLU training), for a total of nine configurations per dataset. Since our analysis requires computing exact decision boundary distances, and size and depth both have a strong adverse impact on solver times, we used small and relatively shallow networks with parameters between ~2k and ~80k. Note that using (more scalable) NN verification approaches that can provide bounds without tightness guarantees is not an option, as they would prevent us from drawing any firm conclusion. For this reason, the natural accuracies for standard training are significantly below the state of the art (89.63% - 95.87% on MNIST and 47.85% - 55.81% on CIFAR10). Adversarial training also had a negative effect on natural accuracies (84.54% - 94.24% on MNIST and 45.19% - 51.35% on CIFAR10), similarly to ReLU training (83.69% - 93.57% on MNIST and 32.27% - 37.33% on CIFAR10).

We first ran a pool of heuristic attacks on each example, namely (Kurakin et al., 2017; Brendel et al., 2019; Carlini & Wagner, 2017c; Moosavi-Dezfooli et al., 2016; Goodfellow et al., 2015; Madry et al., 2018), as well as simply adding uniform noise to the input. Our main choice of attack parameters (from now on referred to as the "strong" parameter set) prioritizes finding adversarial examples at the expense of computational time. For each example, we considered the nearest feasible adversarial example found by any attack in the pool. We then ran the exact solver-based attack MIPVerify (Tjeng et al., 2019), which is able to find the nearest adversarial example to a given input. The entire process (including test runs) required ~45k core-hours on an HPC cluster. Each node of the cluster has 384 GB of RAM and features two Intel CascadeLake 8260 CPUs, each with 24 cores and a clock frequency of 2.4GHz. We removed the examples for which MIPVerify crashed in at least one setting, obtaining 2241 examples for MNIST and 2269 for CIFAR10. We also excluded from our analysis all adversarial examples for which MIPVerify did not find optimal bounds (atol = 1e-5, rtol = 1e-10), which represent on average 11.95% of the examples for MNIST and 16.30% for CIFAR10. Additionally, we ran the same heuristic attacks with a faster parameter set (from now on referred to as the "balanced" set) on a single machine with an AMD Ryzen 5 1600X six-core 3.6 GHz processor, 16 GBs of RAM and an NVIDIA GTX 1060 6 GB GPU. The process took approximately 8 hours. Refer to Appendix G for a more comprehensive overview of our experimental setup.

**Distance Approximation**   Across all settings, the mean distance found by the strong attack pool is 4.09±2.02% higher for MNIST and 2.21±1.16% higher for CIFAR10 than the one found by MIPVerify. For 79.81±15.70% of the MNIST instances and 98.40±1.63% of the CIFAR10 ones, the absolute difference is less than 1/255, which is the minimum distance in 8-bit image formats. The balanced attack pool performs similarly, finding distances that are on average 4.65±2.16% higher for MNIST and 2.04±1.13% higher for CIFAR10. The difference is below 1/255 for 77.78±16.08% of MNIST examples and 98.74±1.13% of CIFAR10 examples. We compare the distances found by the strong attack pool for MNIST A and CIFAR10 (using standard training) with the true decision bound distances in Figure 1. Refer to Appendix I for the full data.

For all datasets, architectures and training techniques there appears to be a strong, linear, correlation between the distance of the output of the heuristic attacks and the true decision boundary distance. We chose to measure this by training a linear regression model linking the two distances. For the

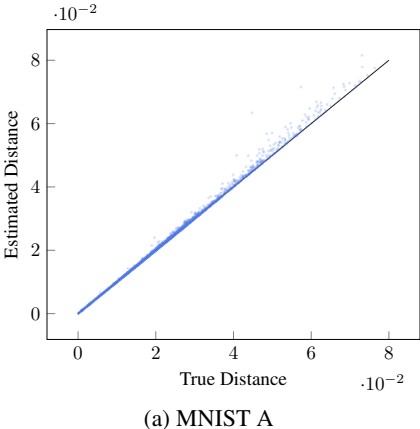

(a) MNIST A

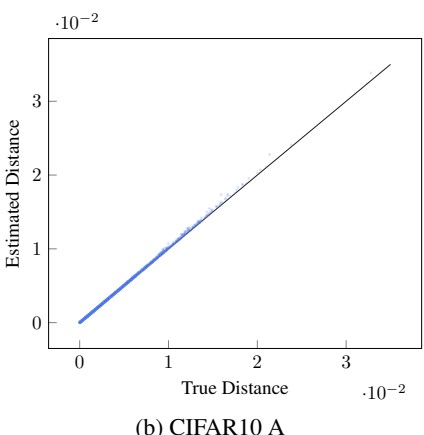

(b) CIFAR10 A

Figure 1: Distances of the nearest adversarial example found by the strong attack pool compared to those found by MIPVerify on MNIST A and CIFAR10 A with standard training. The black line represents the theoretical optimum. Note that no samples are below the black line.

strong parameter set, we find that the average $R^2$ across all settings is 0.992±0.004 for MNIST and 0.997±0.003 for CIFAR10. The balanced parameter set performs similarly, achieving an $R^2$ of 0.990±0.006 for MNIST and 0.998±0.002 for CIFAR10. From these results, we conjecture that increasing the computational budget of heuristic attacks does not necessarily improve predictability, although further tests would be needed to confirm such a claim. Note that such a linear model can also be used as a buffer function for heuristic CA. Another (possibly more reliable) procedure would consist in using quantile fitting; results for this approach are reported in Appendix H.

**Attack Pool Ablation Study** Due to the nontrivial computational requirements of running several attacks on the same input, we now study whether it is possible to drop some attacks from the pool without compromising its predictability. Specifically, we consider all possible pools of size $n$ (with a success rate of 100%) and pick the one with the highest average $R^2$ value over all architectures and training techniques. As show in Figure 2, adding attacks *does* increase predictability, although with diminishing returns. For example, the pool composed of the Basic Iterative Method, the Brendel & Bethge Attack and the Carlini & Wagner attack achieves on its own a $R^2$ value of 0.988±0.004 for MNIST+strong, 0.986±0.005 for MNIST+balanced, 0.935±0.048 for CIFAR10+strong and 0.993±0.003 for CIFAR10+balanced. Moreover, dropping both the Fast Gradient Sign Method and uniform noise leads to negligible ($\ll 0.001$) absolute variations in the mean $R^2$. These findings suggest that, as far as consistency is concerned, the choice of attacks represents a more important factor than the number of attacks in a pool. Refer to Appendix J for a more in-depth overview of how different attack selections affect consistency and accuracy.

**Efficient Attacks** We then explore if it possible to increase the efficiency of attacks by optimizing for fast, rather than accurate, results. We pick three new parameter sets (namely Fast-100, Fast-1k and Fast-10k) designed to find the nearest adversarial examples within the respective number of calls to the model. We find that while Deepfool is not the strongest adversarial attack (see Appendix I), it provides adequate results in very few model calls. For details on these results see Appendix K.

**Fooling the Heuristic Attack-Based CA** An open question from Section 5.2 is the empirical difficulty of fooling the version of CA based on heuristic attacks. Specifically, we carried on a limited experimentation by attempting to fool a CA-defended model. We used Deepfool Fast-1k as a heuristic attack for CA, then we built a proof-of-concept CCA implementation based on the PGD method, thus setting a baseline for attacks against CA. This variant uses a custom loss $\mathcal{L}_{CCA}(\boldsymbol{x}, \boldsymbol{y}) = \mathcal{L}_{PGD}(\boldsymbol{x}, \boldsymbol{y}) + \lambda \|\boldsymbol{x} - a_{f,p}(\boldsymbol{x})\|_p$, which rewards adversarial examples with over-estimated decision boundary distances. We then vary $\lambda$ in order to test various trade-offs between the two terms. In order to estimate the gradient of the second term, we use Natural Evolution Strategies (Wierstra et al., 2014). As a sanity check, we also attack using uniform noise. Due to the high computational requirements of such an experiment (30-60 minutes and ~1.2M model

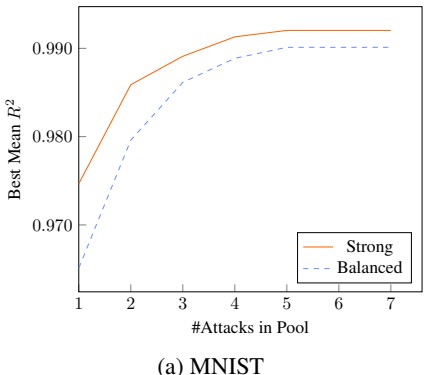
(a) MNIST

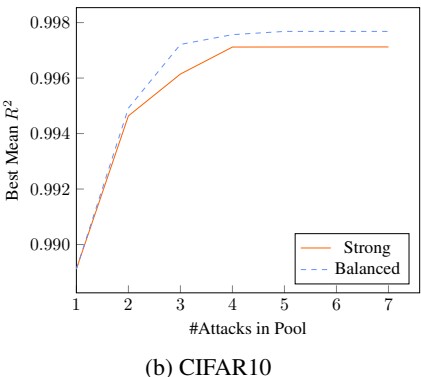
(b) CIFAR10

Figure 2: Best mean $R^2$ value in relation to the number of attacks in the pool.

calls per sample on the GTX 1060 machine), we only attack MNIST A Standard and CIFAR10 A Standard on 100 samples each. For comparison, running DeepFool on ten 250-element batches takes approximately 10 seconds. Overall, the attacks have a success rate of 0% - 3%. However, we find that increasing $\varepsilon'$ (while keeping $\varepsilon$ constant) increases the success rate, up to 100% for $\varepsilon' = 10 \cdot \varepsilon$. This suggests that as the difference between $\varepsilon'$ and $\varepsilon$ grows, so does the feasibility of fooling CA, which is consistent with our analysis. More in-depth results of our experiments can be found in Appendix L.

**UG100 Dataset** We collect all the adversarial examples found by both MIPVerify and the heuristic attacks into a new dataset, which we name UG100. UG100 can be used to benchmark new adversarial attacks. Specifically, we can determine how strong an attack is by comparing it to both the theoretical optimum and heuristic attack pools. Another potential application involves studying factors that affect whether adversarial attacks perform sub-optimally.

## 7 CONCLUSION

We proved that attacking is $NP$-complete in the worst case, while training a robust model is $\Sigma_2^P$-complete, barring collapse of the Polynomial Hierarchy. We then showed how such a structural asymmetry can be sidestepped by adopting a different perspective on defense. This is exemplified by Counter-Attack, a technique that can identify non-robust points in $NP$ time. We showed that CA can provide robustness guarantees up to an arbitrary $\varepsilon$. The CA approach naturally benefits from improvements in the field of adversarial attacks, and can be combined with other forms of defense. Due to its independence from the specific characteristics of the defended model, CA can also be applied to non-ML tools (e.g. signature-based malware detectors). We also believe that it should be possible to extend CA beyond classification. Finally, in an empirical evaluation we showed that heuristic attacks can provide an accurate and consistent approximation of the true decision boundary, which has implications for the viability of a heuristic version of CA. While our investigation is limited to small scale networks, we expect improvements in the field of NN verification will enable testing whether the observed results generalize to larger architectures.

Overall, we hope that our contributions can provide broad benefits to the field of adversarial robustness by 1) highlighting a potential, structural, challenge; 2) pointing out how that can be sidestepped by a change in perspective; 3) showing a proof-of-concept defense based on this idea; 4) providing an experimentation and dataset to serve as a baseline and starting point.

### REPRODUCIBILITY

We provide all our code and data in the repository linked in Section 1. Additionally, we report the key reproducibility information in Section 6, while all the other information can be found in Appendices G and L. To ensure maximum reproducibility, we also used consistent seeds across all experiments (one for parameter tuning and one for actual experiments). We also made sure to only rely on tools that are either open-source or for which there are free academic licenses. Concerning theoretical results, we provide full proofs of all theorems and corollaries in the appendices. Finally,

for users with a slow internet connection, we also provide UG100 in JSON format (containing only the found adversarial distances).

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

# A PROOF PRELIMINARIES

## A.1 NOTATION

We use $f_i$ to denote the $i$-th output of a network. We define $f$ as

$$f(\boldsymbol{x}) = \arg\max_i\{f_i(\boldsymbol{x})\} \qquad (6)$$

for situations where multiple outputs are equal to the maximum, we use the class with the lowest index.

## A.2 $\mu$ ARITHMETIC

Given two FSFP spaces $X$ and $X'$ with distance minorants $\mu$ and $\mu'$, we can compute new positive minorants after applying functions to the spaces as follows:

- Sum of two vectors: $\mu_{X+X'} = min(\mu, \mu')$;
- Multiplication by a constant: $\mu_{\alpha X} = \alpha\mu$;
- ReLU: $\mu_{ReLU(X)} = \mu$.

Since it is possible to compute the distance minorant of a space transformed by any of these functions in polynomial time, it is also possible to compute the distance minorant of a space transformed by any composition of such functions in polynomial time.

## A.3 FUNCTIONS

We now provide an overview of several functions that can be obtained by using linear combinations and ReLUs.

max   Carlini et al. (2017) showed that we can implement the max function using linear combinations and ReLUs as follows:

$$\max(x, y) = ReLU(x - y) + y \qquad (7)$$

We can also obtain an $n$-ary version of max by chaining multiple instances together.

$step$   If $X$ is a FSFP space, then the following scalar function:

$$step_0(x) = \frac{1}{\mu}\left(ReLU(x) - ReLU(x - \mu)\right) \qquad (8)$$

is such that $\forall i.\forall \boldsymbol{x} \in X, step_0(x_i)$ is 0 for $x_i \leq 0$ and 1 for $x_i > 0$.

Similarly, let $step_1$ be defined as follows:

$$step_1(x) = \frac{1}{\mu}\left(ReLU(x + \mu) - ReLU(x)\right) \qquad (9)$$

Note that $\forall i.\forall \boldsymbol{x} \in X, step_1(x_i) = 0$ for $x_i < 0$ and $step_1(x_i) = 1$ for $x_i \geq 0$.

**Boolean Functions**   We then define the Boolean functions $not : \{0, 1\} \to \{0, 1\}, and : \{0, 1\}^2 \to \{0, 1\}, or : \{0, 1\}^2 \to \{0, 1\}$ and $if : \{0, 1\}^3 \to \{0, 1\}$ as follows:

$$not(x) = 1 - x \qquad (10)$$
$$and(x, y) = step_1(x + y - 2) \qquad (11)$$
$$or(x, y) = step_1(x + y) \qquad (12)$$
$$if(a, b, c) = or(and(not(a), b), and(a, c)) \qquad (13)$$

where $if(a, b, c)$ returns $b$ if $a = 0$ and $c$ otherwise.

Note that we can obtain $n$-ary variants of $and$ and $or$ by chaining multiple instances together.

$cnf_3$   Given a set $\boldsymbol{z} = \{\{z_{1,1}, \ldots, z_{1,3}\}, \ldots, \{z_{n,1}, z_{n,3}\}\}$ of Boolean atoms (i.e. $z_{i,j}(\boldsymbol{x}) = x_k$ or $\neg x_k$ for a certain $k$) defined on an $n$-long Boolean vector $\boldsymbol{x}$, $cnf_3(\boldsymbol{z})$ returns the following Boolean function:

$$cnf_3'(\boldsymbol{x}) = \bigwedge_{i=1,\ldots,n} \bigvee_{j=1,\ldots,3} z_{i,j}(\boldsymbol{x}) \tag{14}$$

We refer to $\boldsymbol{z}$ as a 3CNF formula.

Since $cnf_3'$ only uses negation, conjunction and disjunction, it can be implemented using respectively $neg$, $and$ and $or$. Note that, given $\boldsymbol{z}$, we can build $cnf_3'$ in polynomial time w.r.t. the size of $\boldsymbol{z}$.

**Comparison Functions**   We can use $step_0$, $step_1$ and $neg$ to obtain comparison functions as follows:

$$geq(x, k) = step_1(x - k) \tag{15}$$
$$gt(x, k) = step_0(x, k) \tag{16}$$
$$leq(x, k) = not(gt(x, k)) \tag{17}$$
$$lt(x, k) = not(geq(x, k)) \tag{18}$$
$$eq(x, k) = and(geq(x, k), leq(x, k)) \tag{19}$$

Moreover, we define $open : \mathbb{R}^3 \to \{0, 1\}$ as follows:

$$open(x, a, b) = and(gt(x, a), lt(x, b)) \tag{20}$$

# B   Proof of Theorem 1

## B.1   $U\text{-}ATT_\infty \in NP$

To prove that $U\text{-}ATT_\infty \in NP$, we show that there exists a polynomial certificate for $U\text{-}ATT$ that can be checked in polynomial time. The certificate is the value of $\boldsymbol{x}'$, which will have a representation of the same size as $\boldsymbol{x}$ (due to the FSFP space assumption) and can be checked by verifying:

- $\|\boldsymbol{x} - \boldsymbol{x}'\|_\infty \le \varepsilon$, which can be checked in linear time;
- $f_{\boldsymbol{\theta}}(\boldsymbol{x}') \ne f(\boldsymbol{x})$, which can be checked in polynomial time.

## B.2   $U\text{-}ATT_\infty$ is $NP$-Hard

We will prove that $U\text{-}ATT_\infty$ is $NP$-Hard by showing that $3SAT \le U\text{-}ATT_\infty$.

Given a set of 3CNF clauses $\boldsymbol{z} = \{\{z_{11}, z_{12}, z_{13}\}, \ldots, \{z_{m1}, z_{m2}, z_{m3}\}\}$ defined on $n$ Boolean variables $x_1, \ldots, x_n$, we construct the following query $q(\boldsymbol{z})$ for $U\text{-}ATT_\infty$:

$$q(\boldsymbol{z}) = \langle \boldsymbol{x}^{(s)}, \frac{1}{2}, f \rangle \tag{21}$$

where $\boldsymbol{x}^{(s)} = \left(\frac{1}{2}, \ldots, \frac{1}{2}\right)$ is a vector with $n$ elements. Verifying $q(\boldsymbol{z}) \in U\text{-}ATT_\infty$ is equivalent to checking:

$$\exists \boldsymbol{x}' \in B_\infty \left(x_s, \frac{1}{2}\right) . f(\boldsymbol{x}') \ne f(\boldsymbol{x}^{(s)}) \tag{22}$$

Note that $\boldsymbol{x} \in B_\infty \left(\boldsymbol{x}^{(s)}, \frac{1}{2}\right)$ is equivalent to $\boldsymbol{x} \in [0, 1]^n$.

**Truth Values**   We will encode the truth values of $\hat{\boldsymbol{x}}$ as follows:

$$x_i' \in \left[0, \frac{1}{2}\right] \iff \hat{x}_i = 0 \tag{23}$$

$$x_i' \in \left(\frac{1}{2}, 1\right] \iff \hat{x}_i = 1 \tag{24}$$

We can obtain the truth value of a scalar variable by using $isT(x_i) = gt\left(x_i, \frac{1}{2}\right)$. Let $bin(\boldsymbol{x}) = or(isT(x_1), \ldots, isT(x_n))$.

**Definition of $f$**  We define $f$ as follows:

$$f_1(\boldsymbol{x}) = and(not(isx^{(s)}(\boldsymbol{x})), cnf_3'(bin(\boldsymbol{x}))) \tag{25}$$

$$f_0(\boldsymbol{x}) = not(f_1(\boldsymbol{x})) \tag{26}$$

where $cnf_3' = cnf_3(\boldsymbol{z})$ and $isx^{(s)}$ is defined as follows:

$$isx^{(s)}(\boldsymbol{x}) = and\left(eq\left(x_1, \frac{1}{2}\right), \ldots, eq\left(x_n, \frac{1}{2}\right)\right) \tag{27}$$

Note that $f$ is designed such that $f(\boldsymbol{x}^{(s)}) = 0$, while for $\boldsymbol{x}' \neq \boldsymbol{x}^{(s)}$, $f(\boldsymbol{x}') = 1$ iff the formula $\boldsymbol{z}$ is true for the variable assignment $bin(\boldsymbol{x}')$.

**Lemma 1.** $\boldsymbol{z} \in 3SAT \implies q(\boldsymbol{z}) \in U\text{-}ATT_\infty$

*Proof.* Let $\boldsymbol{z} \in 3SAT$. Therefore $\exists \boldsymbol{x}^* \in \{0,1\}^n$ such that $cnf_3(\boldsymbol{z})(\boldsymbol{x}^*) = 1$. Since $bin(\boldsymbol{x}^*) = \boldsymbol{x}^*$ and $\boldsymbol{x}^* \neq \boldsymbol{x}^{(s)}$, $f(\boldsymbol{x}^*) = 1$, which means that it is a valid solution for Equation (22). From this we can conclude that $q(\boldsymbol{z}) \in U\text{-}ATT_\infty$. $\square$

**Lemma 2.** $q(\boldsymbol{z}) \in U\text{-}ATT_\infty \implies \boldsymbol{z} \in 3SAT$

*Proof.* Since $q(\boldsymbol{z}) \in U\text{-}ATT_\infty$, $\exists \boldsymbol{x}^* \in [0,1]^n \setminus \{\boldsymbol{x}^{(s)}\}$ that is a solution to Equation (22) (i.e. $f(\boldsymbol{x}^*) = 1$). Then $cnf_3'(bin(\boldsymbol{x}^*)) = 1$, which means that there exists a $\hat{\boldsymbol{x}}$ (i.e. $bin(\boldsymbol{x}^*)$) such that $cnf_3'(\hat{\boldsymbol{x}}) = 1$. From this we can conclude that $\boldsymbol{z} \in 3SAT$. $\square$

Since:

- $q(\boldsymbol{z})$ can be computed in polynomial time;
- $\boldsymbol{z} \in 3SAT \implies q(\boldsymbol{z}) \in U\text{-}ATT_\infty$;
- $q(\boldsymbol{z}) \in U\text{-}ATT \implies \boldsymbol{z} \in 3SAT$.

we can conclude that $3SAT \leq U\text{-}ATT_\infty$.

### B.3  PROOF OF COROLLARY 1.1

#### B.3.1  $U\text{-}ATT_p \in NP$

The proof is identical to the one for $U\text{-}ATT_\infty$.

#### B.3.2  $U\text{-}ATT_p$ IS $NP$-HARD

The proof that $q(\boldsymbol{z}) \in U\text{-}ATT_p \implies \boldsymbol{z} \in 3SAT$ is very similar to the one for $U\text{-}ATT_\infty$. Since $q(\boldsymbol{z}) \in U\text{-}ATT_p$, we know that $\exists \boldsymbol{x}^* \in B_p(\boldsymbol{x}^{(s)}, \varepsilon) \setminus \{\boldsymbol{x}^{(s)}\}.f(\boldsymbol{x}^*) = 1$, which means that there exists a $\hat{\boldsymbol{x}}$ (i.e. $bin(\boldsymbol{x}^*)$) such that $cnf_3'(\hat{\boldsymbol{x}}) = 1$. From this we can conclude that $\boldsymbol{z} \in 3SAT$.

The proof that $\boldsymbol{z} \in 3SAT \implies q(\boldsymbol{z}) \in U\text{-}ATT_p$ is slightly different, due to the fact that since $\boldsymbol{x}^* \notin B_p(\boldsymbol{x}^{(s)}, \frac{1}{2})$ we need to use a different input to prove that $\exists \boldsymbol{x}' \in B_p(\boldsymbol{x}^{(s)}).f(\boldsymbol{x}') = 1$.

Let $0 < p < \infty$. Given a positive integer $n$ and a real $0 < p < \infty$, let $\rho_{p,n}(r)$ be a positive minorant of the $L^\infty$ norm of a vector on the $L^p$ sphere of radius $r$. For example, for $n = 2$, $p = 2$ and $r = 1$, any positive value less than or equal to $\frac{\sqrt{2}}{2}$ is suitable. Note that, for $0 < p < \infty$ and $n, r > 0$, $\rho_{p,n}(r) < r$.

Let $\boldsymbol{z} \in 3SAT$. Therefore $\exists \boldsymbol{x}^* \in \{0,1\}^n$ such that $cnf_3(\boldsymbol{z})(\boldsymbol{x}^*) = 1$. Let $\boldsymbol{x}^{**}$ be defined as:

$$x_i^{**} = \begin{cases} \frac{1}{2} - \rho_{p,n}\left(\frac{1}{2}\right) & x_i^* = 0 \\ \frac{1}{2} + \rho_{p,n}\left(\frac{1}{2}\right) & x_i^* = 1 \end{cases} \tag{28}$$

By construction, $\boldsymbol{x}^{**} \in B_p\left(\boldsymbol{x}^{(s)}, \rho_{p,n}\left(\frac{1}{2}\right)\right)$. Additionally, $bin(\boldsymbol{x}^{**}) = \boldsymbol{x}^*$, and since we know that $\boldsymbol{z}$ is true for the variable assignment $\boldsymbol{x}^*$, we can conclude that $f(\boldsymbol{x}^{**}) = 1$, which means that $\boldsymbol{x}^{**}$ is a valid solution for Equation (22). From this we can conclude that $q(\boldsymbol{z}) \in U\text{-}ATT_p$.

### B.4 PROOF OF COROLLARY 1.2

The proof is identical to the proof of Theorem 1 (for $p = \infty$) and Corollary 1.1 (for $0 < p < \infty$), with the exception of requiring $f(\boldsymbol{x}') = 1$.

### B.5 PROOF OF COROLLARY 1.3

The proof that attacking a polynomial-time classifier is in $NP$ is the same as that for Theorem 1.

Attacking a polynomial-time classifier is $NP$-hard due to the fact that the ReLU networks defined in the proof of Theorem 1 are polynomial-time classifiers. Since attacking a general polynomial-time classifier is a generalization of attacking a ReLU polynomial-time classifier, the problem is $NP$-hard.

## C PROOF OF THEOREM 2

Proving that $U\text{-}ATT_p(f) \in NP^A$ means proving that it can be solved in polynomial time by a non-deterministic Turing machine with an oracle that can solve a problem in $A$. Since $Z_f \in A$, we can do so by picking a non-deterministic Turing machine with access to an oracle that solves $Z_f$. We then generate non-deterministically the adversarial example and return the output of the oracle. Due to the FSFP assumption, we know that the size of this input is the same as the size of the starting point, which means that it can be generated non-deterministically in polynomial time. Therefore, $U\text{-}ATT_p(f) \in NP^A$.

### C.1 PROOF OF COROLLARY 2.1

Follows directly from Theorem 2 and the definition of $\Sigma_n^P$.

## D PROOF OF THEOREM 3

### D.1 PRELIMINARIES

$\Pi_2^P 3SAT$ is the set of all $\boldsymbol{z}$ such that:

$$\forall \hat{\boldsymbol{x}} \exists \hat{\boldsymbol{y}}. R(\hat{\boldsymbol{x}}, \hat{\boldsymbol{y}}) \tag{29}$$

where $R(\hat{x}, \hat{y}) = cnf_3(\boldsymbol{z})(\hat{x}_1, \ldots, \hat{x}_n, \hat{y}_1, \ldots, \hat{y}_n)$.

Stockmeyer (1976) showed that $\Pi_2 3SAT$ is $\Pi_2^P$-complete. Therefore, $co\Pi_2 3SAT$, which is defined as the set of all $\boldsymbol{z}$ such that:

$$\exists \hat{\boldsymbol{x}} \forall \hat{\boldsymbol{y}} \neg R(\hat{\boldsymbol{x}}, \hat{\boldsymbol{y}}) \tag{30}$$

is $\Sigma_2^P$-complete.

### D.2 $PL\text{-}ROB_\infty \in \Sigma_2^P$

$PL\text{-}ROB_\infty \in \Sigma_2^P$ if there exists a problem $A \in P$ and a polynomial $q$ such that $\forall \Gamma = \langle \boldsymbol{x}, \varepsilon, f_{\boldsymbol{\theta}}, v_f \rangle$:

$$\Gamma \in PL\text{-}ROB \iff \exists \boldsymbol{y}. |\boldsymbol{y}| \leq q(|\Gamma|) \wedge (\forall \boldsymbol{z}. (|\boldsymbol{z}| \leq q(|\Gamma|) \implies \langle \Gamma, \boldsymbol{y}, \boldsymbol{z} \rangle \in A)) \tag{31}$$

This can be proven by setting $\boldsymbol{y} = \boldsymbol{\theta}'$, $\boldsymbol{z} = \boldsymbol{x}'$ and $A$ as the set of triplets $\langle \Gamma, \boldsymbol{\theta}', \boldsymbol{x}' \rangle$ such that all of the following are true:

- $v_f(\boldsymbol{\theta}') = 1$;
- $\|\boldsymbol{x} - \boldsymbol{x}'\|_\infty \leq \varepsilon$;
- $f_{\boldsymbol{\theta}}(\boldsymbol{x}) = f_{\boldsymbol{\theta}}(\boldsymbol{x}')$.

Since all properties can be checked in polynomial time, $A \in P$ and thus $PL\text{-}ROB_\infty \in \Sigma_2^P$.

### D.3 $PL\text{-}ROB_\infty$ IS $\Sigma_2^P$-HARD

We will prove that $PL\text{-}ROB_\infty$ is $\Sigma_2^P$-hard by showing that $co\Pi_2 3SAT \leq PL\text{-}ROB_\infty$.

Let $n_{\hat{\boldsymbol{x}}}$ be the length of $\hat{\boldsymbol{x}}$ and let $n_{\hat{\boldsymbol{y}}}$ be the length of $\hat{\boldsymbol{y}}$.

Given a set $\boldsymbol{z}$ of 3CNF clauses, we construct the following query $q(\boldsymbol{z})$ for $PL\text{-}ROB$:

$$q(\boldsymbol{z}) = \langle \boldsymbol{x}^{(s)}, \frac{1}{2}, f_{\boldsymbol{\theta}}, v_f \rangle \tag{32}$$

where $\boldsymbol{x}^{(s)} = \left(\frac{1}{2}, \ldots, \frac{1}{2}\right)$ is a vector with $n_{\hat{\boldsymbol{y}}}$ elements and $v_f(\boldsymbol{\theta}) = 1 \iff \boldsymbol{\theta} \in \{0, 1\}^{n_{\hat{\boldsymbol{x}}}}$. Note that $\boldsymbol{\theta}' \in \{0, 1\}^{n_{\hat{\boldsymbol{x}}}}$ can be checked in polynomial time w.r.t. the size of the input.

**Truth Values**     We will encode the truth values of $\hat{\boldsymbol{x}}$ as a set of binary parameters $\boldsymbol{\theta}'$, while we will encode the truth values of $\hat{\boldsymbol{y}}$ using $\boldsymbol{x}'$ through the same technique mentioned in Appendix B.2.

**Definition of $f_{\boldsymbol{\theta}}$**     We define $f_{\boldsymbol{\theta}}$ as follows:

- $f_{\boldsymbol{\theta},1}(\boldsymbol{x}) = and(not(isx^{(s)}(\boldsymbol{x})), cnf_3''(\boldsymbol{\theta}, \boldsymbol{x}))$, where $cnf_3''$ is defined over $\boldsymbol{\theta}$ and $bin(\boldsymbol{x})$ using the same technique mentioned in Appendix B.2 and $isx^{(s)}(\boldsymbol{x}) = and_{i=1,\ldots,n} eq(x_i, \frac{1}{2})$;
- $f_{\boldsymbol{\theta},0}(\boldsymbol{x}) = not(f_{\boldsymbol{\theta},1}(\boldsymbol{x}))$.

Note that $f_{\boldsymbol{\theta}}(\boldsymbol{x}^{(s)}) = 0$ for all choices of $\boldsymbol{\theta}$. Additionally, $f_{\boldsymbol{\theta}}$ is designed such that:

$$\forall \boldsymbol{x}' \in B_\infty\left(\boldsymbol{x}^{(s)}, \frac{1}{2}\right) \setminus \{\boldsymbol{x}^{(s)}\}.\forall \boldsymbol{\theta}'.\, (v_f(\boldsymbol{\theta}') = 1 \implies (f_{\boldsymbol{\theta}'}(\boldsymbol{x}') = 1 \iff R(\boldsymbol{\theta}', bin(\boldsymbol{x}')))) \tag{33}$$

**Lemma 3.** $\boldsymbol{z} \in co\Pi_2 3SAT \implies q(\boldsymbol{z}) \in PL\text{-}ROB_\infty$

*Proof.* Since $\boldsymbol{z} \in co\Pi_2 3SAT$, there exists a Boolean vector $\boldsymbol{x}^*$ such that $\forall \hat{\boldsymbol{y}}.\neg R(\boldsymbol{x}^*, \hat{\boldsymbol{y}})$.

Then both of the following statements are true:

- $v_f(\boldsymbol{x}^*) = 1$, since $\boldsymbol{x}^* \in \{0, 1\}^{n_{\hat{\boldsymbol{x}}}}$;
- $\forall \boldsymbol{x}' \in B_\infty(\boldsymbol{x}^{(s)}, \varepsilon).f_{\boldsymbol{x}^*}(\boldsymbol{x}') = 0$, since $f_{\boldsymbol{x}^*}(\boldsymbol{x}') = 1 \iff R(\boldsymbol{x}^*, bin(\boldsymbol{x}'))$;

Therefore, $\boldsymbol{x}^*$ is a valid solution for Equation (3) and thus $q(\boldsymbol{z}) \in PL\text{-}ROB_\infty$. $\qquad\square$

**Lemma 4.** $q(\boldsymbol{z}) \in PL\text{-}ROB_\infty \implies \boldsymbol{z} \in co\Pi_2 3SAT$

*Proof.* Since $q(\boldsymbol{z}) \in PL\text{-}ROB_\infty$, there exists a $\boldsymbol{\theta}^*$ such that:

$$v_f(\boldsymbol{\theta}) = 1 \land \forall \boldsymbol{x}' \in B_\infty(\boldsymbol{x}^{(s)}, \varepsilon).f_{\boldsymbol{\theta}^*}(\boldsymbol{x}') = f_{\boldsymbol{\theta}^*}(\boldsymbol{x}^{(s)}) \tag{34}$$

Note that $\boldsymbol{\theta}^* \in \{0, 1\}^{n_{\hat{\boldsymbol{x}}}}$, since $v_f(\boldsymbol{\theta}^*) = 1$. Moreover, $\forall \hat{\boldsymbol{y}}.\neg R(\boldsymbol{\theta}^*, \hat{\boldsymbol{y}})$, since $bin(\hat{\boldsymbol{y}}) = \hat{\boldsymbol{y}}$ and $f_{\boldsymbol{\theta}^*}(\hat{\boldsymbol{y}}) = 1 \iff R(\boldsymbol{\theta}^*, \hat{\boldsymbol{y}})$.

Therefore, $\boldsymbol{\theta}^*$ is a valid solution for Equation (30), which implies that $\boldsymbol{z} \in co\Pi_2 3SAT$.

$\qquad\square$

Since:

- $q(\boldsymbol{z})$ can be computed in polynomial time;
- $\boldsymbol{z} \in co\Pi_2 3SAT \implies q(\boldsymbol{z}) \in PL\text{-}ROB_\infty$;
- $q(\boldsymbol{z}) \in PL\text{-}ROB_\infty \implies \boldsymbol{z} \in co\Pi_2 3SAT$.

we can conclude that $co\Pi_2 3SAT \leq PL\text{-}ROB_\infty$.

### D.4 PROOF OF COROLLARY 3.1

#### D.4.1 $PL\text{-}ROB_p \in \Sigma_2^P$

The proof is identical to the one for $PL\text{-}ROB_\infty$.

#### D.4.2 $PL\text{-}ROB_p$ IS $\Sigma_2^P$-HARD

We follow the same approach used in the proof for Corollary 1.1.

**Proof of** $q(\boldsymbol{z}) \in PL\text{-}ROB_p \implies \boldsymbol{z} \in co\Pi_2 3SAT$ If $q(\boldsymbol{z}) \in PL\text{-}ROB_p$, it means that $\exists \boldsymbol{\theta}^*.\left(v_f(\boldsymbol{\theta}^*) = 1 \implies \forall \boldsymbol{x}' \in B_p\left(\boldsymbol{x}^{(s)}, \frac{1}{2}\right).f(\boldsymbol{x}') = 0\right)$. Then $\forall \hat{\boldsymbol{y}}$, there exists a corresponding input $\boldsymbol{y}^{**} \in B_p\left(\boldsymbol{x}^{(s)}, \frac{1}{2}\right)$ defined as follows:

$$y_i^{**} = \begin{cases} \frac{1}{2} - \rho_{p,n}\left(\frac{1}{2}\right) & \hat{y}_i = 0 \\ \frac{1}{2} + \rho_{p,n}\left(\frac{1}{2}\right) & \hat{y}_i = 1 \end{cases} \tag{35}$$

such that $e^{(y)}(\boldsymbol{y}^{**}) = \hat{y}$. Since $\boldsymbol{y}^{**} \in B_p\left(\boldsymbol{x}^{(s)}, \frac{1}{2}\right)$, $cnf_3''(\boldsymbol{\theta}^*, bin(\boldsymbol{y}^{**})) = 0$, which means that $R(\boldsymbol{\theta}^*, \hat{\boldsymbol{y}})$ is false. In other words, $\exists \boldsymbol{\theta}^*.\forall \hat{\boldsymbol{y}}.\neg R(\boldsymbol{\theta}^*, \hat{\boldsymbol{y}})$, i.e. $\boldsymbol{z} \in co\Pi_2 3SAT$.

**Proof of** $\boldsymbol{z} \in co\Pi_2 3SAT \implies q(\boldsymbol{z}) \in PL\text{-}ROB_p$ The proof is very similar to the corresponding one for Theorem 3.

If $\boldsymbol{z} \in co\Pi_2 3SAT$, then $\exists \hat{\boldsymbol{x}}^*.\forall \hat{\boldsymbol{y}}.\neg R(\hat{\boldsymbol{x}}, \hat{\boldsymbol{y}})$. Set $\boldsymbol{\theta}^* = \hat{\boldsymbol{x}}^*$. We know that $f_{\boldsymbol{\theta}}^*(\boldsymbol{x}^{(s)}) = 0$. We also know that $\forall \boldsymbol{x}' \in B_p\left(\boldsymbol{x}^{(s)}, \frac{1}{2}\right) \setminus \{\boldsymbol{x}^{(s)}\}.(f_{\boldsymbol{\theta}^*}(\boldsymbol{x}) = 1 \iff cnf_3''(\boldsymbol{\theta}^*, \boldsymbol{x}') = 1)$. In other words, $\forall \boldsymbol{x}' \in B_p\left(\boldsymbol{x}^{(s)}, \frac{1}{2}\right) \setminus \{\boldsymbol{x}^{(s)}\}.(f_{\boldsymbol{\theta}^*}(\boldsymbol{x}') = 1 \iff R(\boldsymbol{\theta}^*, bin(\boldsymbol{x}')))$. Since $R(\boldsymbol{\theta}^*, \hat{\boldsymbol{y}})$ is false for all choices of $\hat{\boldsymbol{y}}$, $\forall \boldsymbol{x}' \in B_p\left(\boldsymbol{x}^{(s)}, \frac{1}{2}\right) \setminus \{\boldsymbol{x}^{(s)}\}.f_{\boldsymbol{\theta}^*}(\boldsymbol{x}') = 0$. Given the fact that $f_{\boldsymbol{\theta}^*}(\boldsymbol{x}^{(s)}) = 0$, we can conclude that $\boldsymbol{\theta}^*$ satisfies Equation (3).

### D.5 PROOF OF COROLLARY 3.2

Similarly to the proof of Corollary 1.3, it follows from the fact that ReLU classifiers are polynomial-time classifiers (w.r.t. the size of the tuple).

## E PROOF OF THEOREM 4

There are two cases:

- $\forall \boldsymbol{x}' \in B_p(\boldsymbol{x}, \varepsilon).f(\boldsymbol{x}') = f(\boldsymbol{x})$: then the attack fails because $f(\boldsymbol{x}) \notin C(\boldsymbol{x})$;

- $\exists \boldsymbol{x}' \in B_p(\boldsymbol{x}, \varepsilon).f(\boldsymbol{x}') \neq f(\boldsymbol{x})$: then due to the symmetry of the $L^p$ norm $\boldsymbol{x} \in B_p(\boldsymbol{x}', \varepsilon)$. Since $f(\boldsymbol{x}) \neq f(\boldsymbol{x}')$, $\boldsymbol{x}$ is a valid adversarial example for $\boldsymbol{x}'$, which means that $f(\boldsymbol{x}') = \star$. Since $\star \notin C(\boldsymbol{x})$, the attack fails.

### E.1 PROOF OF COROLLARY 4.1

Assume that $\forall \boldsymbol{x}.||\boldsymbol{x}||_r \geq \eta||\boldsymbol{x}||_p$ and fix $\boldsymbol{x}^{(s)} \in X$. Let $\boldsymbol{x}' \in B_r(\boldsymbol{x}^{(s)}, \eta\varepsilon)$ be an adversarial example. Then $||\boldsymbol{x}' - \boldsymbol{x}^{(s)}||_r \leq \eta\varepsilon$, and thus $\eta||\boldsymbol{x}' - \boldsymbol{x}^{(s)}||_p \leq \eta\varepsilon$. Dividing by $\eta$, we get $||\boldsymbol{x}' - \boldsymbol{x}^{(s)}||_p \leq \varepsilon$, which means that $\boldsymbol{x}^{(s)}$ is a valid adversarial example for $\boldsymbol{x}'$ and thus $\boldsymbol{x}'$ is rejected by $p$-CA.

We now proceed to find the values of $\eta$.

#### E.1.1 $1 \leq r < p$

We will prove that $||\boldsymbol{x}||_r \geq ||\boldsymbol{x}||_p$.

**Case $p < \infty$** Consider $e = \frac{x}{||x||_p}$. $e$ is such that $||e||_p = 1$ and for all $i$ we have $|e_i| \leq 1$. Since $r < p$, for all $0 \leq t \leq 1$ we have $|t|^p \leq |t|^r$. Therefore:

$$||e||_r = \left( \sum_{i=1}^{n} |e_i|^r \right)^{1/r} \geq \left( \sum_{i=1}^{n} |e_i|^p \right)^{1/r} = ||e||_p^{p/r} = 1 \tag{36}$$

Then, since $||e||_r \geq 1$:

$$||x||_r = || \ ||x||_p e||_r = ||x||_p ||e||_r \geq ||x||_p \tag{37}$$

**Case $p = \infty$** Since $||x||_r \geq ||x||_p$ for all $r < p$ and since the expressions on both sides of the inequality are compositions of continuous functions, as $p \to \infty$ we get $||x||_r \geq ||x||_\infty$.

### E.1.2 $\quad r > p$

We will prove that $||x||_r \geq n^{\frac{1}{r} - \frac{1}{p}} ||x||_p$.

**Case $r < \infty$** Hölder's inequality states that, given $\alpha, \beta \geq 1$ such that $\frac{1}{\alpha} + \frac{1}{\beta} = 1$ and given $f$ and $g$, we have:

$$||fg||_1 \leq ||f||_\alpha ||g||_\beta \tag{38}$$

Setting $\alpha = \frac{r}{r-p}$, $\beta = \frac{r}{p}$, $f = (1, \ldots, 1)$ and $g = (x_1^p, \ldots, x_n^p)$, we know that:

- $||fg||_1 = \sum_{i=1}^{n} (1 \cdot x_i^p) = ||x||_p^p$;
- $||f||_\alpha = \left( \sum_{i=1}^{n} 1 \right)^{1/\alpha} = n^{1/\alpha}$;
- $||g||_\beta = \left( \sum_{i} x_i^{pr/p} \right)^{p/r} = \left( \sum_{i} x_i^r \right)^{p/r} = ||x||_r^p$.

Therefore $||x||_p^p \leq n^{1/\alpha} ||x||_r^p$. Raising both sides to the power of $1/p$, we get $||x||_p \leq n^{1/(p\alpha)} ||x||_r$. Therefore:

$$||x||_p \leq n^{(r-p)/(pr)} ||x||_r = n^{\frac{1}{p} - \frac{1}{r}} ||x||_r \tag{39}$$

Dividing by $n^{\frac{1}{p} - \frac{1}{r}}$ we get:

$$n^{\frac{1}{r} - \frac{1}{p}} ||x||_p \leq ||x||_r \tag{40}$$

**Case $r = \infty$** Since the expressions on both sides of the inequality are compositions of continuous functions, as $r \to \infty$ we get $||x||_\infty \geq n^{-\frac{1}{p}} ||x||_p$.

## F    PROOF OF THEOREM 5

### F.1    $CCA_\infty \in \Sigma_2^P$

$CCA_\infty \in \Sigma_2^P$ iff there exists a problem $A \in P$ and a polynomial $p$ such that $\forall \Gamma = \langle x, \varepsilon, \varepsilon', C, f \rangle$:

$$\Gamma \in CCA_\infty \iff \exists y.|y| \leq p(|\Gamma|) \wedge (\forall z.(|z| \leq p(|\Gamma|) \implies \langle \Gamma, y, z \rangle \in A)) \tag{41}$$

This can be proven by setting $y = x'$, $z = x''$ and $A$ as the set of all triplets $\langle \Gamma, x', x'' \rangle$ such that all of the following are true:

- $||x - x'||_\infty \leq \varepsilon'$
- $f(x') \in C(x)$
- $||x'' - x'||_\infty \leq \varepsilon$
- $f(x'') = f(x')$

Since all properties can be checked in polynomial time, $A \in P$.

## F.2 $CCA_\infty$ IS $\Sigma_2^P$-HARD

We will show that $CCA_\infty$ is $\Sigma_2^P$-hard by proving that $co\Pi_2 3SAT \leq CCA_\infty$.

First, suppose that the length of $\hat{x}$ and $\hat{y}$ differ. In that case, we pad the shortest one with additional variables that will not be used.

Let $n$ be the maximum of the lengths of $\hat{x}$ and $\hat{y}$.

Given a set $z$ of 3CNF clauses, we construct the following query $q(z)$ for $CCA_\infty$:

$$q(z) = \langle x^{(s)}, \gamma, \frac{1}{2}, C_u, h \rangle \tag{42}$$

where $\frac{1}{4} < \gamma < \frac{1}{2}$ and $x^{(s)} = \left(\frac{1}{2}, \ldots, \frac{1}{2}\right)$ is a vector with $n$ elements. Verifying $q(z) \in CCA_\infty$ is equivalent to checking:

$$\exists x' \in B\left(x_s, \frac{1}{2}\right) . \left(h(x') \neq h(x) \wedge \left(\forall x'' \in B\left(x', \frac{1}{4}\right) . h(x'') = h(x')\right)\right) \tag{43}$$

Note that $x' \in [0, 1]^n$.

**Truth Values** We will encode the truth values of $\hat{x}$ and $\hat{y}$ as follows:

$$
\begin{aligned}
x_i'' \in \left(0, \frac{1}{4}\right) &\iff \hat{x}_i = 0 \wedge \hat{y}_i = 0 \\
x_i'' \in \left(\frac{1}{4}, \frac{1}{2}\right) &\iff \hat{x}_i = 0 \wedge \hat{y}_i = 1 \\
x_i'' \in \left(\frac{1}{2}, \frac{3}{4}\right) &\iff \hat{x}_i = 1 \wedge \hat{y}_i = 0 \\
x_i'' \in \left(\frac{3}{4}, 1\right) &\iff \hat{x}_i = 1 \wedge \hat{y}_i = 1
\end{aligned} \tag{44}
$$

Let $e_{\hat{x}i}(x) = gt\left(x_i, \frac{1}{2}\right)$. Let:

$$e_{\hat{y}i}(x) = or\left(open\left(x_i, \frac{1}{4}, \frac{1}{2}\right), open\left(x_i, \frac{3}{4}, 1\right)\right) \tag{45}$$

Note that $e_{\hat{x}i}(x_i'')$ returns the truth value of $\hat{x}_i$ and $e_{\hat{y}i}(x_i'')$ returns the truth value of $\hat{y}_i$ (as long as the input is within one of the ranges described in Equation (44)).

**Invalid Encodings** All the encodings other than the ones described in Equation (44) are not valid. We define $inv_F$ as follows:

$$inv_F(x) = or_{i=1,\ldots,n} or(out(x_i), edge(x_i)) \tag{46}$$

where $out(x_i) = or(leq(x_i, 0), geq(x_i, 1))$ and

$$edge(x_i) = or\left(eq\left(x_i, \frac{1}{4}\right), eq\left(x_i, \frac{1}{2}\right), eq\left(x_i, \frac{3}{4}\right)\right) \tag{47}$$

On the other hand, we define $inv_T$ as follows:

$$inv_T(x) = or_{i=1,\ldots,n} eq\left(x_i, \frac{1}{2}\right) \tag{48}$$

**Definition of $h$**    Let $g$ be a Boolean formula defined over $e^{(x)}(\boldsymbol{x})$ and $e^{(y)}(\boldsymbol{x})$ that returns the value of $R$ (using the same technique as $cnf'_3$).

We define $h$ as a two-class classifier, where:

$$h_1(\boldsymbol{x}) = or(inv_T(\boldsymbol{x}), and(not(inv_F(\boldsymbol{x})), g(\boldsymbol{x}))) \tag{49}$$

and $h_0(\boldsymbol{x}) = not(h_1(\boldsymbol{x}))$.

Note that:

- If $x_i = \frac{1}{2}$ for some $i$, the top class is 1; therefore, $h(\boldsymbol{x}^{(s)}) = 1$;
- Otherwise, if $\boldsymbol{x}$ is not a valid encoding, the top class is 0;
- Otherwise, the top class is 1 if $R(e^{(x)}(\boldsymbol{x}), e^{(y)}(\boldsymbol{x}))$ is true and 0 otherwise.

**Lemma 5.** $\boldsymbol{z} \in co\Pi_2 3SAT \implies q(\boldsymbol{z}) \in CCA_\infty$

*Proof.* If $\boldsymbol{z} \in co\Pi_2 3SAT$, then there exists a Boolean vector $\boldsymbol{x}^*$ such that $\forall \hat{\boldsymbol{y}}.\neg R(\boldsymbol{x}^*, \hat{\boldsymbol{y}})$.

We now prove that setting $\boldsymbol{x}' = \boldsymbol{x}^*$ satisfies Equation (5). First, note that $h(\boldsymbol{x}^*) = 0$, which satisfies $h(\boldsymbol{x}') \neq h(\boldsymbol{x})$. Then we need to verify that $\forall \boldsymbol{x}'' \in B_\infty(\boldsymbol{x}^*, \gamma).h(\boldsymbol{x}) = 0$.

For every $\boldsymbol{x}'' \in B_\infty(\boldsymbol{x}^*, \gamma)$, we know that $\boldsymbol{x}'' \in ([-\gamma, \gamma] \cup [1-\gamma, 1+\gamma])^n$. There are thus two cases:

- $\boldsymbol{x}''$ is not a valid encoding, i.e. $x_i'' \leq 0 \vee x_i'' \geq 1 \vee x_i'' \in \left\{\frac{1}{4}, \frac{3}{4}\right\}$ for some $i$. Then $h(\boldsymbol{x}'') = 0$. Note that, since $\gamma < \frac{1}{2}$, $\frac{1}{2} \notin [-\gamma, \gamma] \cup [1-\gamma, 1+\gamma]$, so it is not possible for $\boldsymbol{x}''$ to be an invalid encoding that is classified as 1;
- $\boldsymbol{x}''$ is a valid encoding. Then, since $\gamma < \frac{1}{2}$, $e^{(x)}(\boldsymbol{x}'') = \boldsymbol{x}^*$. Since $h(\boldsymbol{x}'') = 1$ iff $R(e^{(x)}(\boldsymbol{x}''), e^{(y)}(\boldsymbol{x}''))$ is true and since $R(\boldsymbol{x}^*, \hat{\boldsymbol{y}})$ is false for all choices of $\hat{\boldsymbol{y}}$, $h(\boldsymbol{x}'') = 0$.

Therefore, $\boldsymbol{x}^*$ satisfies Equation (43) and thus $q(\boldsymbol{z}) \in CCA_\infty$.

$\square$

**Lemma 6.** $q(\boldsymbol{z}) \in CCA_\infty \implies \boldsymbol{z} \in co\Pi_2 3SAT$

*Proof.* Since $q(\boldsymbol{z}) \in CCA$, there exists a $\boldsymbol{x}^* \in B\left(\boldsymbol{x}^{(s)}, \frac{1}{2}\right)$ such that $h(\boldsymbol{x}^*) \neq h(\boldsymbol{x}^{(s)})$ and $\forall \boldsymbol{x}'' \in B_\infty(\boldsymbol{x}^*, \gamma).h(\boldsymbol{x}'') = h(\boldsymbol{x}')$. We will prove that $e^{(x)}(\boldsymbol{x}^*)$ is a solution to $co\Pi_2 3SAT$.

Since $h(\boldsymbol{x}^{(s)}) = 1$, $h(\boldsymbol{x}^*) = 0$, which means that $\forall \boldsymbol{x}'' \in B_\infty(\boldsymbol{x}^*, \gamma).h(\boldsymbol{x}'') = 0$.

We know that $\boldsymbol{x}^* \in B_\infty\left(\boldsymbol{x}^{(s)}, \frac{1}{2}\right) = [0, 1]^n$. We first prove by contradiction that $\boldsymbol{x}^* \in \left([0, \frac{1}{2} - \gamma) \cup (\frac{1}{2} + \gamma, 1]\right)^n$. If $x_i^* \in [\frac{1}{2} - \gamma, \frac{1}{2} + \gamma]$ for some $i$, then the vector $\boldsymbol{x}^{(w)}$ defined as follows:

$$x_j^{(w)} = \begin{cases} \frac{1}{2} & i = j \\ x_i^* & \text{otherwise} \end{cases} \tag{50}$$

is such that $\boldsymbol{x}^{(w)} \in B_\infty(x_i^*, \gamma)$ and $h\left(\boldsymbol{x}^{(w)}\right) = 1$ (since $inv_T\left(\boldsymbol{x}^{(w)}\right) = 1$). This contradicts the fact that $\forall \boldsymbol{x}'' \in B_p(\boldsymbol{x}^*, \gamma).h(\boldsymbol{x}) = 0$. Therefore, $\boldsymbol{x}^* \in \left([0, \frac{1}{2} - \gamma) \cup (\frac{1}{2} + \gamma, 1]\right)^n$.

As a consequence, $\forall \boldsymbol{x}'' \in B_\infty(\boldsymbol{x}^*, \gamma).e^{(x)}(\boldsymbol{x}'') = e^{(x)}(\boldsymbol{x}^*)$.

We now prove that $\forall \hat{\boldsymbol{y}}^*.\exists \boldsymbol{x}''^* \in B_\infty(\boldsymbol{x}^*, \gamma)$ such that $e^{(y)}(\boldsymbol{x}''^*) = \hat{\boldsymbol{y}}^*$. We can construct such $\boldsymbol{x}''^*$ as follows. For every $i$:

- If $e^{(x)}(\boldsymbol{x}^*) = 0$ and $e^{(y)}(\boldsymbol{x}^*) = 0$, set $\boldsymbol{x}_i''^*$ equal to a value in $\left(0, \frac{1}{4}\right)$;
- If $e^{(x)}(\boldsymbol{x}^*) = 0$ and $e^{(y)}(\boldsymbol{x}^*) = 1$, set $\boldsymbol{x}_i''^*$ equal to a value in $\left(\frac{1}{4}, \gamma\right)$;
- If $e^{(x)}(\boldsymbol{x}^*) = 1$ and $e^{(y)}(\boldsymbol{x}^*) = 0$, set $\boldsymbol{x}_i''^*$ equal to a value in $\left(1 - \gamma, \frac{3}{4}\right)$;

- If $e^{(x)}(\boldsymbol{x}^*) = 1$ and $e^{(y)}(\boldsymbol{x}^*) = 1$, set $\boldsymbol{x}_i''^*$ equal to a value in $\left(\frac{3}{4}, 1\right)$.

By doing so, we have obtained a $\boldsymbol{x}''^*$ such that $\boldsymbol{x}''^* \in B_\infty(\boldsymbol{x}^*, \gamma)$ and $e^{(y)}(\boldsymbol{x}''^*) = \hat{\boldsymbol{y}}^*$.

Since:

- $e^{(x)}(\boldsymbol{x}'') = e^{(x)}(\boldsymbol{x}^*)$ for all $\boldsymbol{x}''$;

- $h(\boldsymbol{x}'') = 0$ for all $\boldsymbol{x}''$;

- $h(\boldsymbol{x}'') = 1$ iff $R(e^{(x)}(\boldsymbol{x}'')), e^{(y)}(\boldsymbol{x}''))$ is true;

$R(e^{(x)}(\boldsymbol{x}^*), \hat{\boldsymbol{y}}^*)$ is false for all choices of $\hat{\boldsymbol{y}}^*$. In other words, $\hat{\boldsymbol{x}}^*$ is a solution to Equation (30) and thus $\boldsymbol{z} \in co\Pi_2 3SAT$.

$\square$

Since:

- $q(\boldsymbol{z})$ can be computed in polynomial time;
- $\boldsymbol{z} \in co\Pi_2 3SAT \implies q(\boldsymbol{z}) \in CCA_\infty$;
- $q(\boldsymbol{z}) \in CCA_\infty \implies \boldsymbol{z} \in co\Pi_2 3SAT$;

we can conclude that $co\Pi_2 3SAT \leq CCA$.

### F.3 Proof of Corollary 5.1

The proof of $CCA_p \in \Sigma_2^P$ is the same as the one for Theorem 5.

For the hardness proof, we follow a more involved approach compared to those for Corollaries 1.1 and 3.1.

First, let $\varepsilon_{\rho_{p,n}}$ be the value of epsilon such that $\rho_{p,n}\left(\varepsilon_{\rho_{p,n}}\right) = \frac{1}{2}$. In other words, $B_p(\boldsymbol{x}^{(s)}, \varepsilon_{\rho_{p,n}})$ is an $L^p$ ball that contains $[0,1]^n$, while the intersection of the corresponding $L^p$ sphere and $[0,1]^n$ is the set $\{0,1\}^n$ (for $p < \infty$).

Let $inv_T'(\boldsymbol{x})$ be defined as follows:

$$inv_T'(\boldsymbol{x}) = or_{i=1,\dots,n}\left(or\left(eq\left(x_i, \frac{1}{2}\right), leq(x_i, 0), geq(x_i, 1)\right)\right) \tag{51}$$

Let $inv_F'(\boldsymbol{x})$ be defined as follows:

$$inv_F'(\boldsymbol{x}) = or_{i=1,\dots,n}\left(or\left(eq\left(x_i, \frac{1}{4}\right), eq\left(x_i, \frac{3}{4}\right)\right)\right) \tag{52}$$

We define $h'$ as follows:
$$h_1' = or(inv_T'(\boldsymbol{x}), and(not(inv_F'(\boldsymbol{x})), g(\boldsymbol{x})) \tag{53}$$
with $h_0'(\boldsymbol{x}) = not(h_1'(\boldsymbol{x}))$.

Note that:

- If $x_i \in (-\infty, 0] \cup \{\frac{1}{2}\} \cup [1, \infty)$ for some $i$, then the top class is 1;
- Otherwise, if $\boldsymbol{x}$ is not a valid encoding, the top class is 0;
- Otherwise, the top class is 1 if $R(e^{(x)}(\boldsymbol{x}), e^{(y)}(\boldsymbol{x}))$ is true and 0 otherwise.

Finally, let $\frac{1}{8} < \gamma' < \frac{1}{4}$. Our query is thus:

$$q(\boldsymbol{z}) = \langle \boldsymbol{x}^{(s)}, \gamma', \frac{1}{2}, C_u, h' \rangle \tag{54}$$

**Proof of $z \in co\Pi_2 3SAT \implies q(z) \in CCA_p$**   If $z \in co\Pi_2 3SAT$, then $\exists x^*.\forall \hat{y}.\neg R(x^*, \hat{y})$. Let $x^{**}$ be defined as follows:

$$x_i^{**} = \begin{cases} \frac{1}{4} & x_i^* = 0 \\ \frac{3}{4} & x_i^* = 1 \end{cases} \tag{55}$$

Note that:

- $x^{**} \in B_p\left(x^{(s)}, \varepsilon_{\rho_{p,n}}\right)$;
- $e^{(x)}(x^{**}) = x^*$;
- $f(x^{**}) = 0$, since $x^{**} \in \{\frac{1}{4}, \frac{3}{4}\}^n$;
- Since $\gamma' < \frac{1}{4}$, there is no $i$ such that $\exists x'' \in B_p(x^{**}, \gamma').x_i'' \in (-\infty, 0] \cup \{\frac{1}{2}\} \cup [1, \infty)$;
- For all $x'' \in B_p(x^{**}, \gamma')$:
    - If $x''$ is not a valid encoding (i.e. $x_i'' \in \{\frac{1}{4}, \frac{3}{4}\}$ for some $i$), then $h'(x'') = 0$;
    - Otherwise, $h'(x'') = 1$ iff $R(e^{(x)}(x''), e^{(y)}(x''))$ is true.

Therefore, since $\forall \hat{y}.\neg R(x^*, \hat{y})$, we know that $\forall x'' \in B_p(x^{**}, \gamma').f(x'') = 0$. In other words, $x^{**}$ is a solution to Equation (5).

**Proof of $q(z) \in CCA_p \implies z \in co\Pi_2 3SAT$**   If $q(z) \in CCA_p$, then we know that $\exists x^* \in B_p\left(x^{(s)}, \varepsilon_{\rho_{p,n}}\right). \left(h'(x^*) \neq h(x^{(s)}) \wedge \forall x'' \in B_p(x^*, \gamma').h'(x'') = h'(x^*)\right)$. In other words, $\exists x^* \in B_p\left(x^{(s)}, \varepsilon_{\rho_{p,n}}\right). (h'(x^*) = 0 \wedge \forall x'' \in B_p(x^*, \gamma').h'(x'') = 0)$.

We will first prove by contradiction that $x^* \in \left((\gamma', \frac{1}{2} - \gamma') \cup (\frac{1}{2} + \gamma', 1 - \gamma')\right)^n$.

First, suppose that $x_i^* \in (-\infty, 0) \cup (1, \infty)$ for some $i$. Then $h'(x^*) = 0$ due to the fact that $inv_T(x^*) = 1$.

Second, suppose that $x_i^* \in [0, \gamma'] \cup [1 - \gamma', 1]$ for some $i$. Then $x^{(w)}$, defined as follows:

$$x_j^{(w)} = \begin{cases} 0 & i = j \wedge x_i^* \in [0, \gamma'] \\ 1 & i = j \wedge x_i^* \in [1 - \gamma', 1] \\ x_j^* & j \neq i \end{cases} \tag{56}$$

is such that $x^{(w)} \in B_p(x^*, \gamma')$ but $h'(x^{(w)}) = 1$.

Finally, suppose that $x_i^* \in [\frac{1}{2} - \gamma, \frac{1}{2} + \gamma]$ for some $i$. Then $x^{(w)}$, defined as follows:

$$x_j^{(w)} = \begin{cases} \frac{1}{2} & i = j \\ x_j^* & \text{otherwise} \end{cases} \tag{57}$$

is such that $x^{(w)} \in B_p(x^*, \gamma')$ but $h'(x^{(w)}) = 1$.

Therefore, $x^* \in \left((\gamma', \frac{1}{2} - \gamma') \cup (\frac{1}{2} + \gamma', 1 - \gamma')\right)^n$.

As a consequence $\forall x'' \in B_p(x^*, \gamma').e^{(x)}(x'') = e^{(x)}(x')$.

From this, due to the fact that $\gamma' > \frac{1}{8}$ and that $p > 0$, we can conclude that for all $\hat{y}$, there exists a $x'' \in B_p(x^*, \gamma')$ such that:

$$\begin{aligned} x_i'' \in \left(0, \frac{1}{4}\right) \text{ for } x_i^* \in \left(\gamma', \frac{1}{2} - \gamma'\right), \hat{y}_i = 0 \\ x_i'' \in \left(\frac{1}{4}, \frac{1}{2}\right) \text{ for } x_i^* \in \left(\gamma', \frac{1}{2} - \gamma'\right), \hat{y}_i = 1 \\ x_i'' \in \left(\frac{1}{2}, \frac{3}{4}\right) \text{ for } x_i^* \in \left(\frac{1}{2} + \gamma', 1 - \gamma'\right), \hat{y}_i = 0 \\ x_i'' \in \left(\frac{3}{4}, 1\right) \text{ for } x_i^* \in \left(\frac{1}{2} + \gamma', 1 - \gamma'\right), \hat{y}_i = 1 \end{aligned} \tag{58}$$

In other words, for all $\hat{\boldsymbol{y}}$ there exists a corresponding $\boldsymbol{x}'' \in B_p(\boldsymbol{x}^*, \gamma')$ such that $e^{(y)}(\boldsymbol{x}'') = \hat{\boldsymbol{y}}$.

Therefore, since $h'(\boldsymbol{x}'') = 1$ iff $R(e^{(x)}(\boldsymbol{x}''), e^{(y)}(\boldsymbol{x}''))$ is true and since $\forall \boldsymbol{x}'' \in B_p(\boldsymbol{x}^*, \gamma').h'(\boldsymbol{x}'') = 0$, we can conclude that $\forall \hat{\boldsymbol{y}}.\neg R(e^{(x)}(\boldsymbol{x}^*), \hat{\boldsymbol{y}})$. In other words, $\boldsymbol{z} \in co\Pi_2 3SAT$.

## F.4 Proof of Corollary 5.2

Similarly to the proof of Corollary 1.3, it follows from the fact that ReLU classifiers are polynomial-time classifiers (w.r.t. the size of the tuple).

# G Full Experimental Setup

All our code is written in Python + PyTorch (Paszke et al., 2019), with the exception of the MIPVerify interface, which is written in Julia. When possible, most experiments were run in parallel, in order to minimize execution times.

**Models** All models were trained using Adam (Kingma & Ba, 2014) and dataset augmentation. We performed a manual hyperparameter and architecture search to find a suitable compromise between accuracy and MIPVerify convergence. The process required approximately 4 months. When performing adversarial training, following (Madry et al., 2018) we used the final adversarial example found by the Projected Gradient Descent attack, instead of the closest. To maximize uniformity, we used for each configuration the same training and pruning hyperparameters (when applicable), which we report in Table 1. We report the chosen architectures in Tables 2 and 3, while Table 4 outlines their accuracies and parameter counts.

**UG100** The first 250 samples of the test set of each dataset were used for hyperparameter tuning and were thus not considered in our analysis. For our G100 dataset, we sampled uniformly across each ground truth label and removed the examples for which MIPVerify crashed. Table 5 details the composition of the dataset by ground truth label.

**Attacks** For the Basic Iterative Method (BIM), the Fast Gradient Sign Method (FGSM) and the Projected Gradient Descent (PGD) attack, we used the implementations provided by the AdverTorch library (Ding et al., 2019). For the Brendel & Bethge (B&B) attack and the Deepfool (DF) attack, we used the implementations provided by the Foolbox Native library (Rauber et al., 2020). The Carlini & Wagner and the uniform noise attacks were instead implemented by the authors. We modified the attacks that did not return the closest adversarial example found (i.e. BIM, Carlini & Wagner, Deepfool, FGSM and PGD) to do so. For the attacks that accept $\varepsilon$ as a parameter (i.e. BIM, FGSM, PGD and uniform noise), for each example we first performed an initial search with a decaying value of $\varepsilon$, followed by a binary search. In order to pick the attack parameters, we first selected the strong set by performing an extensive manual search. The process took approximately 3 months. We then modified the strong set in order to obtain the balanced parameter set. We report the parameters of both sets (as well as the parameters of the binary and $\varepsilon$ decay searches) in Table 6.

**MIPVerify** We ran MIPVerify using the Julia library MIPVerify.jl and Gurobi (Gurobi Optimization, LLC, 2022). Since MIPVerify can be sped up by providing a distance upper bound, we used the same pool of adversarial examples utilized throughout the paper. For CIFAR10 we used the strong parameter set, while for MNIST we used the strong parameter set with some differences (reported in Table 7). Since numerical issues might cause the distance upper bound computed by the heuristic attacks to be slightly different from the one computed by MIPVerify, we ran a series of *exploratory runs*, each with a different correction factor (1.05, 1.25, 1.5, 2), and picked the first factor that caused MIPVerify to find a feasible (but not necessarily optimal) solution. If the solution was not optimal, we then performed a *main run* with a higher computational budget. We provide the parameters of MIPVerify in Table 8. We also report in Table 9 the percentage of tight bounds for each combination.

Table 1: Training and pruning hyperparameters.

| Parameter Name | Value | |
|---|---|---|
| | **MNIST** | **CIFAR10** |
| Common Hyperparameters | | |
| Epochs | 425 | |
| Learning Rate | 1e-4 | |
| Batch Size | 32 | 128 |
| Adam $\beta$ | (0.9, 0.999) | |
| Flip % | 50% | |
| Translation Ratio | 0.1 | |
| Rotation (deg.) | 15° | |
| Adversarial Hyperparameters (Adversarial and ReLU only) | | |
| Attack | PGD | |
| Attack #Iterations | 200 | |
| Attack Learning Rate | 0.1 | |
| Adversarial Ratio | 1 | |
| $\varepsilon$ | 0.05 | 2/255 |
| ReLU Hyperparameters (ReLU only) | | |
| L1 Regularization Coeff. | 2e-5 | 1e-5 |
| RS Loss Coeff. | 1.2e-4 | 1e-3 |
| Weight Pruning Threshold | 1e-3 | |
| ReLU Pruning Threshold | 90% | |

Table 2: MNIST Architectures.

(a) MNIST A

| Input |
|---|
| Flatten |
| Linear (in = 784, out = 100) |
| ReLU |
| Linear (in = 100, out = 10) |
| Output |

(b) MNIST B

| Input |
|---|
| Conv2D (in = 1, out = 4, 5x5 kernel, stride = 3, padding = 0) |
| ReLU |
| Flatten |
| Linear (in = 256, out = 10) |
| Output |

(c) MNIST C

| Input |
|---|
| Conv2D (in = 1, out = 8, 5x5 kernel, stride = 4, padding = 0) |
| ReLU |
| Flatten |
| Linear (in = 288, out = 10) |
| Output |

Table 3: CIFAR10 architectures.

(a) CIFAR10 A

| Input |
|---|
| Conv2D (in = 3, out = 8, 3x3 kernel, stride = 2, padding = 0) |
| ReLU |
| Flatten |
| Linear (in = 1800, out = 10) |
| Output |

(b) CIFAR10 B

| Input |
|---|
| Conv2D (in = 3, out = 20, 5x5 kernel, stride = 4, padding = 0) |
| ReLU |
| Flatten |
| Linear (in = 980, out = 10) |
| Output |

(c) CIFAR10 C

| Input |
|---|
| Conv2D (in = 3, out = 8, 5x5 kernel, stride = 4, padding = 0) |
| ReLU |
| Conv2D (in = 8, out = 8, 3x3 kernel, stride = 2, padding = 0) |
| ReLU |
| Flatten |
| Linear (in = 72, out = 10) |
| Output |

Table 4: Parameter counts and accuracies of trained models.

| Architecture | #Parameters | Training | Accuracy |
|---|---|---|---|
| MNIST A | 79510 | Standard | 95.87% |
| | | Adversarial | 94.24% |
| | | ReLU | 93.57% |
| MNIST B | 2674 | Standard | 89.63% |
| | | Adversarial | 84.54% |
| | | ReLU | 83.69% |
| MNIST C | 3098 | Standard | 90.71% |
| | | Adversarial | 87.35% |
| | | ReLU | 85.67% |
| CIFAR10 A | 18234 | Standard | 53.98% |
| | | Adversarial | 50.77% |
| | | ReLU | 32.85% |
| CIFAR10 B | 11330 | Standard | 55.81% |
| | | Adversarial | 51.35% |
| | | ReLU | 37.33% |
| CIFAR10 C | 1922 | Standard | 47.85% |
| | | Adversarial | 45.19% |
| | | ReLU | 32.27% |

Table 5: Ground truth labels of the UG100 dataset.

(a) MNIST

| Ground Truth | Count | % |
|---|---|---|
| 0 | 219 | 9.77% |
| 1 | 228 | 10.17% |
| 2 | 225 | 10.04% |
| 3 | 225 | 10.04% |
| 4 | 225 | 10.04% |
| 5 | 220 | 9.82% |
| 6 | 227 | 10.13% |
| 7 | 221 | 9.86% |
| 8 | 225 | 10.04% |
| 9 | 226 | 10.08% |

(b) CIFAR10

| Ground Truth | Count | % |
|---|---|---|
| Airplane | 228 | 10.05% |
| Automobile | 227 | 10.00% |
| Bird | 228 | 10.05% |
| Cat | 228 | 10.05% |
| Deer | 226 | 9.96% |
| Dog | 227 | 10.00% |
| Frog | 227 | 10.00% |
| Horse | 227 | 10.00% |
| Ship | 225 | 9.92% |
| Truck | 226 | 9.96% |

## H  QUANTILE-BASED CALIBRATION

The buffer function in CA can be empirically calibrated so as to control the chance of false positives (i.e. inputs wrongly reported as not robust) and false negatives (i.e. non-robust inputs reported as being robust).

Given the strong correlation that we observed between the distance of heuristic adversarial examples and the true decision boundary distance, using a linear model for $b_\alpha$ seems a reasonable choice. Under this assumption, the buffer value depends only on the distance between the original example and the adversarial one, i.e. on $d(x, a_{f,\theta}(x))$. This property allows us to rewrite the main check performed by CA as:

$$||x - a_f(x))||_p - b(x) = \alpha_1 ||x - a_{f,\theta}(x)||_p + \alpha_0 \leq \varepsilon \tag{59}$$

The parameters $\alpha_1, \alpha_0$ can then be obtained via quantile regression (Koenker & Bassett Jr, 1978) by using the true decision boundary distance (i.e. $d_p^*(x)$) as a target.

The approach provides a simple, interpretable mechanism to control how conservative the detection check should be: with a small quantile, CA will tend to underestimate the decision boundary distance, leading to fewer missed detections, but more false alarms; using a high quantile will lead to the opposite behavior.

We test this type of buffer using 5-fold cross-validation on each configuration. Specifically, we calibrate the model using 1%, 50% and 99% as quantiles. Tables 10 to 13 provide a comparison between the expected quantile and the average true quantile of each configuration on the validation folds. Additionally, we plot in Figures 3 to 8 the mean $F_1$ score in relation to the choice of $\varepsilon$.

## I  ADDITIONAL RESULTS

Tables 14 to 17 detail the performance of the various attack sets on every combination, while Figures 9 to 14 showcase the relation between the true and estimated decision boundary distances.

## J  ABLATION STUDY

We outline the best attack pools by size in Tables 18 to 21. Additionally, we report the performance of pools composed of individual attacks in Tables 22 to 25. Finally, we detail the performance of dropping a specific attack in Tables 26 to 29.

## K  FAST PARAMETER SET TESTS

We list the chosen parameter sets for Fast-100, Fast-1k and Fast-10k in Table 30. We plot the difference between the distance of the closest adversarial examples and the true decision boundary

Table 6: Parameters of heuristic attacks.

| Attack | Parameter Name | MNIST | | CIFAR10 | |
|---|---|---|---|---|---|
| | | Strong | Balanced | Strong | Balanced |
| BIM | Initial Search Factor | | | 0.75 | |
| | Initial Search Steps | | | 30 | |
| | Initial Search Factor | | | 0.75 | |
| | Binary Search Steps | | | 20 | |
| | #Iterations | 2k | 200 | 5k | 200 |
| | Learning Rate | 1e-3 | 1e-2 | 1e-5 | 1e-3 |
| Brendel & Bethge | Initial Attack | | | Blended Noise | |
| | Overshoot | | | 1.1 | |
| | LR Decay | | | 0.75 | |
| | LR Decay Every n Steps | | | 50 | |
| | #Iterations | 5k | 200 | 5k | 200 |
| | Learning Rate | 1e-3 | 1e-3 | 1e-5 | 1e-3 |
| | Momentum | | | 0.8 | |
| | Initial Directions | | | 1000 | |
| | Init Steps | | | 1000 | |
| Carlini & Wagner | Minimum $\tau$ | | | 1e-5 | |
| | Initial $\tau$ | | | 1 | |
| | $\tau$ Factor | 0.95 | 0.9 | 0.99 | 0.9 |
| | Initial Const | | | 1e-5 | |
| | Const Factor | | | 2 | |
| | Maximum Const | | | 20 | |
| | Reduce Const | | | False | |
| | Warm Start | | | True | |
| | Abort Early | | | True | |
| | Learning Rate | 1e-2 | 1e-2 | 1e-5 | 1e-4 |
| | Max Iterations | 1k | 100 | 5k | 100 |
| | $\tau$ Check Every n Steps | | | 1 | |
| | Const Check Every n Steps | | | 5 | |
| | Iter. Check Every n Steps | | | Disabled | |
| Deepfool | #Iterations | | | 5k | |
| | Candidates | | | 10 | |
| | Overshoot | | | 1e-5 | |
| | Loss | | | Logits | |
| FGSM | Initial Search Factor | | | 0.75 | |
| | Initial Search Steps | | | 30 | |
| | Initial Search Factor | | | 0.75 | |
| | Binary Search Steps | | | 20 | |
| | Starting $\varepsilon$ | | | 1 | |
| PGD | Initial Search Factor | | | 0.75 | |
| | Initial Search Steps | | | 30 | |
| | Initial Search Factor | | | 0.75 | |
| | Binary Search Steps | | | 20 | |
| | #Iterations | 5k | 200 | 5k | 200 |
| | Learning Rate | 1e-4 | 1e-3 | 1e-4 | 1e-3 |
| | Random Initialization | | | True | |
| Uniform Noise | Initial Search Factor | | | 0.75 | |
| | Initial Search Steps | | | 30 | |
| | Initial Search Factor | | | 0.75 | |
| | Binary Search Steps | | | 20 | |
| | Runs | 8k | 200 | 8k | 200 |

Table 7: Parameter set used to initialize MIPVerify for MNIST. All other parameters are identical to the strong MNIST attack parameter set.

| Attack Name | Parameter Name | Value |
|---|---|---|
| BIM | #Iterations | 5k |
| | Learning Rate | 1e-5 |
| Brendel & Bethge | Learning Rate | 1e-3 |
| Carlini & Wagner | Tau Factor | 0.99 |
| | Learning Rate | 1e-3 |
| | #Iterations | 5k |

Table 8: Parameters of MIPVerify.

| Parameter Name | Value | |
|---|---|---|
| | Exploration | Main |
| Absolute Tolerance | 1e-5 | |
| Relative Tolerance | 1e-10 | |
| Threads | 1 | |
| Timeout (s) | 120 | 7200 |
| Tightening Absolute Tolerance | 1e-4 | |
| Tightening Relative Tolerance | 1e-10 | |
| Tightening Timeout (s) | 20 | 240 |
| Tightening Threads | 1 | |

Table 9: MIPVerify bound tightness statistics.

| Architecture | Training | % Tight |
|---|---|---|
| MNIST A | Standard | 95.40% |
| | Adversarial | 99.60% |
| | ReLU | 82.46% |
| MNIST B | Standard | 74.61% |
| | Adversarial | 85.68% |
| | ReLU | 75.55% |
| MNIST C | Standard | 86.21% |
| | Adversarial | 97.28% |
| | ReLU | 95.63% |
| CIFAR10 A | Standard | 81.18% |
| | Adversarial | 82.50% |
| | ReLU | 92.73% |
| CIFAR10 B | Standard | 56.32% |
| | Adversarial | 58.88% |
| | ReLU | 81.67% |
| CIFAR10 C | Standard | 100.00% |
| | Adversarial | 100.00% |
| | ReLU | 100.00% |

Table 10: Expected vs true quantile for MNIST strong with 5-fold cross validation.

| Architecture | Training | Expected Quantile | True Quantile |
|---|---|---|---|
| A | Standard | 1.00%
50.00%
99.00% | 0.99±1.02%
49.93±2.35%
95.60±3.77% |
| | Adversarial | 1.00%
50.00%
99.00% | 1.11±0.53%
50.25±1.58%
89.84±6.42% |
| | ReLU | 1.00%
50.00%
99.00% | 1.11±0.45%
50.02±1.72%
91.95±5.64% |
| B | Standard | 1.00%
50.00%
99.00% | 1.07±0.48%
49.80±0.76%
97.76±0.71% |
| | Adversarial | 1.00%
50.00%
99.00% | 1.22±1.01%
49.88±4.63%
98.10±0.36% |
| | ReLU | 1.00%
50.00%
99.00% | 1.04±0.77%
49.98±3.17%
97.69±1.41% |
| C | Standard | 1.00%
50.00%
99.00% | 1.07±0.37%
50.17±1.64%
98.73±0.42% |
| | Adversarial | 1.00%
50.00%
99.00% | 1.05±0.29%
49.87±3.58%
99.00±0.47% |
| | ReLU | 1.00%
50.00%
99.00% | 1.06±0.67%
50.02±1.85%
93.99±3.51% |

Table 11: Expected vs true quantile for MNIST balanced with 5-fold cross validation.

| Architecture | Training | Expected Quantile | True Quantile |
|---|---|---|---|
| A | Standard | 1.00%
50.00%
99.00% | 1.30±0.79%
49.98±3.10%
93.99±2.59% |
| | Adversarial | 1.00%
50.00%
99.00% | 0.97±0.40%
50.12±1.14%
90.44±1.90% |
| | ReLU | 1.00%
50.00%
99.00% | 1.02±0.31%
50.02±1.05%
95.10±2.82% |
| B | Standard | 1.00%
50.00%
99.00% | 1.03±0.36%
49.98±0.70%
98.88±0.45% |
| | Adversarial | 1.00%
50.00%
99.00% | 1.17±0.97%
50.17±4.54%
98.69±0.59% |
| | ReLU | 1.00%
50.00%
99.00% | 1.04±0.49%
50.34±2.49%
98.73±0.53% |
| C | Standard | 1.00%
50.00%
99.00% | 1.07±0.33%
49.98±0.91%
98.88±0.55% |
| | Adversarial | 1.00%
50.00%
99.00% | 1.10±0.37%
50.12±4.15%
99.00±0.35% |
| | ReLU | 1.00%
50.00%
99.00% | 1.06±0.67%
50.12±2.67%
98.62±0.50% |

Table 12: Expected vs true quantile for CIFAR10 strong with 5-fold cross validation.

| Architecture | Training | Expected Quantile | True Quantile |
|---|---|---|---|
| A | Standard | 1.00%
50.00%
99.00% | 1.09±0.86%
50.09±1.84%
98.82±0.63% |
| | Adversarial | 1.00%
50.00%
99.00% | 1.05±0.23%
49.86±3.59%
98.90±0.62% |
| | ReLU | 1.00%
50.00%
99.00% | 0.97±0.41%
49.93±3.42%
97.66±1.35% |
| B | Standard | 1.00%
50.00%
99.00% | 0.98±0.18%
49.91±1.18%
98.84±0.56% |
| | Adversarial | 1.00%
50.00%
99.00% | 0.91±0.48%
50.00±3.58%
98.69±0.72% |
| | ReLU | 1.00%
50.00%
99.00% | 1.10±0.72%
49.98±2.21%
98.85±0.61% |
| C | Standard | 1.00%
50.00%
99.00% | 0.93±0.60%
50.00±1.86%
98.71±0.71% |
| | Adversarial | 1.00%
50.00%
99.00% | 1.09±0.17%
50.14±2.63%
98.27±0.81% |
| | ReLU | 1.00%
50.00%
99.00% | 1.01±0.62%
50.02±2.09%
96.17±2.40% |

Table 13: Expected vs true quantile for CIFAR10 balanced with 5-fold cross validation.

| Architecture | Training | Expected Quantile | True Quantile |
|---|---|---|---|
| A | Standard | 1.00%
50.00%
99.00% | 0.95±0.61%
50.32±2.38%
98.87±0.59% |
| | Adversarial | 1.00%
50.00%
99.00% | 1.05±0.23%
50.23±2.65%
98.81±0.96% |
| | ReLU | 1.00%
50.00%
99.00% | 4.14±5.32%
50.37±1.02%
94.62±2.87% |
| B | Standard | 1.00%
50.00%
99.00% | 1.07±0.46%
49.91±2.78%
98.93±0.73% |
| | Adversarial | 1.00%
50.00%
99.00% | 1.13±0.57%
50.18±2.05%
98.82±0.71% |
| | ReLU | 1.00%
50.00%
99.00% | 1.23±0.38%
50.11±0.38%
98.77±0.51% |
| C | Standard | 1.00%
50.00%
99.00% | 0.98±0.50%
50.09±2.21%
98.85±0.43% |
| | Adversarial | 1.00%
50.00%
99.00% | 1.09±0.26%
49.96±2.72%
98.86±0.32% |
| | ReLU | 1.00%
50.00%
99.00% | 1.01±0.36%
49.93±1.60%
97.93±0.63% |

Table 14: Performance of the strong attack set on MNIST.

| Architecture | Training | Success Rate | Difference | % Below 1/255 | $R^2$ |
|---|---|---|---|---|---|
| MNIST A | Standard | 100.00% | 1.51% | 98.16% | 0.996 |
| | Adversarial | 100.00% | 2.48% | 81.43% | 0.994 |
| | ReLU | 100.00% | 2.14% | 84.33% | 0.995 |
| MNIST B | Standard | 100.00% | 3.38% | 97.36% | 0.995 |
| | Adversarial | 100.00% | 4.34% | 75.09% | 0.991 |
| | ReLU | 100.00% | 4.80% | 68.02% | 0.992 |
| MNIST C | Standard | 100.00% | 4.52% | 96.92% | 0.996 |
| | Adversarial | 100.00% | 8.76% | 48.78% | 0.981 |
| | ReLU | 100.00% | 4.84% | 68.24% | 0.988 |

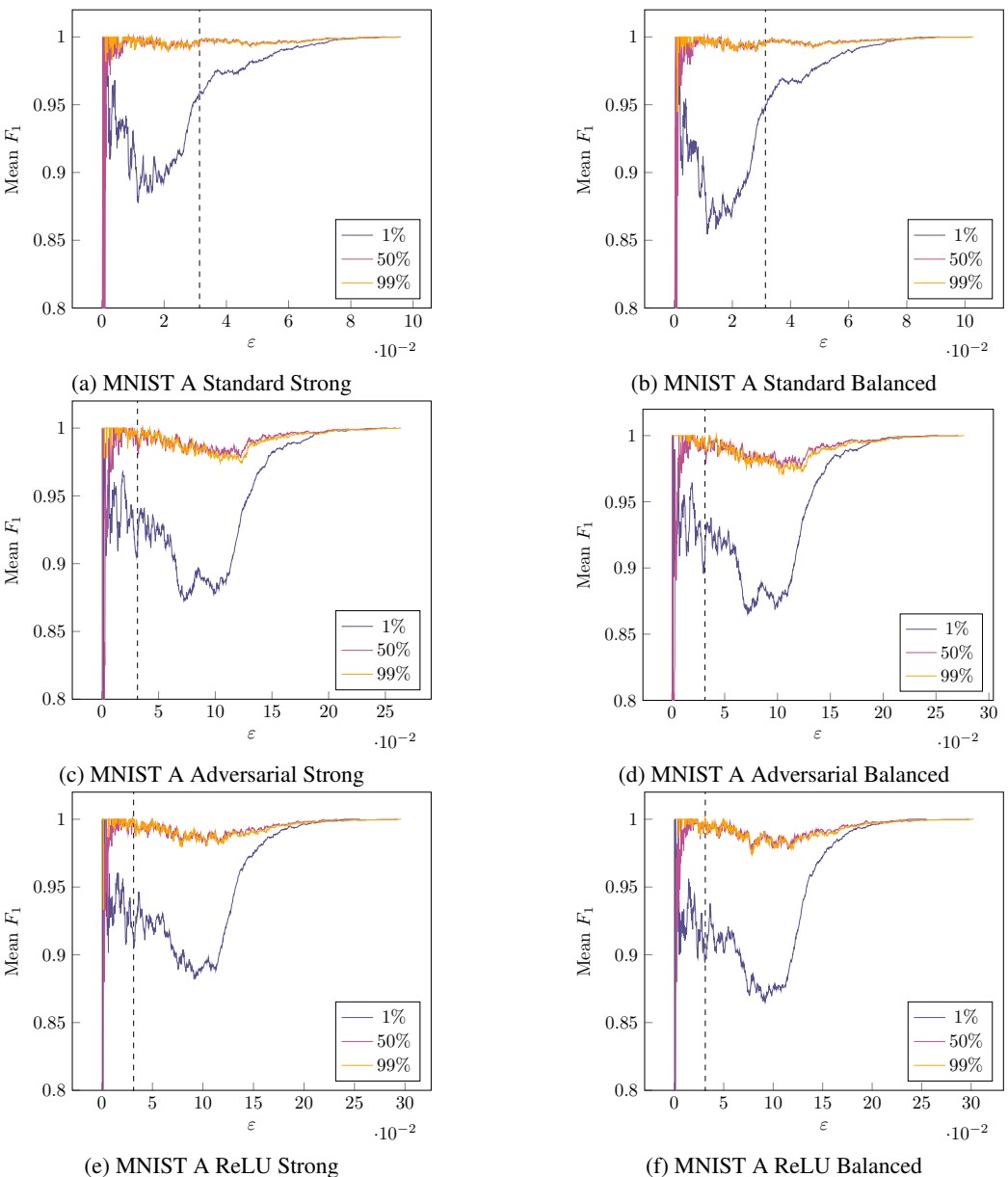

Figure 3: $F_1$ scores in relation to $\varepsilon$ for MNIST A for each considered percentile. For ease of visualization, we set the graph cutoff at $F_1 = 0.8$. We also mark 8/255 (a common choice for $\varepsilon$) with a dotted line.

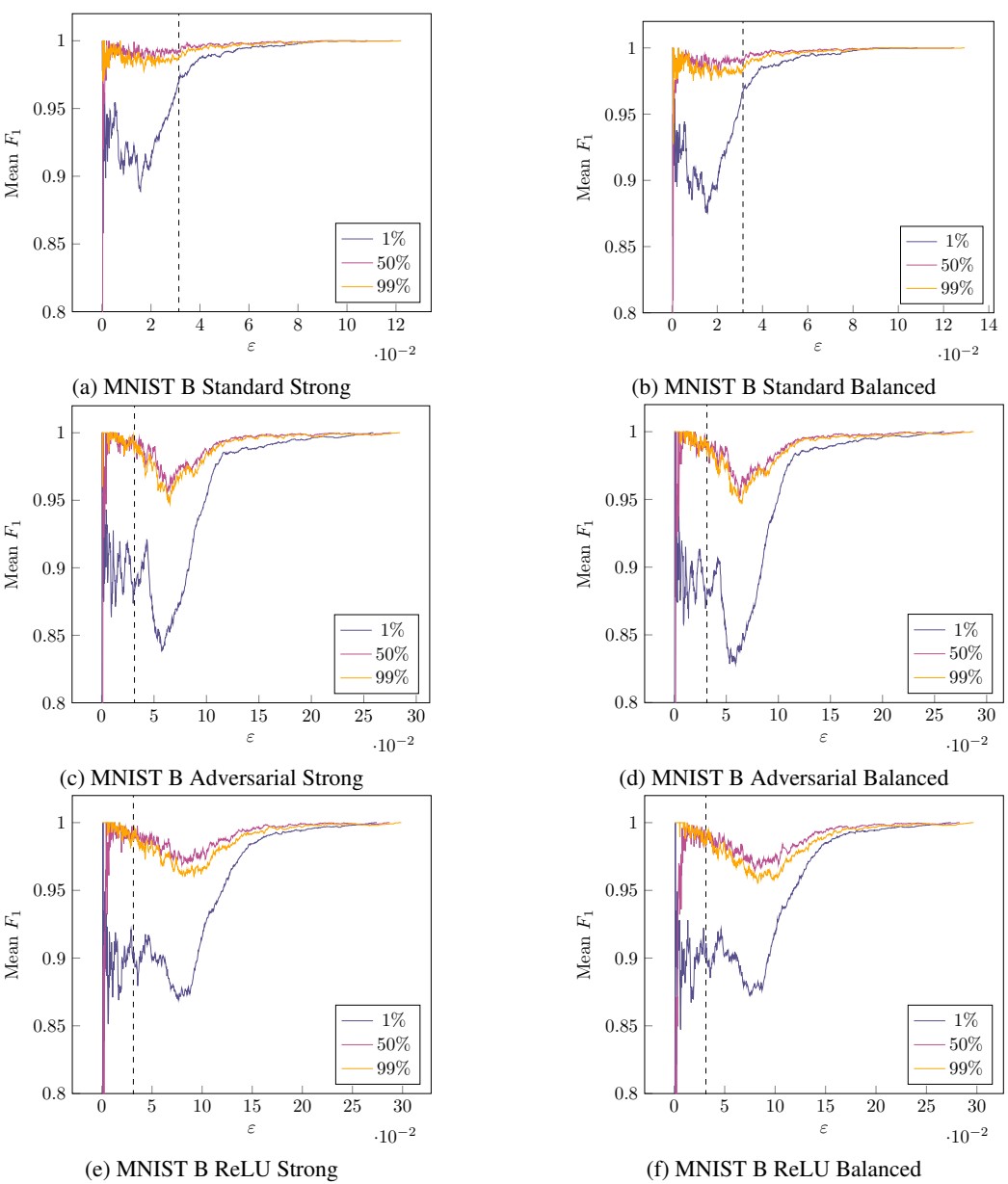

Figure 4: $F_1$ scores in relation to $\varepsilon$ for MNIST B for each considered percentile. For ease of visualization, we set the graph cutoff at $F_1 = 0.8$. We also mark 8/255 (a common choice for $\varepsilon$) with a dotted line.

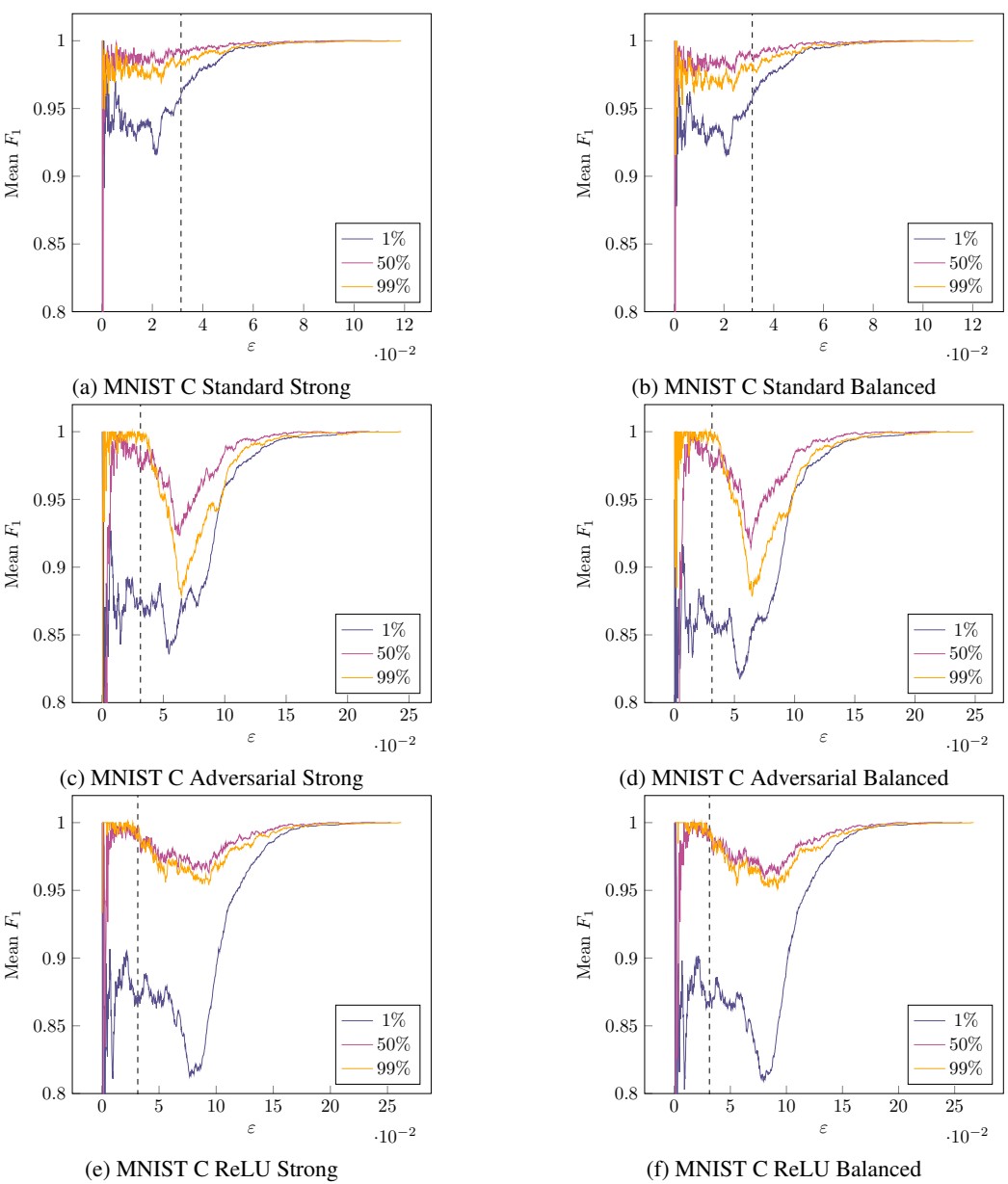

(a) MNIST C Standard Strong

(b) MNIST C Standard Balanced

(c) MNIST C Adversarial Strong

(d) MNIST C Adversarial Balanced

(e) MNIST C ReLU Strong

(f) MNIST C ReLU Balanced

Figure 5: $F_1$ scores in relation to $\varepsilon$ for MNIST C for each considered percentile. For ease of visualization, we set the graph cutoff at $F_1 = 0.8$. We also mark 8/255 (a common choice for $\varepsilon$) with a dotted line.

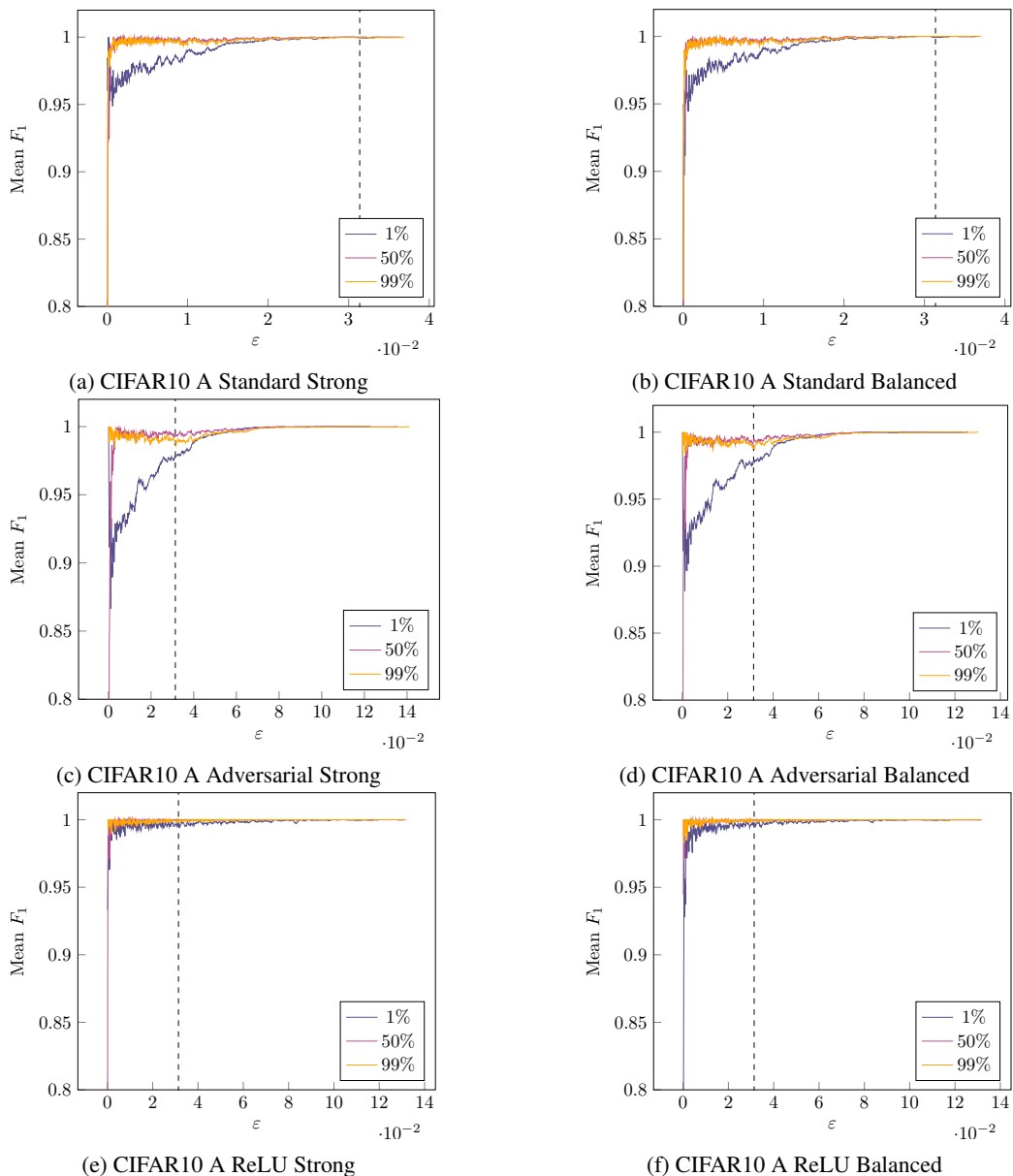

Figure 6: $F_1$ scores in relation to $\varepsilon$ for CIFAR10 A for each considered percentile. For ease of visualization, we set the graph cutoff at $F_1 = 0.8$. We also mark 8/255 (a common choice for $\varepsilon$) with a dotted line.

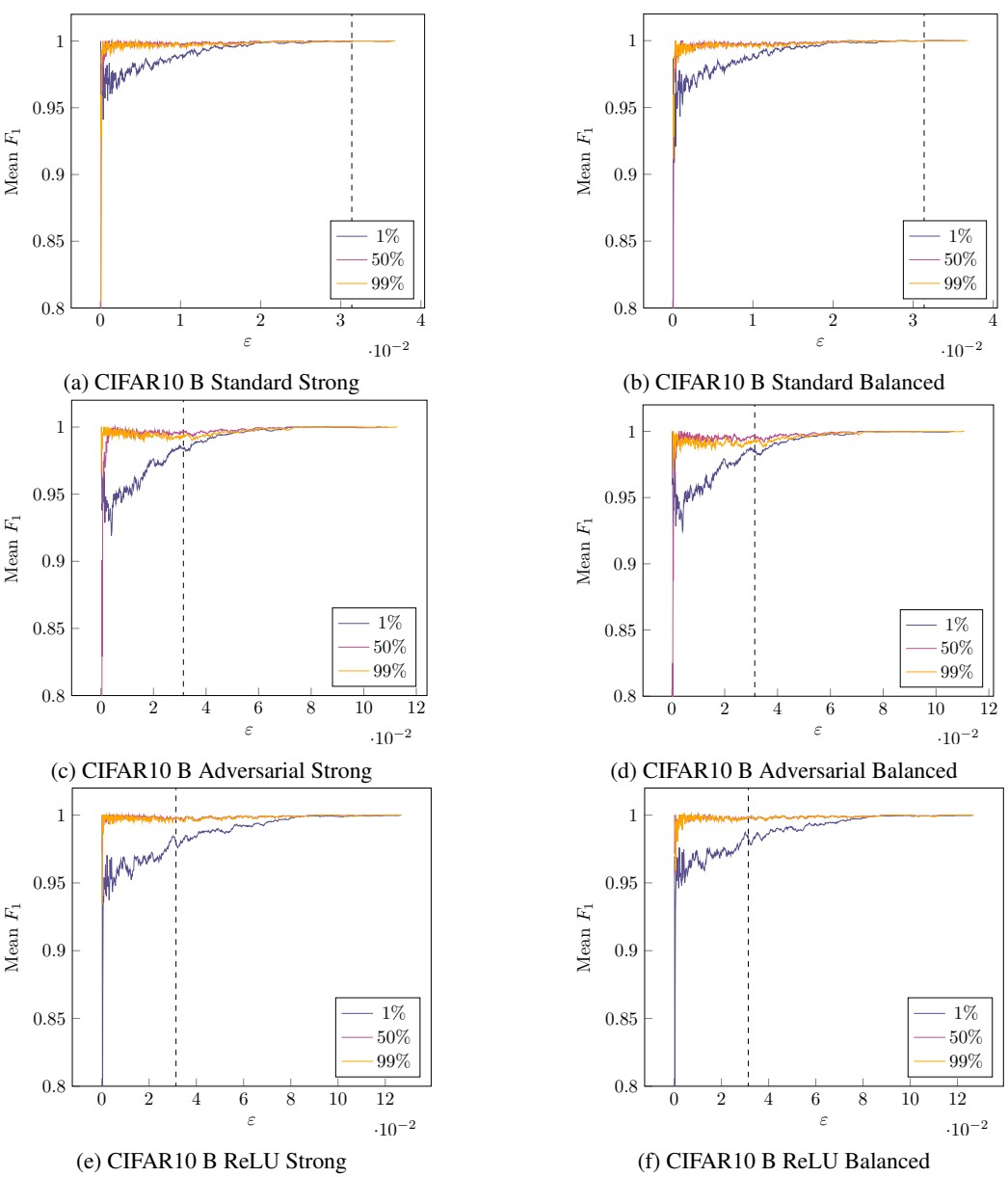

Figure 7: $F_1$ scores in relation to $\varepsilon$ for CIFAR10 B for each considered percentile. For ease of visualization, we set the graph cutoff at $F_1 = 0.8$. We also mark 8/255 (a common choice for $\varepsilon$) with a dotted line.

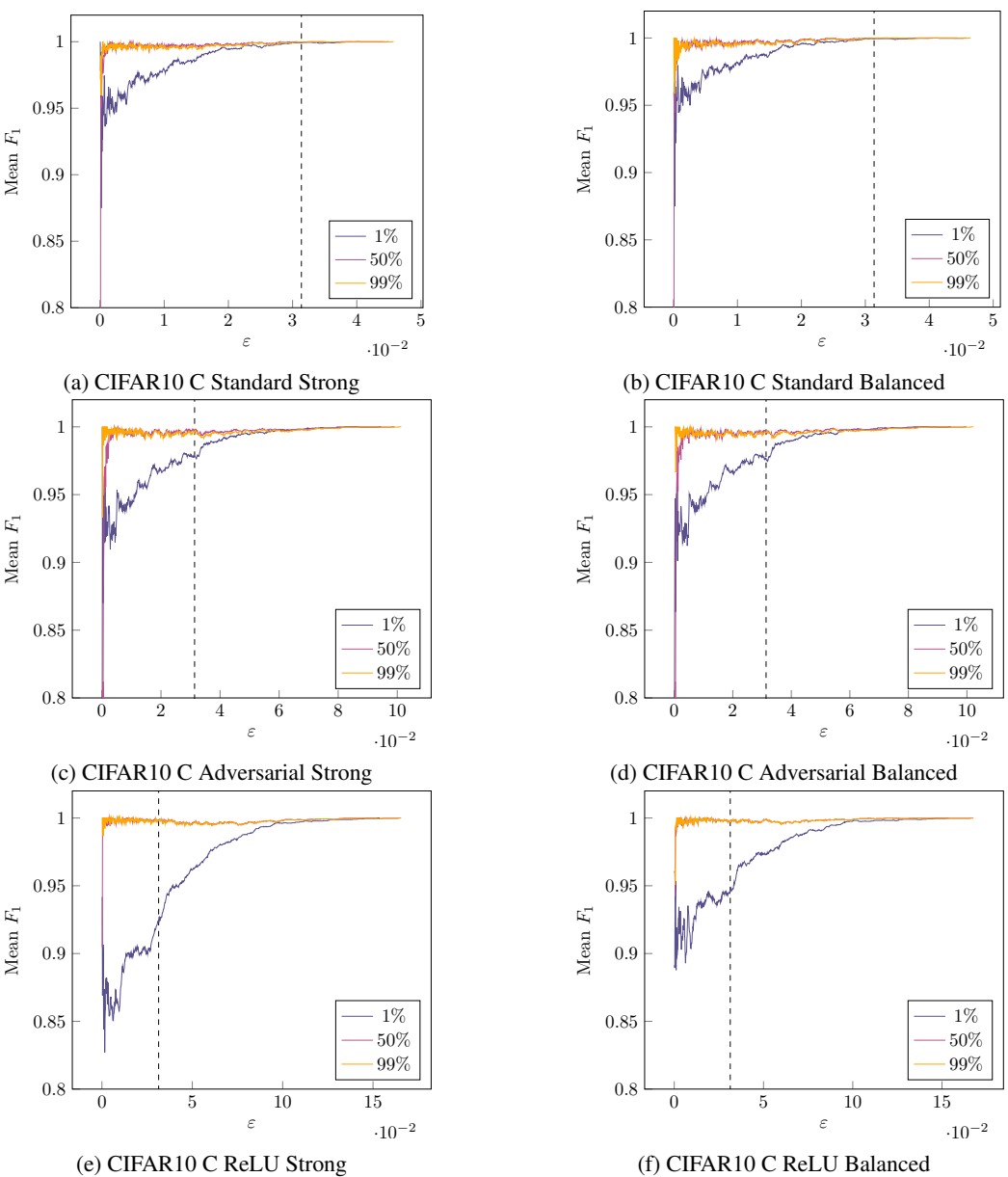

Figure 8: $F_1$ scores in relation to $\varepsilon$ for CIFAR10 C for each considered percentile. For ease of visualization, we set the graph cutoff at $F_1 = 0.8$. We also mark 8/255 (a common choice for $\varepsilon$) with a dotted line.

Table 15: Performance of the balanced attack set on MNIST.

| Architecture | Training | Success Rate | Difference | % Below 1/255 | $R^2$ |
|---|---|---|---|---|---|
| MNIST A | Standard | 100.00% | 1.68% | 97.94% | 0.995 |
| | Adversarial | 100.00% | 2.87% | 77.64% | 0.993 |
| | ReLU | 100.00% | 2.55% | 80.86% | 0.993 |
| MNIST B | Standard | 100.00% | 4.09% | 96.55% | 0.995 |
| | Adversarial | 100.00% | 4.90% | 72.60% | 0.988 |
| | ReLU | 100.00% | 5.53% | 62.96% | 0.989 |
| MNIST C | Standard | 100.00% | 5.43% | 96.04% | 0.995 |
| | Adversarial | 100.00% | 9.50% | 48.43% | 0.977 |
| | ReLU | 100.00% | 5.28% | 66.96% | 0.986 |

Table 16: Performance of the strong attack set on CIFAR10.

| Architecture | Training | Success Rate | Difference | % Below 1/255 | $R^2$ |
|---|---|---|---|---|---|
| CIFAR10 A | Standard | 100.00% | 1.62% | 100.00% | 0.999 |
| | Adversarial | 100.00% | 4.42% | 95.88% | 0.995 |
| | ReLU | 100.00% | 0.26% | 100.00% | 1.000 |
| CIFAR10 B | Standard | 100.00% | 1.44% | 100.00% | 0.999 |
| | Adversarial | 100.00% | 3.17% | 97.69% | 0.997 |
| | ReLU | 100.00% | 1.38% | 98.81% | 0.999 |
| CIFAR10 C | Standard | 100.00% | 2.11% | 100.00% | 0.999 |
| | Adversarial | 100.00% | 3.10% | 97.14% | 0.996 |
| | ReLU | 100.00% | 2.35% | 96.12% | 0.990 |

Table 17: Performance of the balanced attack set on CIFAR10.

| Architecture | Training | Success Rate | Difference | % Below 1/255 | $R^2$ |
|---|---|---|---|---|---|
| CIFAR10 A | Standard | 100.00% | 1.71% | 100.00% | 0.999 |
| | Adversarial | 100.00% | 4.18% | 96.57% | 0.995 |
| | ReLU | 100.00% | 0.18% | 100.00% | 1.000 |
| CIFAR10 B | Standard | 100.00% | 1.53% | 100.00% | 0.999 |
| | Adversarial | 100.00% | 2.92% | 98.46% | 0.996 |
| | ReLU | 100.00% | 1.19% | 98.94% | 0.999 |
| CIFAR10 C | Standard | 100.00% | 2.06% | 100.00% | 0.999 |
| | Adversarial | 100.00% | 3.12% | 97.28% | 0.996 |
| | ReLU | 100.00% | 1.45% | 97.44% | 0.995 |

Table 18: Best pools of a given size by success rate and $R^2$ for MNIST strong.

| $n$ | Attacks | Success Rate | Difference | < 1/255 | $R^2$ |
|---|---|---|---|---|---|
| 1 | PGD | 100.00±0.00% | 10.98±4.41% | 51.83±27.78% | 0.975±0.010 |
| 2 | C&W, PGD | 100.00±0.00% | 7.99±3.31% | 60.68±25.43% | 0.986±0.005 |
| 3 | B&B, C&W, PGD | 100.00±0.00% | 4.71±1.97% | 77.97±15.52% | 0.989±0.004 |
| 4 | B&B, C&W, DF, PGD | 100.00±0.00% | 4.36±2.03% | 79.02±15.62% | 0.991±0.005 |
| 5 | No FGSM, Uniform | 100.00±0.00% | 4.09±2.02% | 79.81±15.70% | 0.992±0.005 |
| 6 | No Uniform | 100.00±0.00% | 4.09±2.02% | 79.81±15.70% | 0.992±0.005 |
| 7 | All | 100.00±0.00% | 4.09±2.02% | 79.81±15.70% | 0.992±0.005 |

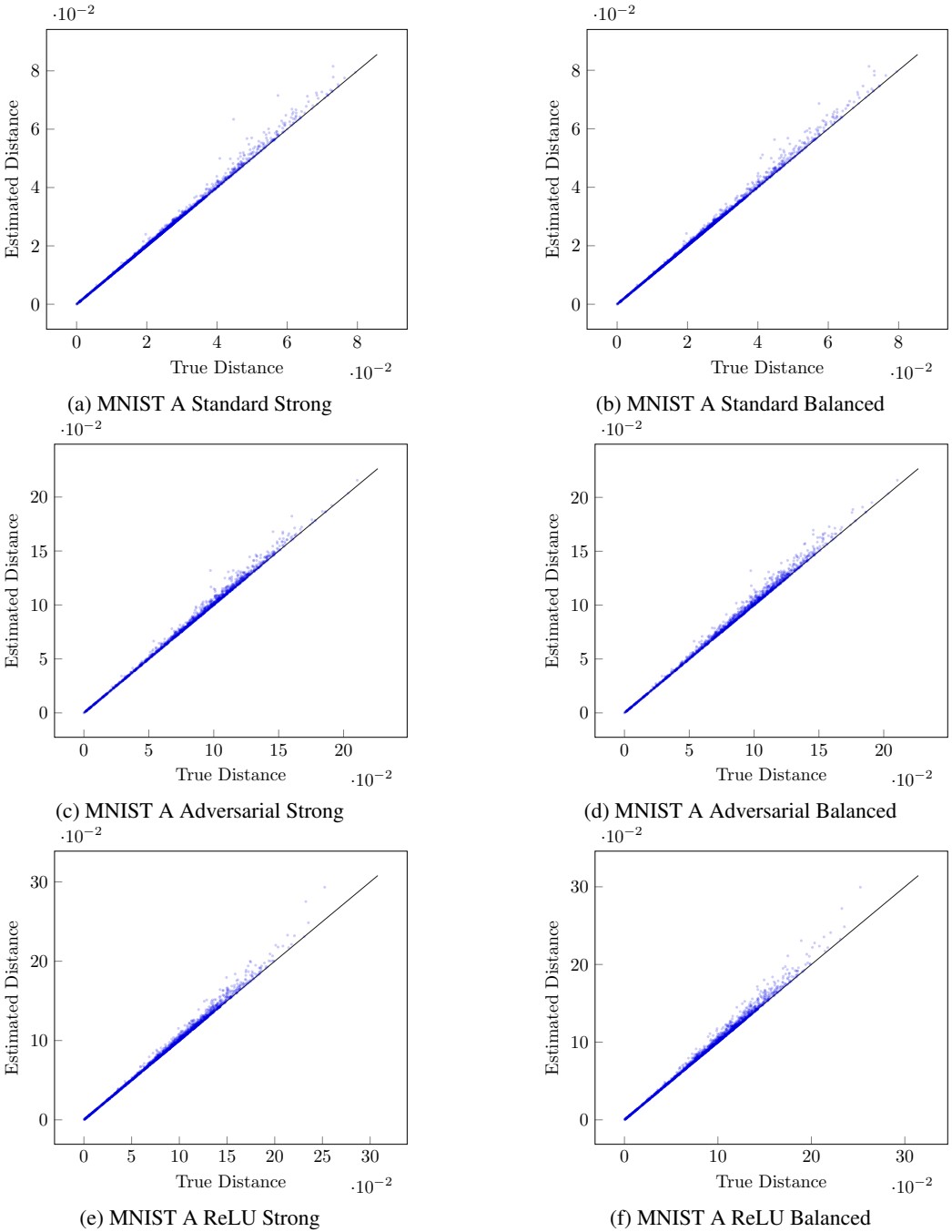

(a) MNIST A Standard Strong

(b) MNIST A Standard Balanced

(c) MNIST A Adversarial Strong

(d) MNIST A Adversarial Balanced

(e) MNIST A ReLU Strong

(f) MNIST A ReLU Balanced

Figure 9: Decision boundary distances found by the attack pools compared to those found by MIPVerify on MNIST A. The black line represents the theoretical optimum. Note that no samples are below the black line.

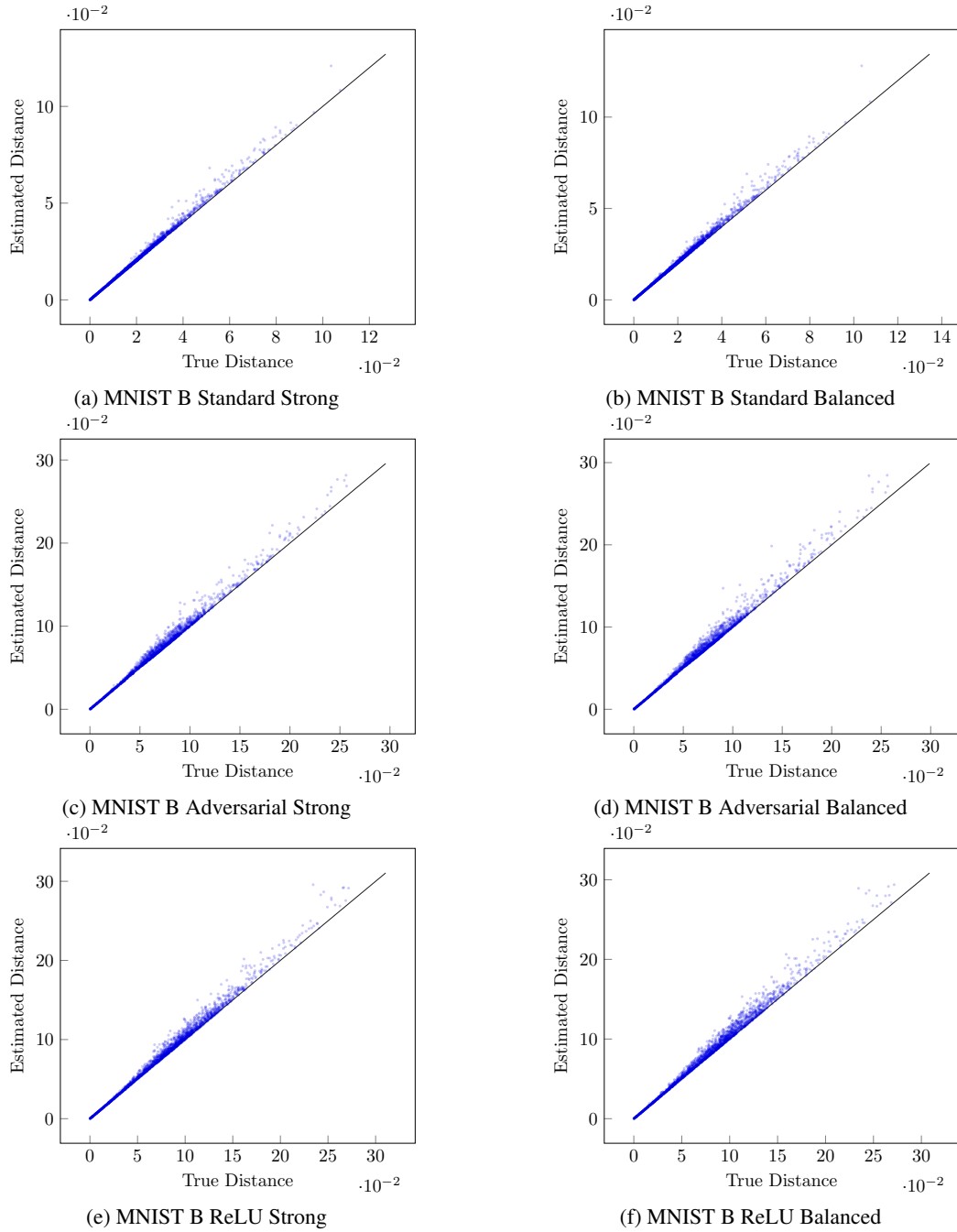

Figure 10: Decision boundary distances found by the attack pools compared to those found by MIPVerify on MNIST B. The black line represents the theoretical optimum. Note that no samples are below the black line.

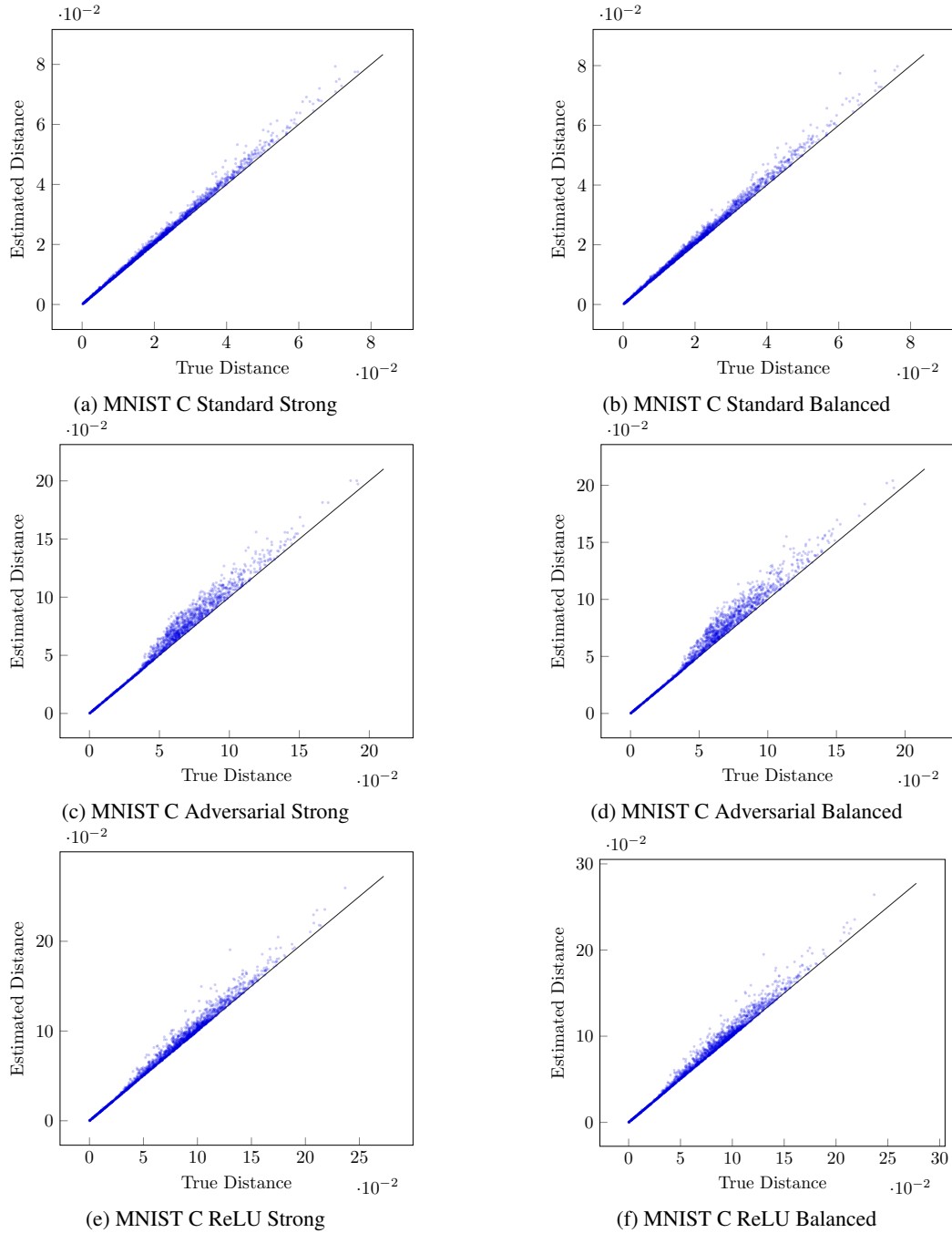

Figure 11: Decision boundary distances found by the attack pools compared to those found by MIPVerify on MNIST C. The black line represents the theoretical optimum. Note that no samples are below the black line.

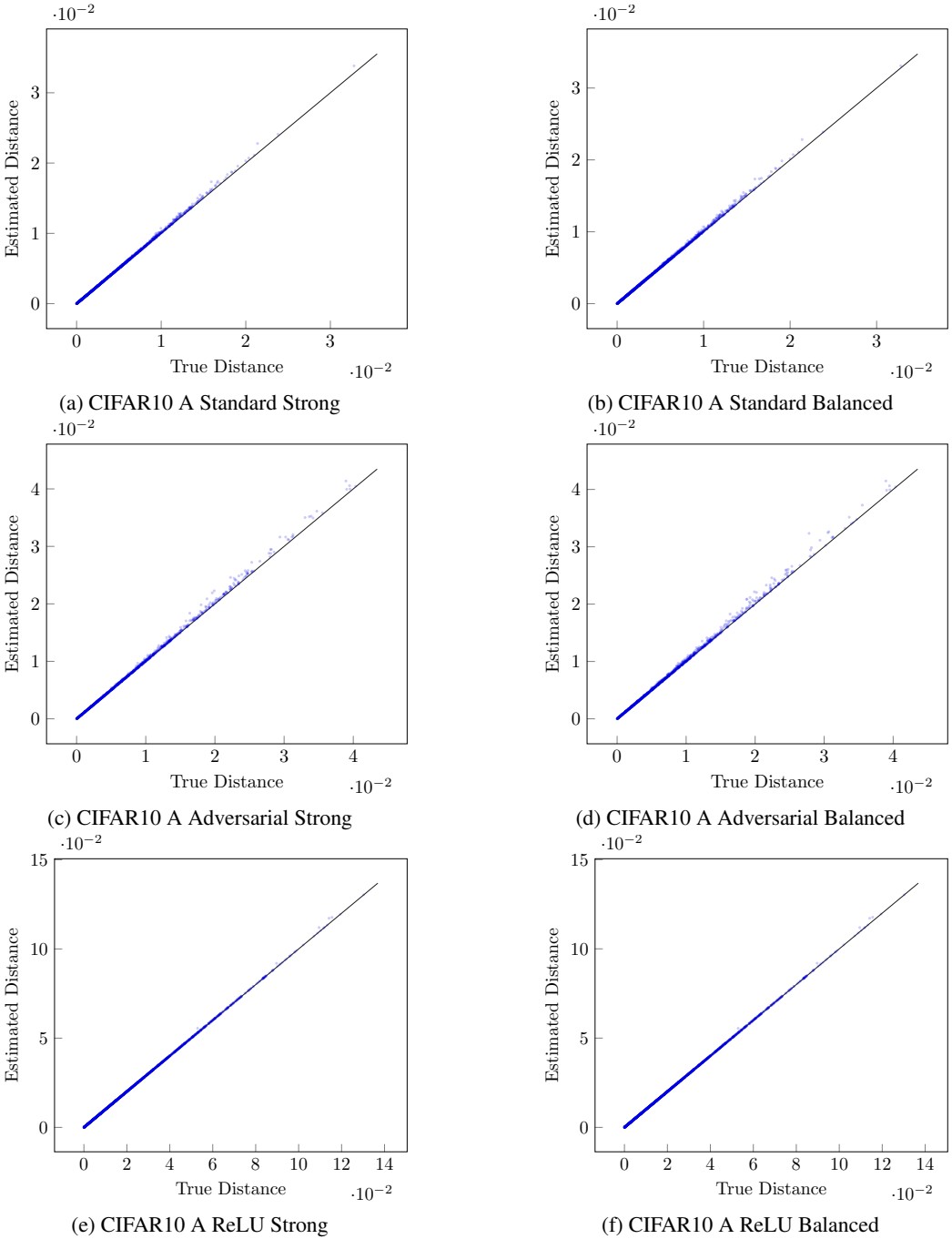

Figure 12: Decision boundary distances found by the attack pools compared to those found by MIPVerify on CIFAR10 A. The black line represents the theoretical optimum. Note that no samples are below the black line.

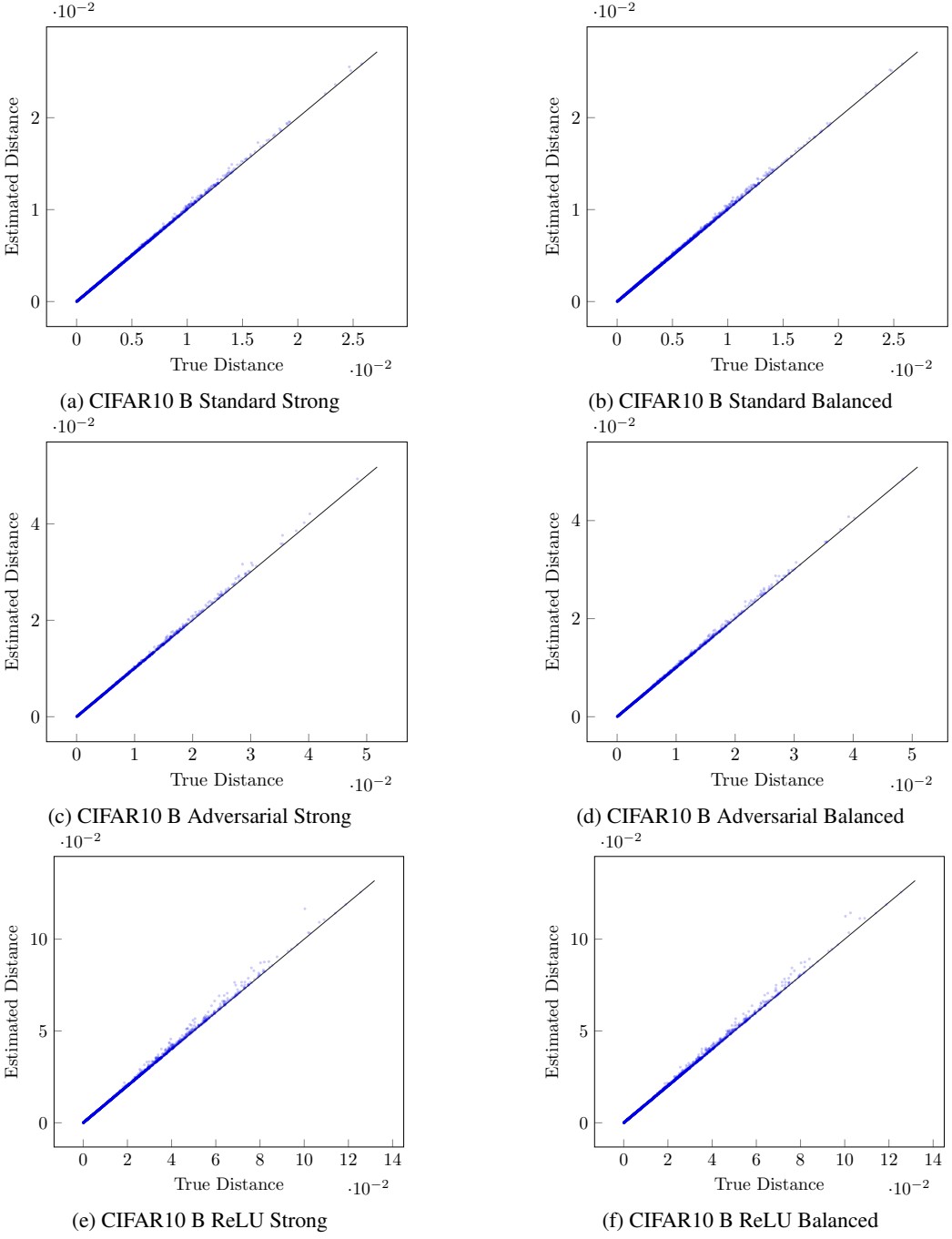

Figure 13: Decision boundary distances found by the attack pools compared to those found by MIPVerify on CIFAR10 B. The black line represents the theoretical optimum. Note that no samples are below the black line.

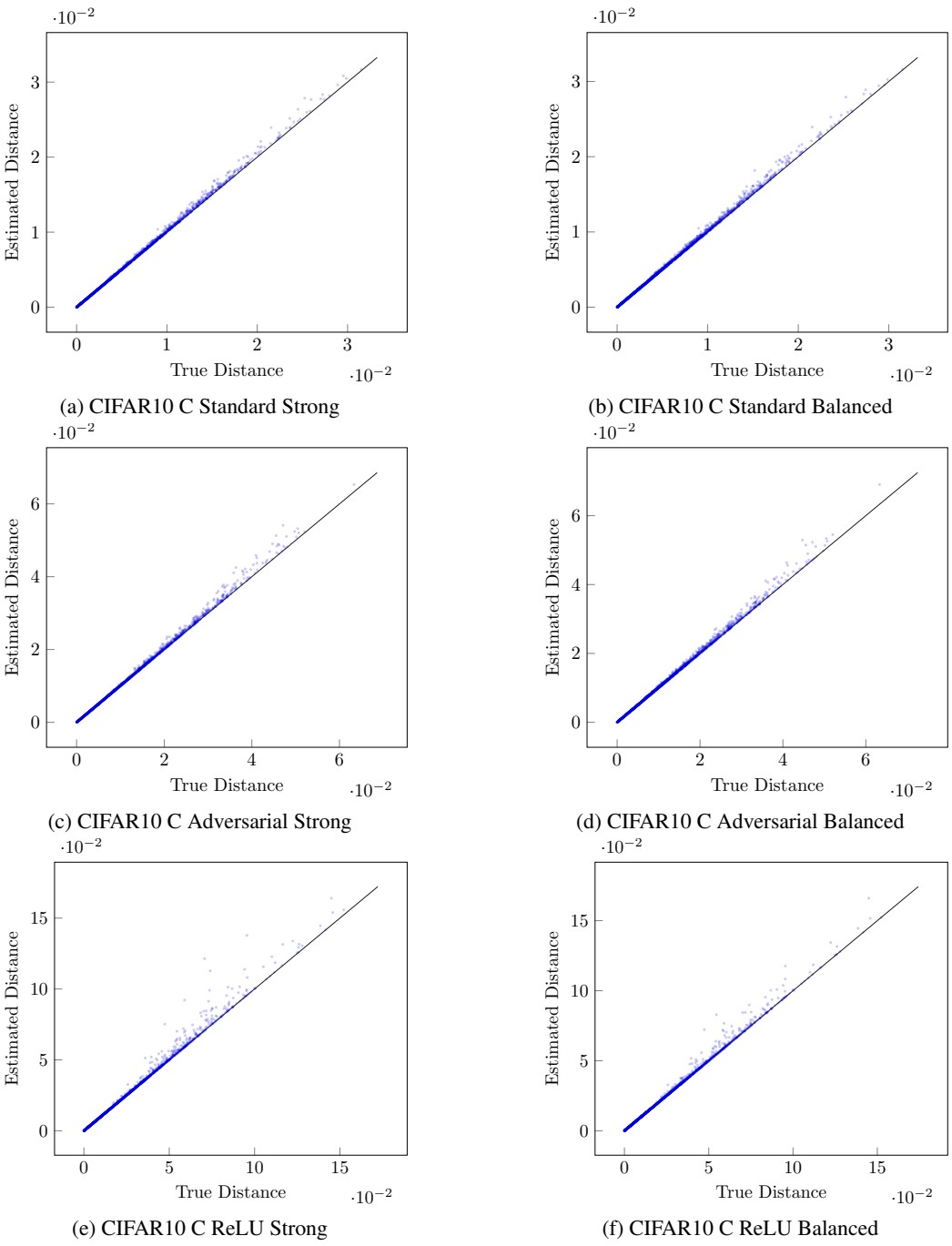

(a) CIFAR10 C Standard Strong

(b) CIFAR10 C Standard Balanced

(c) CIFAR10 C Adversarial Strong

(d) CIFAR10 C Adversarial Balanced

(e) CIFAR10 C ReLU Strong

(f) CIFAR10 C ReLU Balanced

Figure 14: Decision boundary distances found by the attack pools compared to those found by MIPVerify on CIFAR10 C. The black line represents the theoretical optimum. Note that no samples are below the black line.

Table 19: Best pools of a given size by success rate and $R^2$ for MNIST balanced.

| $n$ | Attacks | Success Rate | Difference | < 1/255 | $R^2$ |
|---|---|---|---|---|---|
| 1 | BIM | 100.00±0.00% | 11.72±4.18% | 50.92±26.43% | 0.965±0.010 |
| 2 | BIM, B&B | 100.00±0.00% | 6.11±2.28% | 73.23±15.90% | 0.980±0.007 |
| 3 | BIM, B&B, C&W | 100.00±0.00% | 5.29±2.06% | 75.72±16.10% | 0.986±0.005 |
| 4 | BIM, B&B, C&W, DF | 100.00±0.00% | 4.85±2.10% | 77.33±15.85% | 0.989±0.005 |
| 5 | No FGSM, Uniform | 100.00±0.00% | 4.65±2.16% | 77.78±16.08% | 0.990±0.006 |
| 6 | No Uniform | 100.00±0.00% | 4.65±2.16% | 77.78±16.08% | 0.990±0.006 |
| 7 | All | 100.00±0.00% | 4.65±2.16% | 77.78±16.08% | 0.990±0.006 |

Table 20: Best pools of a given size by success rate and $R^2$ for CIFAR10 strong.

| $n$ | Attacks | Success Rate | Difference | < 1/255 | $R^2$ |
|---|---|---|---|---|---|
| 1 | DF | 100.00±0.00% | 6.11±3.49% | 95.06±4.81% | 0.989±0.011 |
| 2 | DF, PGD | 100.00±0.00% | 4.71±2.37% | 96.32±3.56% | 0.995±0.007 |
| 3 | C&W, DF, PGD | 100.00±0.00% | 2.54±1.30% | 98.17±2.00% | 0.996±0.006 |
| 4 | B&B, C&W, DF, PGD | 100.00±0.00% | 2.21±1.16% | 98.40±1.63% | 0.997±0.003 |
| 5 | No FGSM, Uniform | 100.00±0.00% | 2.21±1.16% | 98.40±1.63% | 0.997±0.003 |
| 6 | No Uniform | 100.00±0.00% | 2.21±1.16% | 98.40±1.63% | 0.997±0.003 |
| 7 | All | 100.00±0.00% | 2.21±1.16% | 98.40±1.63% | 0.997±0.003 |

Table 21: Best pools of a given size by success rate and $R^2$ for CIFAR10 balanced.

| $n$ | Attacks | Success Rate | Difference | < 1/255 | $R^2$ |
|---|---|---|---|---|---|
| 1 | DF | 100.00±0.00% | 6.11±3.49% | 95.06±4.81% | 0.989±0.011 |
| 2 | B&B, DF | 100.00±0.00% | 2.52±1.51% | 98.23±1.81% | 0.995±0.004 |
| 3 | BIM, B&B, DF | 100.00±0.00% | 2.21±1.25% | 98.53±1.52% | 0.997±0.002 |
| 4 | BIM, B&B, C&W, DF | 100.00±0.00% | 2.06±1.16% | 98.73±1.32% | 0.998±0.002 |
| 5 | No FGSM, Uniform | 100.00±0.00% | 2.04±1.13% | 98.74±1.29% | 0.998±0.002 |
| 6 | No FGSM | 100.00±0.00% | 2.04±1.13% | 98.74±1.29% | 0.998±0.002 |
| 7 | All | 100.00±0.00% | 2.04±1.13% | 98.74±1.29% | 0.998±0.002 |

Table 22: Performance of individual attacks for MNIST strong.

| Attack | Success Rate | Difference | < 1/255 | $R^2$ |
|---|---|---|---|---|
| BIM | 100.00±0.00% | 10.90±4.42% | 53.57±28.07% | 0.966±0.012 |
| B&B | 99.99±0.01% | 18.50±7.09% | 58.78±9.91% | 0.812±0.044 |
| C&W | 100.00±0.00% | 17.52±2.74% | 48.02±21.28% | 0.910±0.024 |
| Deepfool | 100.00±0.00% | 21.59±7.73% | 44.15±20.02% | 0.923±0.027 |
| FGSM | 99.72±0.51% | 44.43±15.76% | 28.20±17.30% | 0.761±0.132 |
| PGD | 100.00±0.00% | 10.98±4.41% | 51.83±27.78% | 0.975±0.010 |
| Uniform | 99.52±0.91% | 414.47±140.54% | 0.82±0.55% | 0.623±0.138 |

Table 23: Performance of individual attacks for MNIST balanced.

| Attack | Success Rate | Difference | < 1/255 | $R^2$ |
|---|---|---|---|---|
| BIM | 100.00±0.00% | 11.72±4.18% | 50.92±26.43% | 0.965±0.010 |
| B&B | 99.99±0.03% | 18.65±7.29% | 58.43±9.61% | 0.812±0.039 |
| C&W | 100.00±0.00% | 22.55±3.83% | 38.95±22.49% | 0.904±0.025 |
| Deepfool | 100.00±0.00% | 21.59±7.73% | 44.15±20.02% | 0.923±0.027 |
| FGSM | 99.72±0.51% | 44.43±15.76% | 28.20±17.30% | 0.761±0.132 |
| PGD | 100.00±0.00% | 16.23±6.59% | 48.08±28.88% | 0.905±0.070 |
| Uniform | 98.66±1.90% | 521.61±181.40% | 0.57±0.38% | 0.484±0.122 |

Table 24: Performance of individual attacks for CIFAR10 strong.

| Attack | Success Rate | Difference | < 1/255 | $R^2$ |
|---|---|---|---|---|
| BIM | 91.96±7.40% | 19.97±5.95% | 80.32±12.97% | 0.934±0.041 |
| B&B | 100.00±0.00% | 508.66±196.37% | 42.74±7.85% | 0.174±0.074 |
| C&W | 99.98±0.02% | 10.67±3.64% | 90.09±5.51% | 0.926±0.030 |
| Deepfool | 100.00±0.00% | 6.11±3.49% | 95.06±4.81% | 0.989±0.011 |
| FGSM | 100.00±0.00% | 31.80±11.12% | 69.20±17.72% | 0.847±0.123 |
| PGD | 100.00±0.00% | 19.36±5.99% | 77.23±15.89% | 0.952±0.027 |
| Uniform | 99.99±0.02% | 1206.79±277.68% | 2.48±0.88% | 0.910±0.044 |

Table 25: Performance of individual attacks for CIFAR10 balanced.

| Attack | Success Rate | Difference | < 1/255 | $R^2$ |
|---|---|---|---|---|
| BIM | 100.00±0.00% | 19.23±5.92% | 77.33±15.89% | 0.954±0.025 |
| B&B | 100.00±0.00% | 50.64±52.17% | 81.20±10.68% | 0.615±0.349 |
| C&W | 99.89±0.09% | 17.44±4.01% | 84.82±8.51% | 0.923±0.026 |
| Deepfool | 100.00±0.00% | 6.11±3.49% | 95.06±4.81% | 0.989±0.011 |
| FGSM | 100.00±0.00% | 31.80±11.12% | 69.20±17.72% | 0.847±0.123 |
| PGD | 100.00±0.00% | 20.18±6.56% | 76.97±16.07% | 0.947±0.031 |
| Uniform | 99.85±0.26% | 1617.74±390.50% | 1.80±0.67% | 0.853±0.068 |

Table 26: Performance of pools without a specific attack for MNIST strong.

| Dropped Attack | Success Rate | Difference | < 1/255 | $R^2$ |
|---|---|---|---|---|
| None | 100.00±0.00% | 4.09±2.02% | 79.81±15.70% | 0.992±0.005 |
| BIM | 100.00±0.00% | 4.35±2.03% | 79.02±15.62% | 0.991±0.005 |
| B&B | 100.00±0.00% | 6.76±3.31% | 64.46±25.01% | 0.990±0.005 |
| C&W | 100.00±0.00% | 4.65±2.20% | 77.70±16.02% | 0.989±0.006 |
| Deepfool | 100.00±0.00% | 4.33±1.97% | 79.04±15.75% | 0.990±0.004 |
| FGSM | 100.00±0.00% | 4.09±2.02% | 79.81±15.70% | 0.992±0.005 |
| PGD | 100.00±0.00% | 4.26±1.99% | 79.36±15.59% | 0.991±0.004 |
| Uniform | 100.00±0.00% | 4.09±2.02% | 79.81±15.70% | 0.992±0.005 |

Table 27: Performance of pools without a specific attack for MNIST balanced.

| Dropped Attack | Success Rate | Difference | < 1/255 | $R^2$ |
|---|---|---|---|---|
| None | 100.00±0.00% | 4.65±2.16% | 77.78±16.08% | 0.990±0.006 |
| BIM | 100.00±0.00% | 5.13±2.27% | 76.14±15.98% | 0.988±0.007 |
| B&B | 100.00±0.00% | 7.93±3.69% | 60.79±25.99% | 0.987±0.006 |
| C&W | 100.00±0.00% | 4.93±2.22% | 77.05±15.96% | 0.988±0.006 |
| Deepfool | 100.00±0.00% | 5.03±2.14% | 76.34±16.36% | 0.988±0.005 |
| FGSM | 100.00±0.00% | 4.65±2.16% | 77.78±16.08% | 0.990±0.006 |
| PGD | 100.00±0.00% | 4.85±2.10% | 77.33±15.85% | 0.989±0.005 |
| Uniform | 100.00±0.00% | 4.65±2.16% | 77.78±16.08% | 0.990±0.006 |

Table 28: Performance of pools without a specific attack for CIFAR10 strong.

| Dropped Attack | Success Rate | Difference | < 1/255 | $R^2$ |
|---|---|---|---|---|
| None | 100.00±0.00% | 2.21±1.16% | 98.40±1.63% | 0.997±0.003 |
| BIM | 100.00±0.00% | 2.21±1.16% | 98.40±1.63% | 0.997±0.003 |
| B&B | 100.00±0.00% | 2.54±1.30% | 98.17±2.00% | 0.996±0.006 |
| C&W | 100.00±0.00% | 3.83±2.06% | 96.84±3.12% | 0.996±0.004 |
| Deepfool | 100.00±0.00% | 4.02±1.19% | 95.65±3.10% | 0.992±0.005 |
| FGSM | 100.00±0.00% | 2.21±1.16% | 98.40±1.63% | 0.997±0.003 |
| PGD | 100.00±0.00% | 2.50±1.48% | 98.11±1.93% | 0.995±0.005 |
| Uniform | 100.00±0.00% | 2.21±1.16% | 98.40±1.63% | 0.997±0.003 |

Table 29: Performance of pools without a specific attack for CIFAR10 balanced.

| Dropped Attack | Success Rate | Difference | < 1/255 | $R^2$ |
|---|---|---|---|---|
| None | 100.00±0.00% | 2.04±1.13% | 98.74±1.29% | 0.998±0.002 |
| BIM | 100.00±0.00% | 2.07±1.15% | 98.72±1.31% | 0.998±0.002 |
| B&B | 100.00±0.00% | 4.08±1.95% | 97.26±2.70% | 0.996±0.006 |
| C&W | 100.00±0.00% | 2.18±1.22% | 98.54±1.50% | 0.997±0.002 |
| Deepfool | 100.00±0.00% | 4.00±0.99% | 95.89±3.13% | 0.993±0.003 |
| FGSM | 100.00±0.00% | 2.04±1.13% | 98.74±1.29% | 0.998±0.002 |
| PGD | 100.00±0.00% | 2.06±1.16% | 98.73±1.32% | 0.998±0.002 |
| Uniform | 100.00±0.00% | 2.04±1.13% | 98.74±1.29% | 0.998±0.002 |

distance in Figures 15 to 23, while we plot the $R^2$ values in Figures 24 to 32. We do not study the Brendel & Bethge and the Carlini & Wagner attacks due to the fact that the number of model calls varies depending on how many inputs are attacked at the same time. Note that, for attacks that do not have the a 100% success rate, the mean adversarial example distance can increase with the number of steps as new adversarial examples (for inputs for which there were previously no successful adversarial examples) are added.

## L    RESULTS FOR ATTACKS AGAINST CA

We report the parameters for the variant of our attack in Table 31, while we report its success rate in Table 32. We set $\varepsilon = \varepsilon'$ equal to $\{0.025, 0.05, 0.1\}$ for MNIST and $\{2/255, 4/255, 8/255\}$ for CIFAR10. Note that, since Deepfool is a deterministic attack, no measures against randomizations were taken.

Table 30: Parameters for the Fast-100, Fast-1k and Fast-10k sets.

| Attack | Parameter Name | MNIST 100 | MNIST 1k | MNIST 10k | CIFAR10 100 | CIFAR10 1k | CIFAR10 10k |
|---|---|---|---|---|---|---|---|
| BIM | Initial Search Factor | | | N/A | | | |
| | Initial Search Steps | | | N/A | | | |
| | Binary Search Steps | 10 | 20 | 20 | 10 | 20 | 20 |
| | Starting $\varepsilon$ | | 0.5 | | | 0.1 | |
| | #Iterations | 10 | 50 | 500 | 10 | 50 | 500 |
| | Learning Rate | 0.1 | 0.01 | 1e-3 | 0.01 | 1e-3 | 1e-3 |
| Deepfool | #Iterations | 100 | 500 | 500 | | 500 | |
| | Candidates | | | 10 | | | |
| | Overshoot | 0.1 | 1e-5 | 1e-5 | | 1e-4 | |
| | Loss | | | Logits | | | |
| FGSM | Initial Search Factor | | 0.75 | | | 0.5 | |
| | Initial Search Steps | | 30 | | | 10 | |
| | Binary Search Steps | | | 20 | | | |
| | Starting $\varepsilon$ | | 1 | | | 0.1 | |
| PGD | Initial Search Factor | N/A | 0.5 | 0.5 | N/A | 0.5 | 0.75 |
| | Initial Search Steps | N/A | 10 | 10 | N/A | 10 | 30 |
| | Binary Search Steps | | 10 | | 10 | 10 | 20 |
| | Starting $\varepsilon$ | | 0.1 | | 0.1 | 0.1 | 1 |
| | #Iterations | 10 | 50 | 500 | 10 | 50 | 200 |
| | Learning Rate | 0.1 | 0.01 | 1e-3 | 0.01 | 1e-3 | 1e-3 |
| | Random Initialization | | | True | | | |
| Uniform Noise | Initial Search Factor | | 0.75 | | 0.75 | 0.75 | 0.25 |
| | Initial Search Steps | | 30 | | 30 | 30 | 5 |
| | Binary Search Steps | | 20 | | 20 | 20 | 15 |
| | Starting $\varepsilon$ | | 1 | | 1 | 1 | 0.5 |
| | Runs | 200 | 500 | 200 | 10 | 50 | 500 |

Table 31: Parameters for the variant of the PGD attack.

| Parameter | Value |
|---|---|
| $\lambda$ | $\{10^{-4}, 10^{-2}, 1, 10^2, 10^4\}$ |
| NES Estimation Samples | 200 |
| NES Estimation $\varepsilon$ | $10^{-4}$ |
| PGD Iterations | 400 |
| PGD $\varepsilon$ | $10^{-3}$ |

Table 32: Success rate of the pool composed of the anti-CA variant of PGD and uniform noise for the A architectures.

| $\varepsilon$ | $\varepsilon'$ Multiplier 1x | 2x | 5x | 10x |
|---|---|---|---|---|
| 0.025 | 0% | 23% | 95% | 100% |
| 0.05 | 3% | 41% | 97% | 100% |
| 0.1 | 2% | 50% | 96% | 100% |

(a) MNIST A Standard

| $\varepsilon$ | $\varepsilon'$ Multiplier 1x | 2x | 5x | 10x |
|---|---|---|---|---|
| 2/255 | 0% | 34% | 96% | 100% |
| 4/255 | 1% | 50% | 100% | 100% |
| 8/255 | 1% | 58% | 100% | 100% |

(b) CIFAR10 A Standard

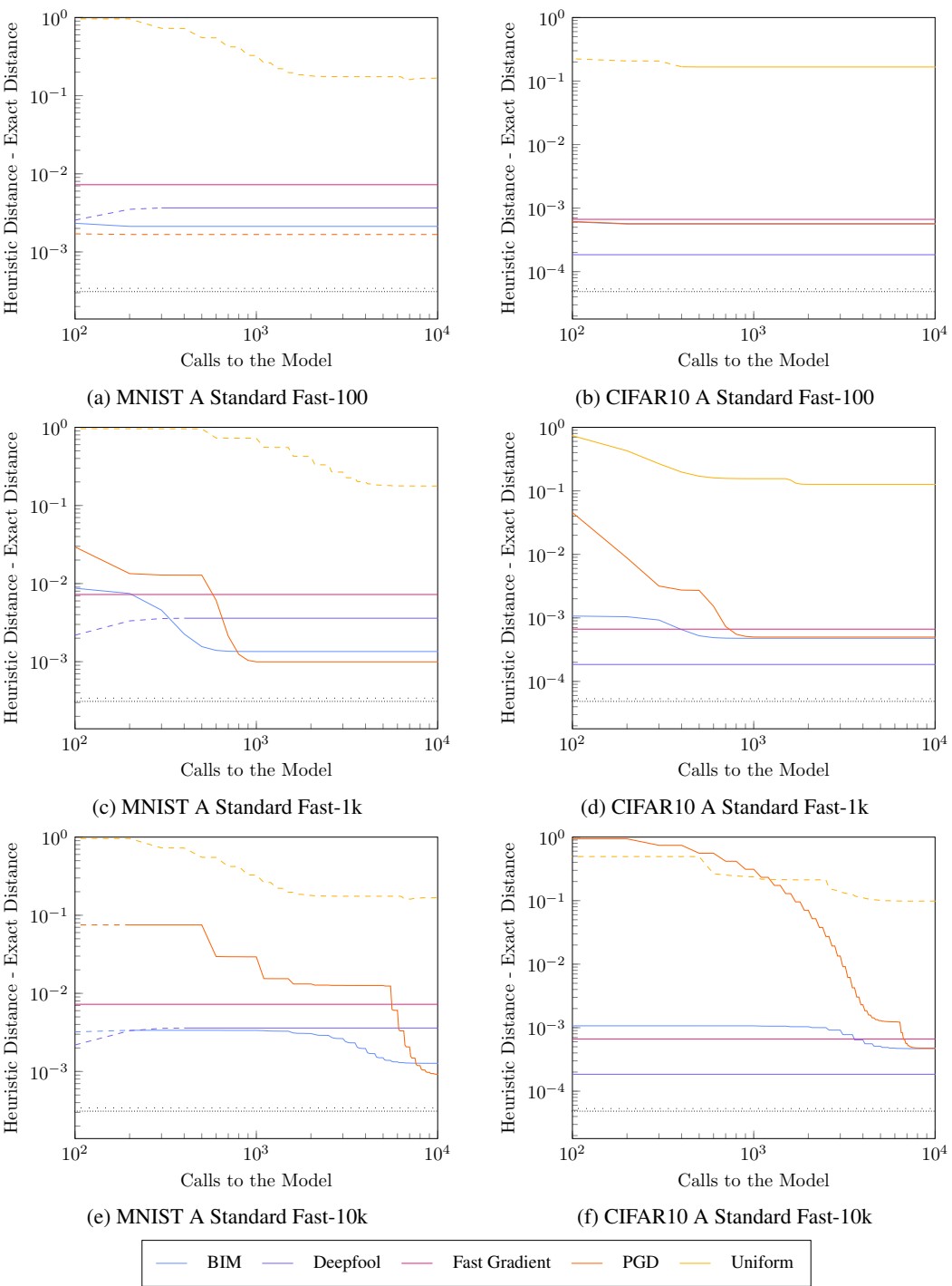

Figure 15: Mean difference between the distance of the closest adversarial examples and the exact decision boundary distance for MNIST & CIFAR10 A Standard. A dashed line means that the attack found adversarial examples (of any distance) for only some inputs, while the absence of a line means that the attack did not find any adversarial examples. The loosely and densely dotted black lines respectively represent the balanced and strong attack pools. Both axes are logarithmic.

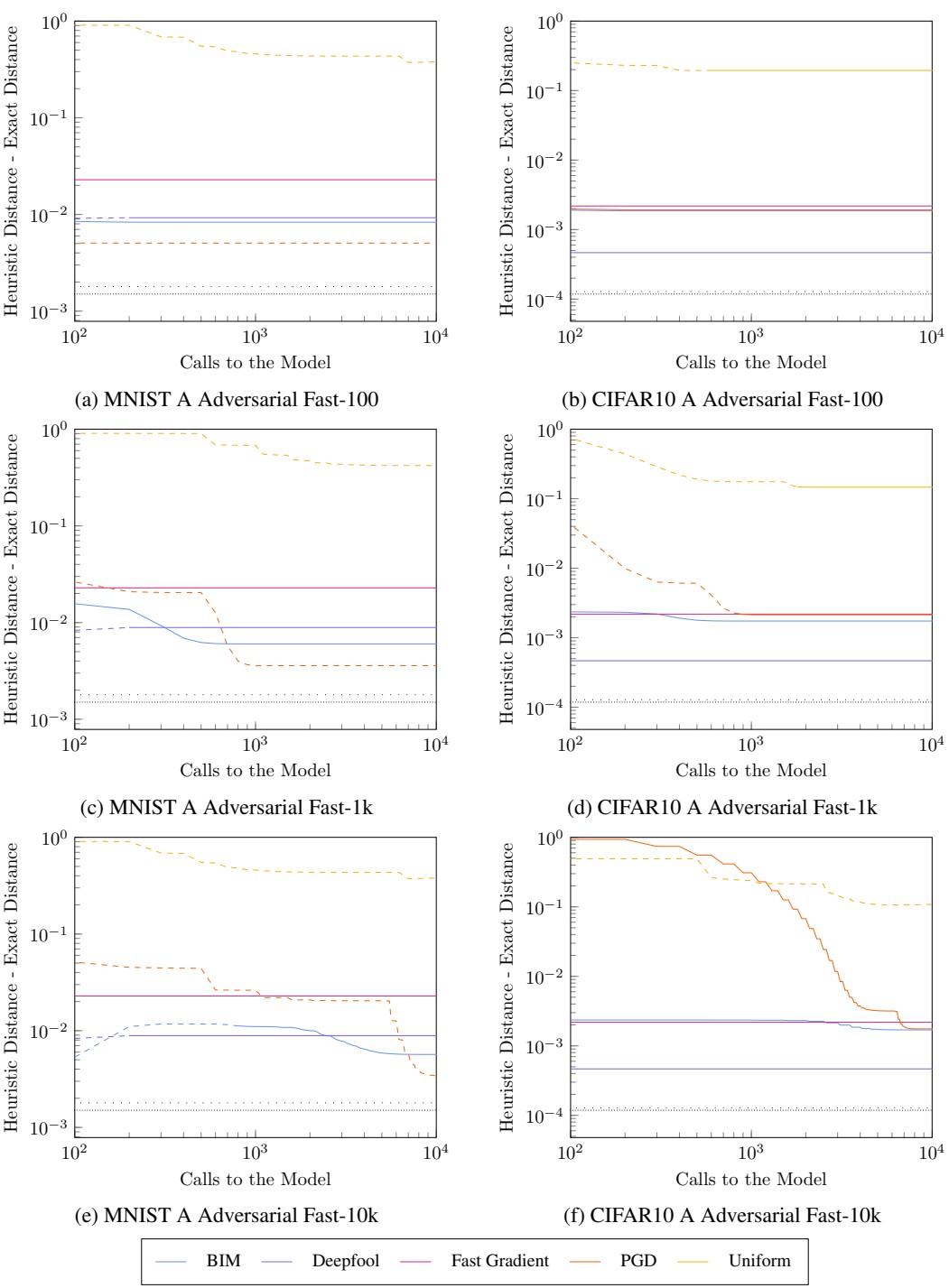

Figure 16: Mean difference between the distance of the closest adversarial examples and the exact decision boundary distance for MNIST & CIFAR10 A Adversarial. A dashed line means that the attack found adversarial examples (of any distance) for only some inputs, while the absence of a line means that the attack did not find any adversarial examples. The loosely and densely dotted black lines respectively represent the balanced and strong attack pools. Both axes are logarithmic.

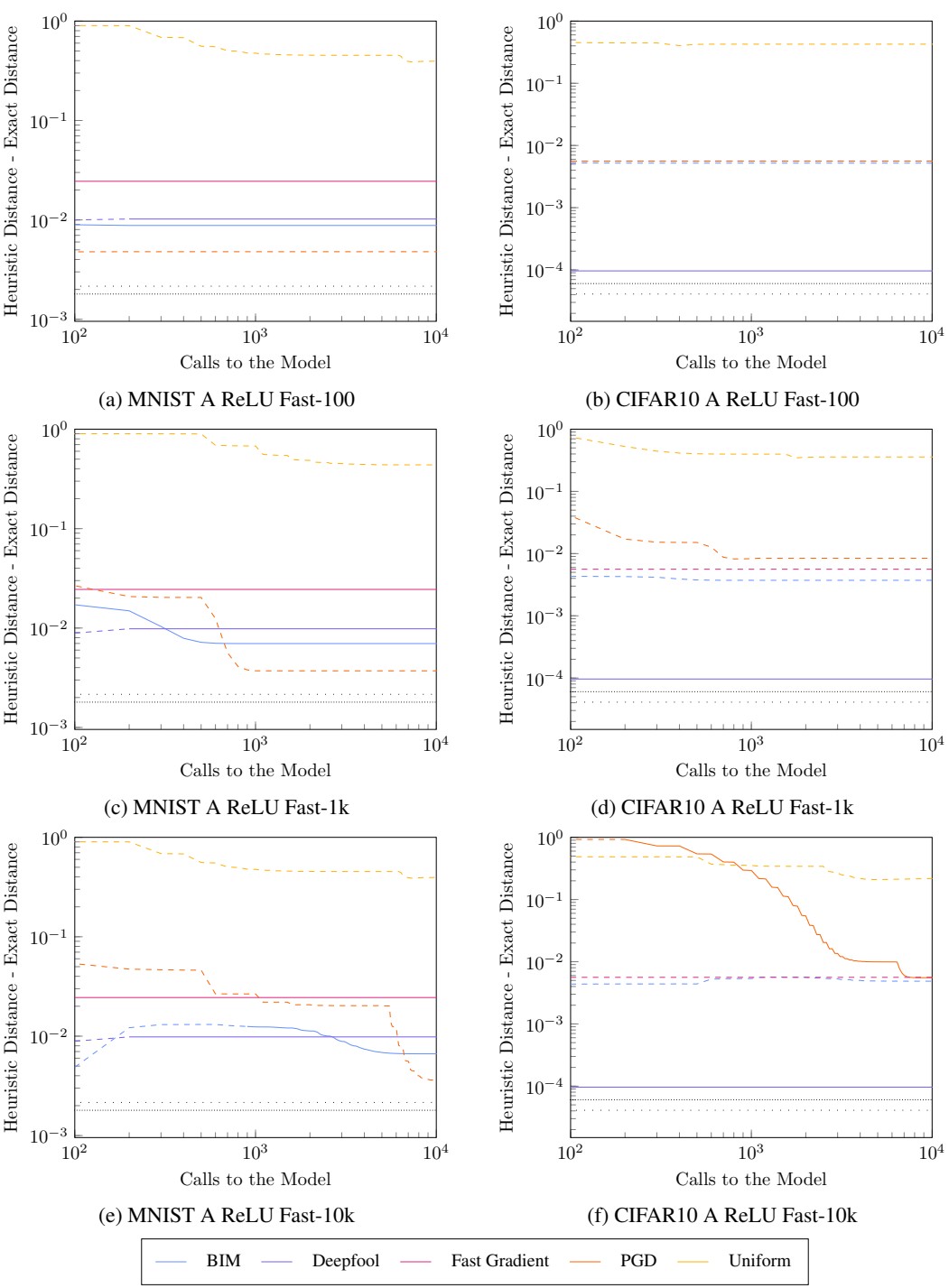

Figure 17: Mean difference between the distance of the closest adversarial examples and the exact decision boundary distance for MNIST & CIFAR10 A ReLU. A dashed line means that the attack found adversarial examples (of any distance) for only some inputs, while the absence of a line means that the attack did not find any adversarial examples. The loosely and densely dotted black lines respectively represent the balanced and strong attack pools. Both axes are logarithmic.

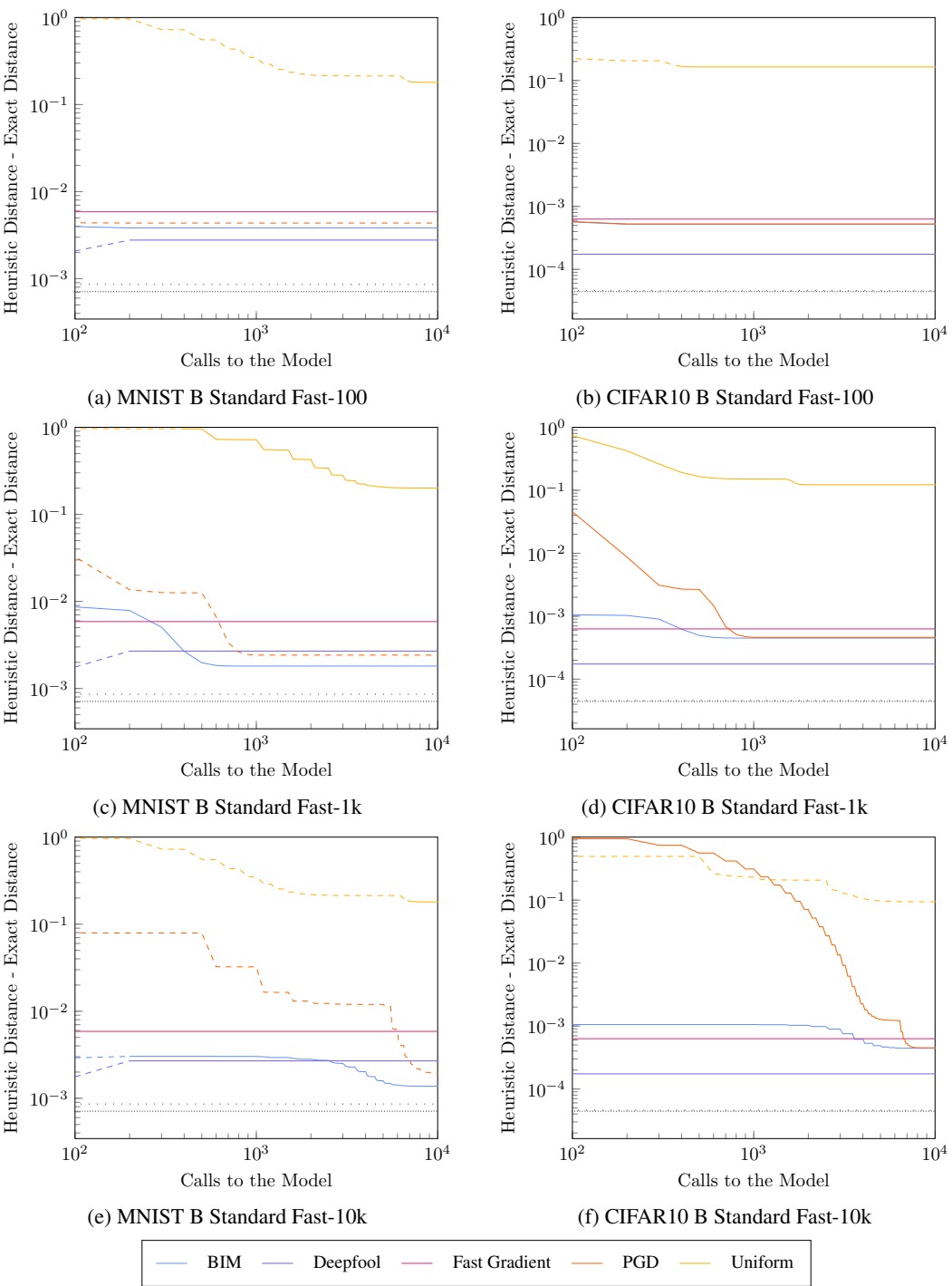

Figure 18: Mean difference between the distance of the closest adversarial examples and the exact decision boundary distance for MNIST & CIFAR10 B Standard. A dashed line means that the attack found adversarial examples (of any distance) for only some inputs, while the absence of a line means that the attack did not find any adversarial examples. The loosely and densely dotted black lines respectively represent the balanced and strong attack pools. Both axes are logarithmic.

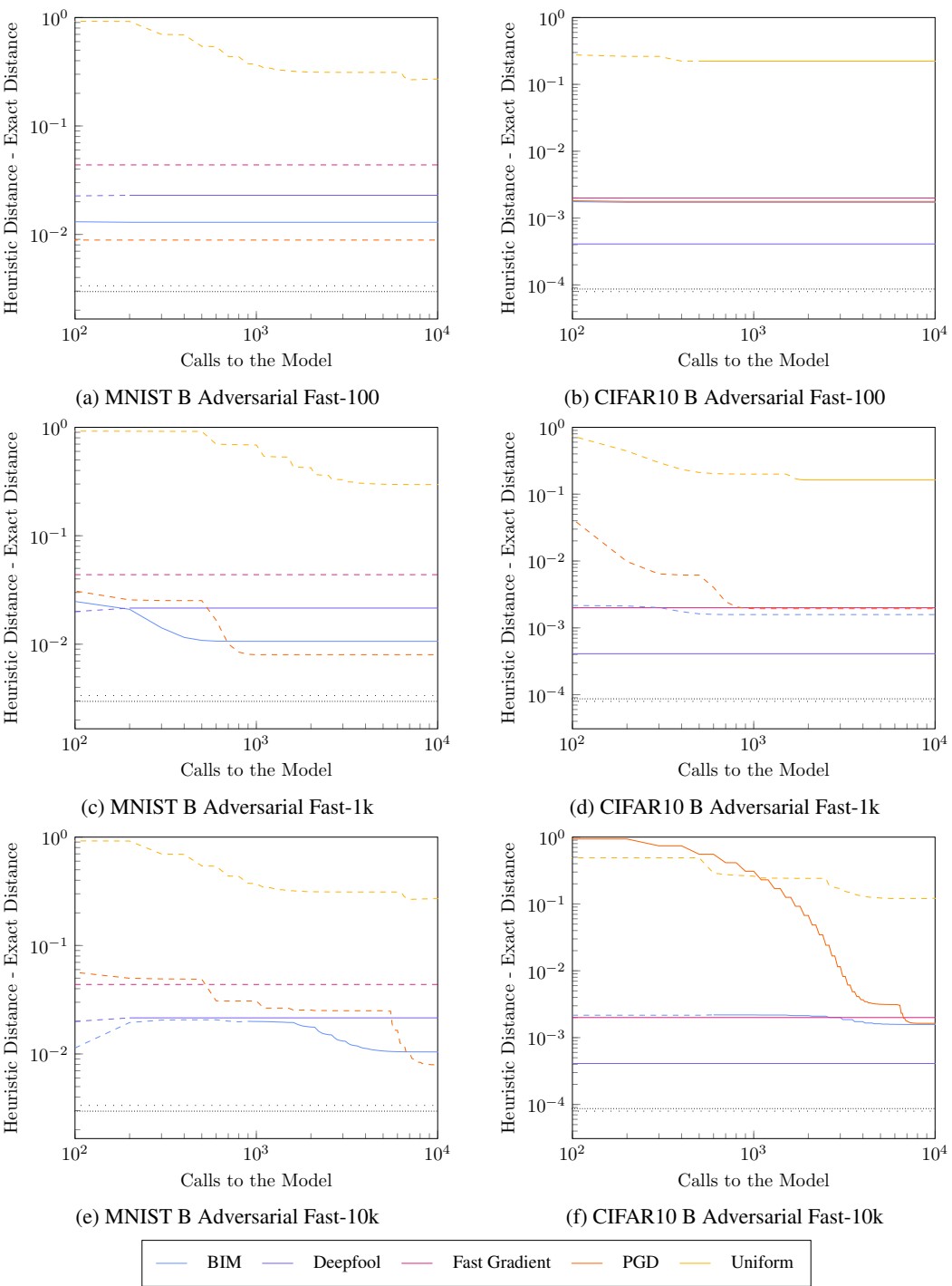

Figure 19: Mean difference between the distance of the closest adversarial examples and the exact decision boundary distance for MNIST & CIFAR10 B Adversarial. A dashed line means that the attack found adversarial examples (of any distance) for only some inputs, while the absence of a line means that the attack did not find any adversarial examples. The loosely and densely dotted black lines respectively represent the balanced and strong attack pools. Both axes are logarithmic.

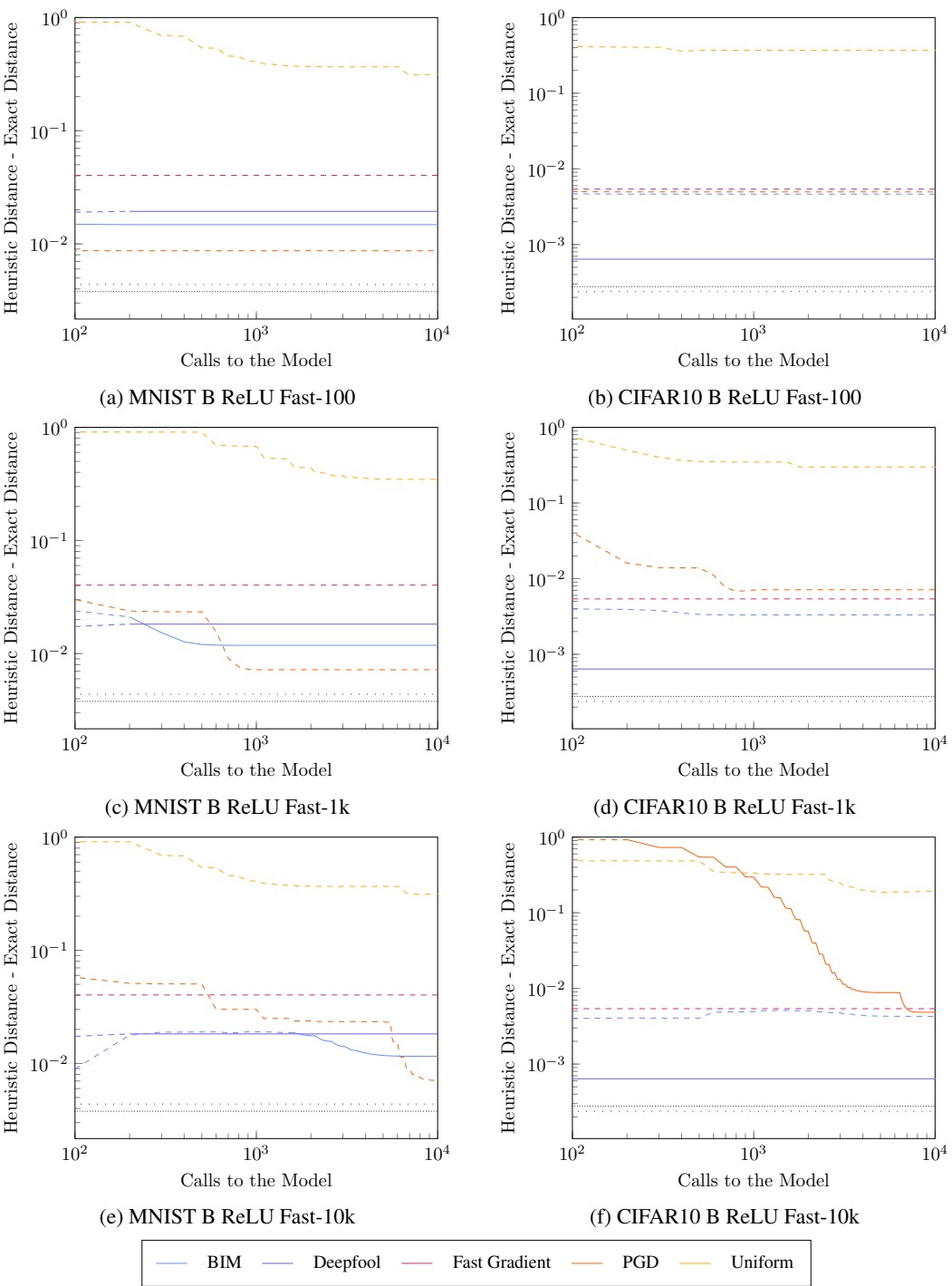

Figure 20: Mean difference between the distance of the closest adversarial examples and the exact decision boundary distance for MNIST & CIFAR10 B ReLU. A dashed line means that the attack found adversarial examples (of any distance) for only some inputs, while the absence of a line means that the attack did not find any adversarial examples. The loosely and densely dotted black lines respectively represent the balanced and strong attack pools. Both axes are logarithmic.

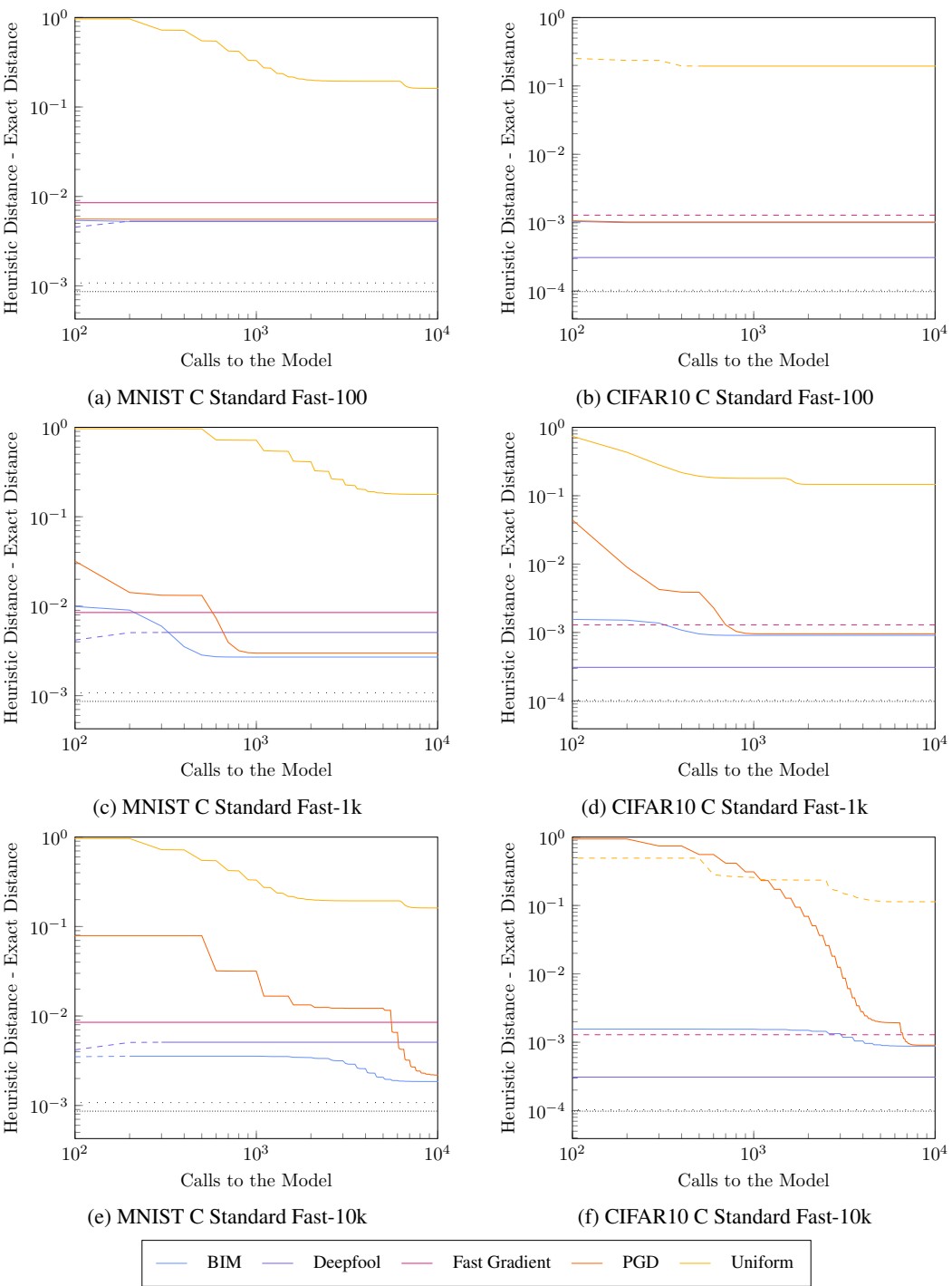

(a) MNIST C Standard Fast-100

(b) CIFAR10 C Standard Fast-100

(c) MNIST C Standard Fast-1k

(d) CIFAR10 C Standard Fast-1k

(e) MNIST C Standard Fast-10k

(f) CIFAR10 C Standard Fast-10k

Figure 21: Mean difference between the distance of the closest adversarial examples and the exact decision boundary distance for MNIST & CIFAR10 C Standard. A dashed line means that the attack found adversarial examples (of any distance) for only some inputs, while the absence of a line means that the attack did not find any adversarial examples. The loosely and densely dotted black lines respectively represent the balanced and strong attack pools. Both axes are logarithmic.

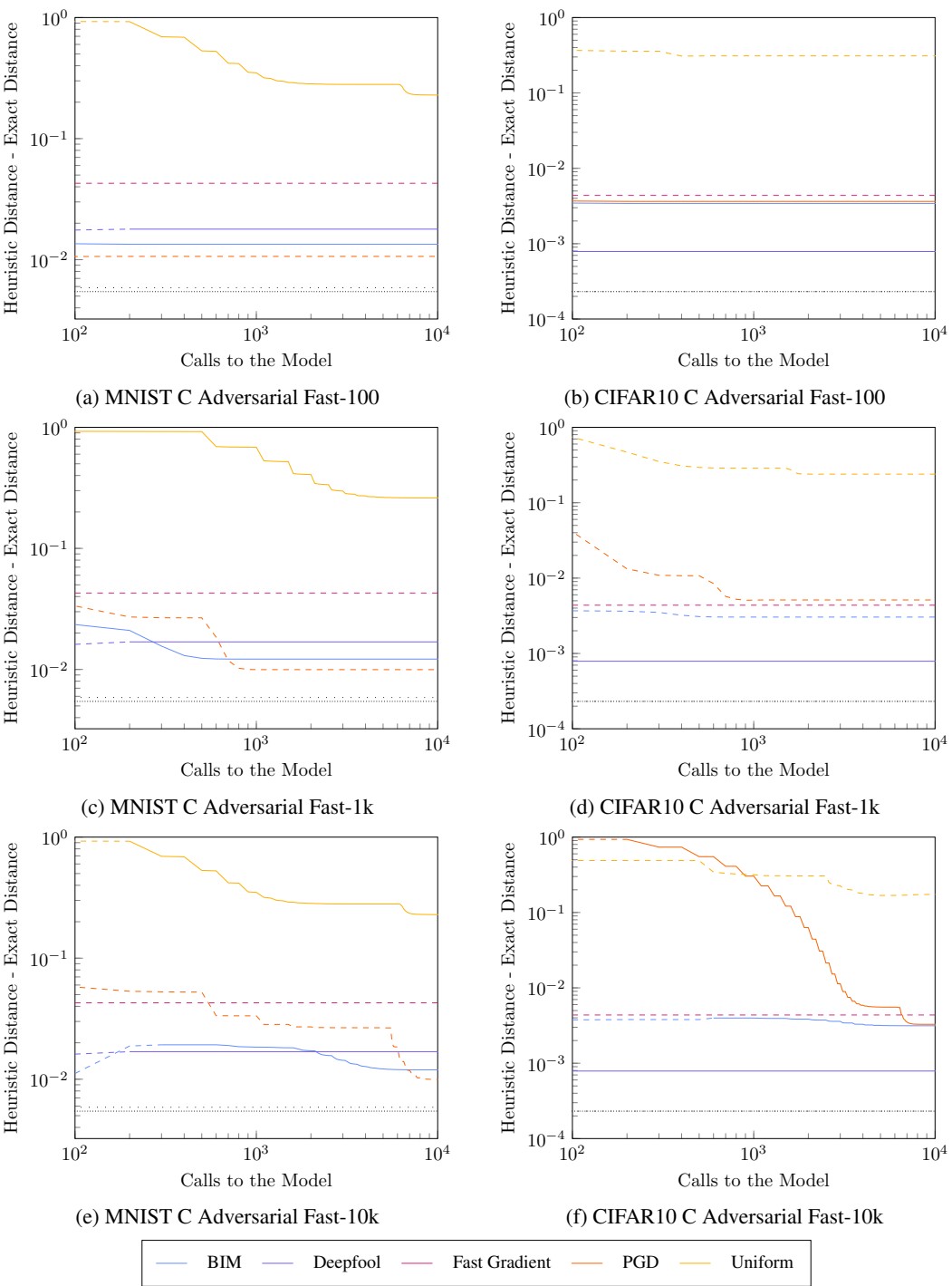

Figure 22: Mean difference between the distance of the closest adversarial examples and the exact decision boundary distance for MNIST & CIFAR10 C Adversarial. A dashed line means that the attack found adversarial examples (of any distance) for only some inputs, while the absence of a line means that the attack did not find any adversarial examples. The loosely and densely dotted black lines respectively represent the balanced and strong attack pools. Both axes are logarithmic.

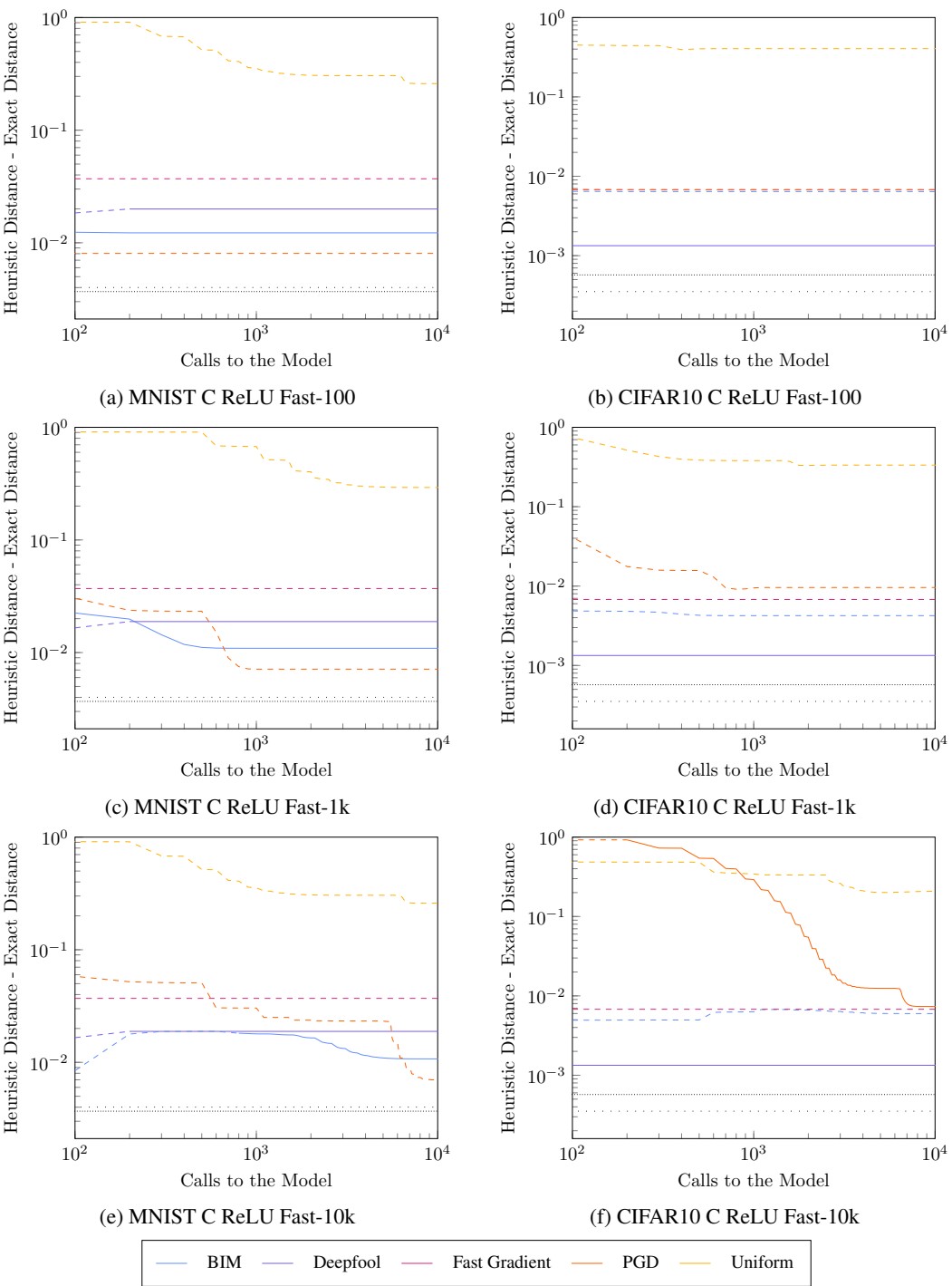

Figure 23: Mean difference between the distance of the closest adversarial examples and the exact decision boundary distance for MNIST & CIFAR10 C ReLU. A dashed line means that the attack found adversarial examples (of any distance) for only some inputs, while the absence of a line means that the attack did not find any adversarial examples. The loosely and densely dotted black lines respectively represent the balanced and strong attack pools. Both axes are logarithmic.

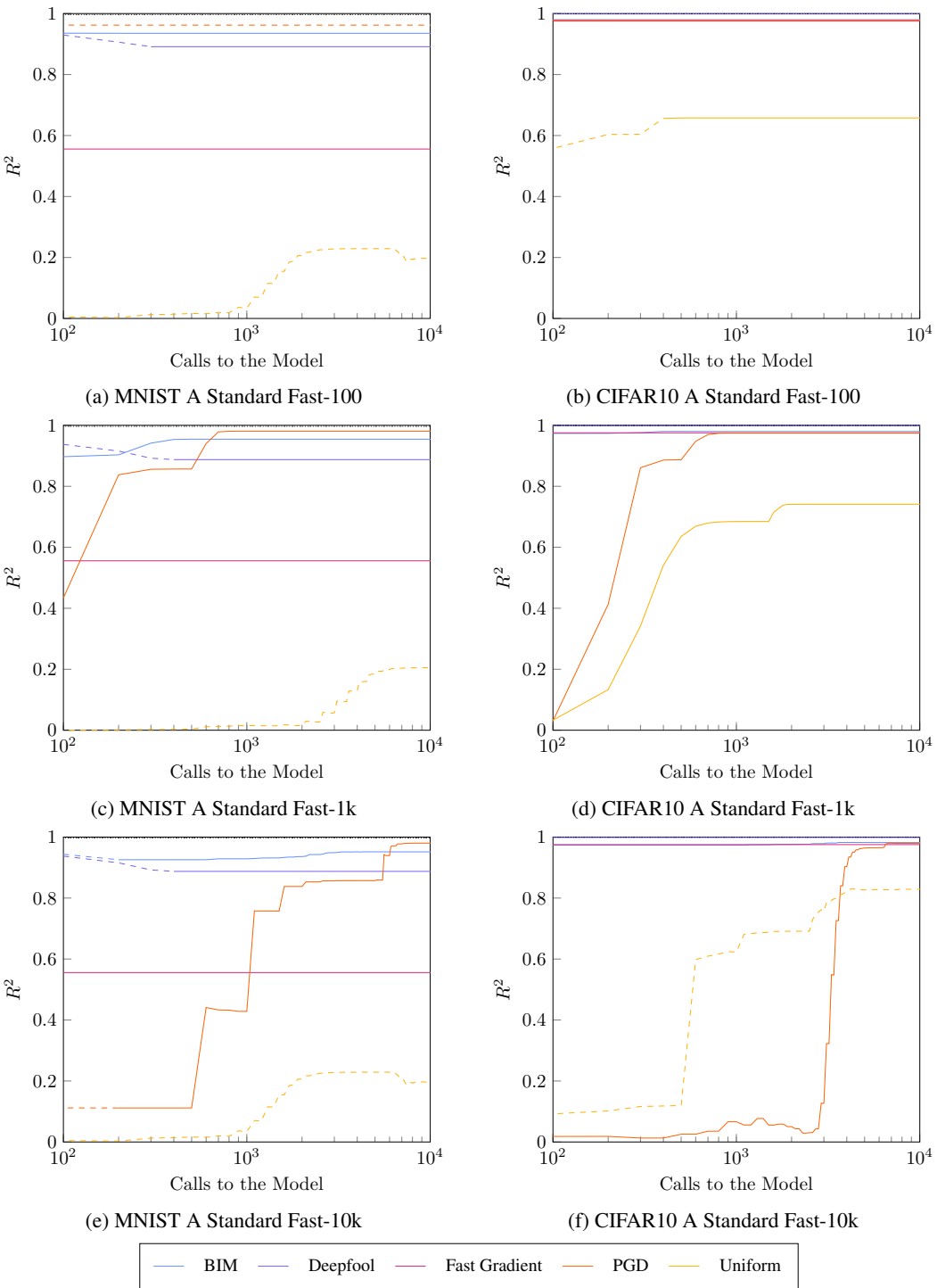

Figure 24: $R^2$ of linear model for the heuristic adversarial distances given the exact decision boundary distances for MNIST & CIFAR10 A Standard. A dashed line means that the attack found adversarial examples (of any distance) for only some inputs, while the absence of a line means that the attack did not find any adversarial examples. The loosely and densely dotted black lines respectively represent the balanced and strong attack pools. The x axis is logarithmic.

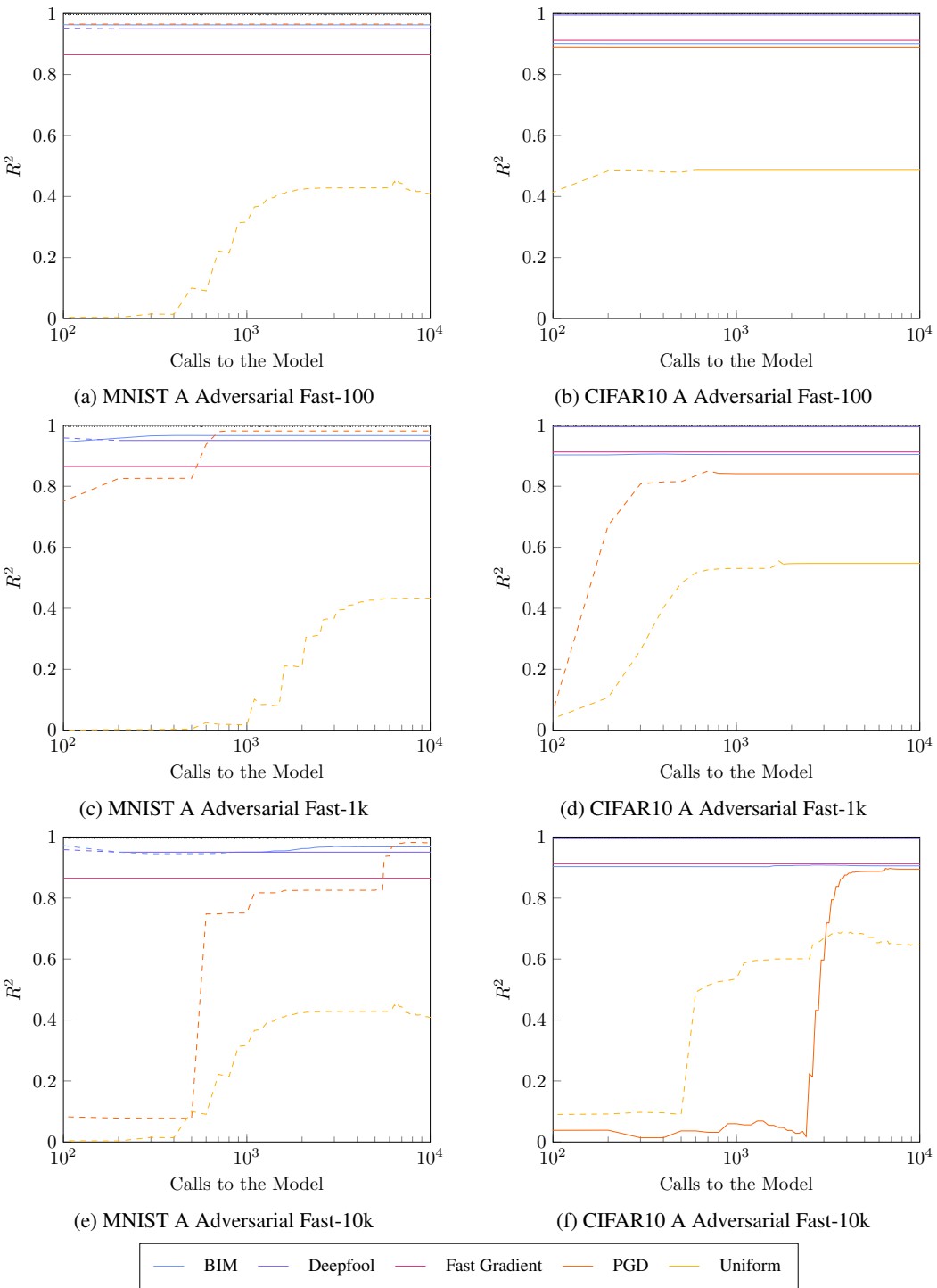

Figure 25: $R^2$ of linear model for the heuristic adversarial distances given the exact decision boundary distances for MNIST & CIFAR10 A Adversarial. A dashed line means that the attack found adversarial examples (of any distance) for only some inputs, while the absence of a line means that the attack did not find any adversarial examples. The loosely and densely dotted black lines respectively represent the balanced and strong attack pools. The x axis is logarithmic.

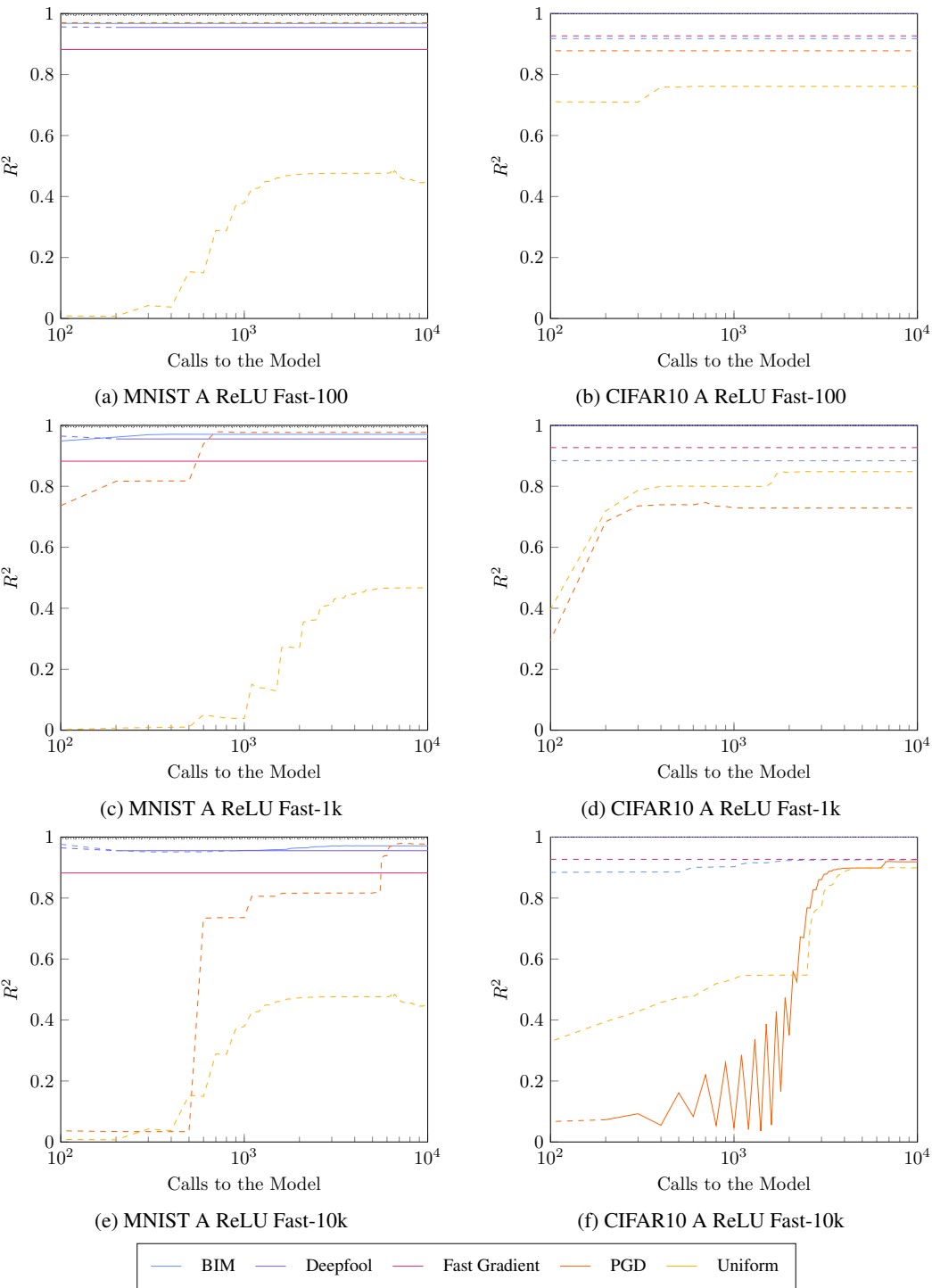

Figure 26: $R^2$ of linear model for the heuristic adversarial distances given the exact decision boundary distances for MNIST & CIFAR10 A ReLU. A dashed line means that the attack found adversarial examples (of any distance) for only some inputs, while the absence of a line means that the attack did not find any adversarial examples. The loosely and densely dotted black lines respectively represent the balanced and strong attack pools. The x axis is logarithmic.

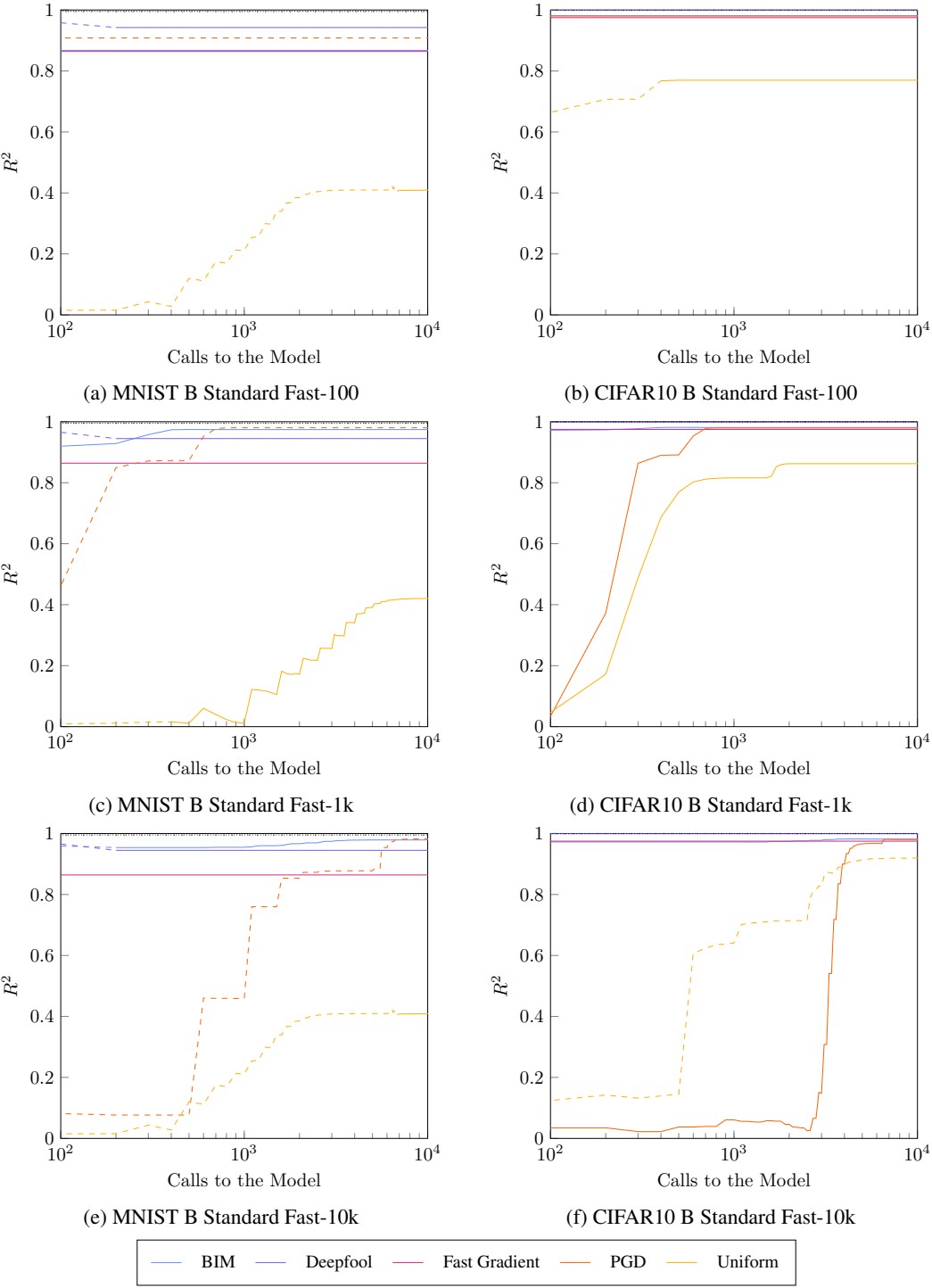

(a) MNIST B Standard Fast-100      (b) CIFAR10 B Standard Fast-100

(c) MNIST B Standard Fast-1k      (d) CIFAR10 B Standard Fast-1k

(e) MNIST B Standard Fast-10k      (f) CIFAR10 B Standard Fast-10k

—— BIM    —— Deepfool    —— Fast Gradient    —— PGD    —— Uniform

Figure 27: $R^2$ of linear model for the heuristic adversarial distances given the exact decision boundary distances for MNIST & CIFAR10 B Standard. A dashed line means that the attack found adversarial examples (of any distance) for only some inputs, while the absence of a line means that the attack did not find any adversarial examples. The loosely and densely dotted black lines respectively represent the balanced and strong attack pools. The x axis is logarithmic.

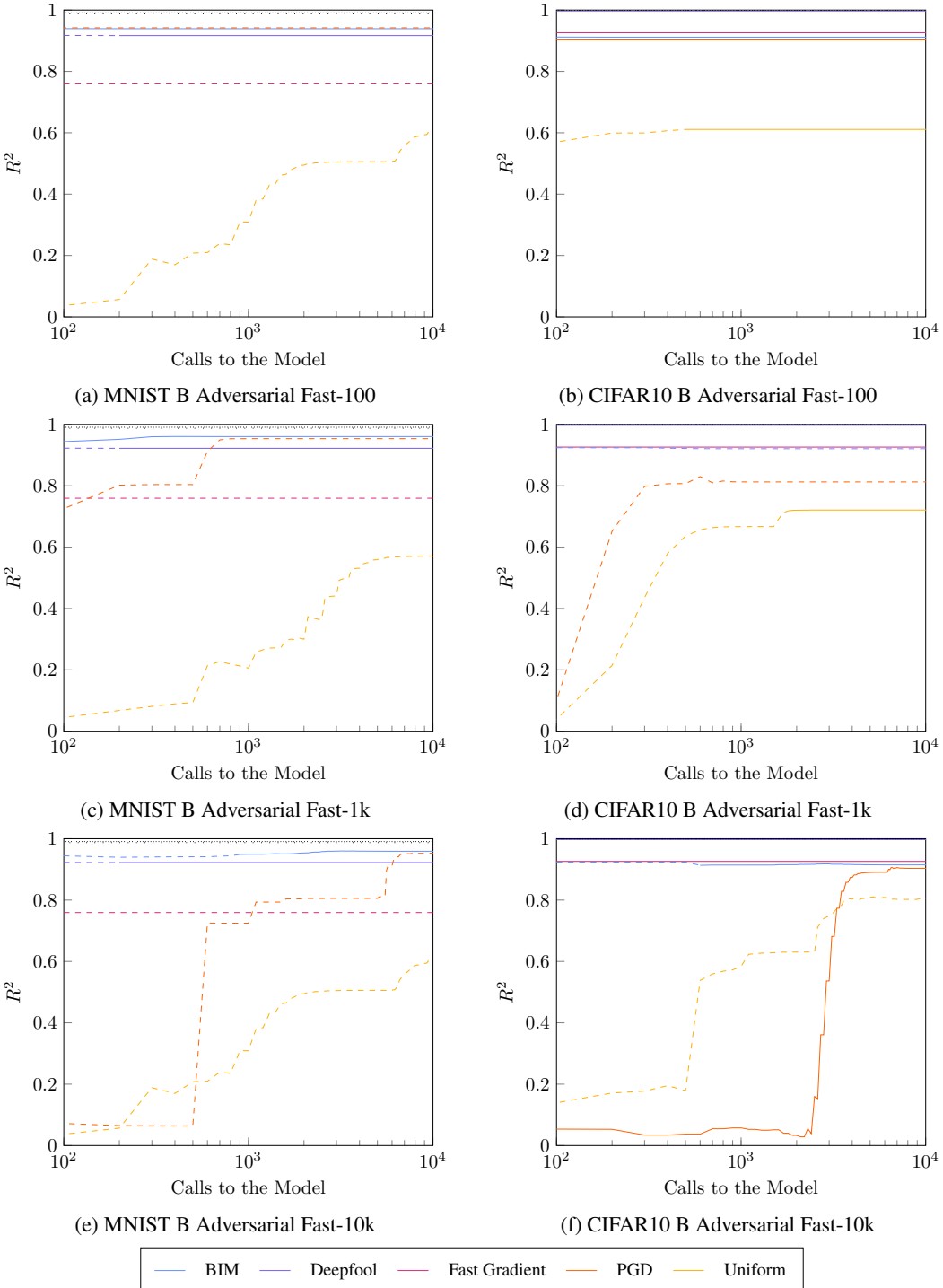

Figure 28: $R^2$ of linear model for the heuristic adversarial distances given the exact decision boundary distances for MNIST & CIFAR10 B Adversarial. A dashed line means that the attack found adversarial examples (of any distance) for only some inputs, while the absence of a line means that the attack did not find any adversarial examples. The loosely and densely dotted black lines respectively represent the balanced and strong attack pools. The x axis is logarithmic.

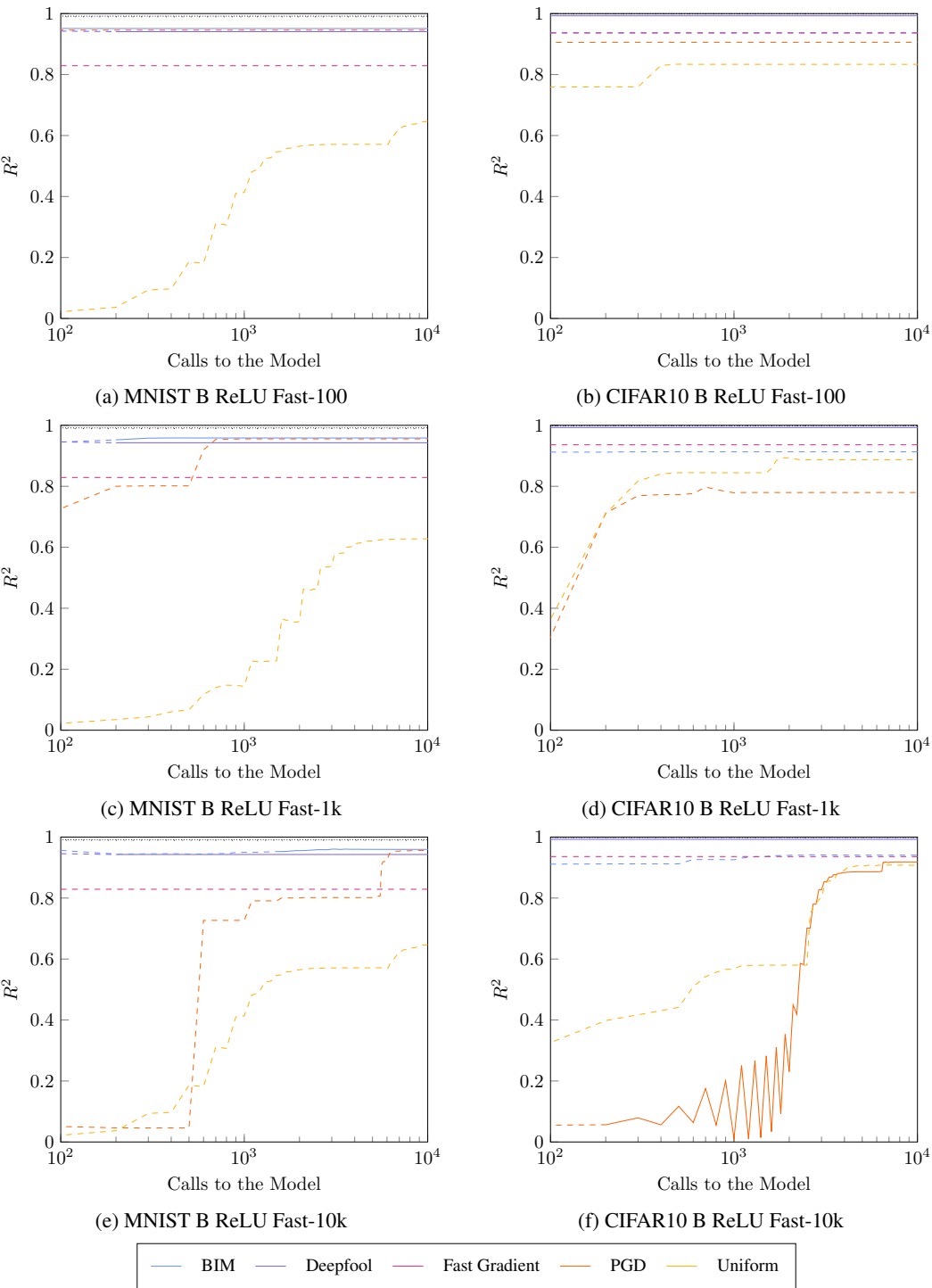

Figure 29: $R^2$ of linear model for the heuristic adversarial distances given the exact decision boundary distances for MNIST & CIFAR10 B ReLU. A dashed line means that the attack found adversarial examples (of any distance) for only some inputs, while the absence of a line means that the attack did not find any adversarial examples. The loosely and densely dotted black lines respectively represent the balanced and strong attack pools. The x axis is logarithmic.

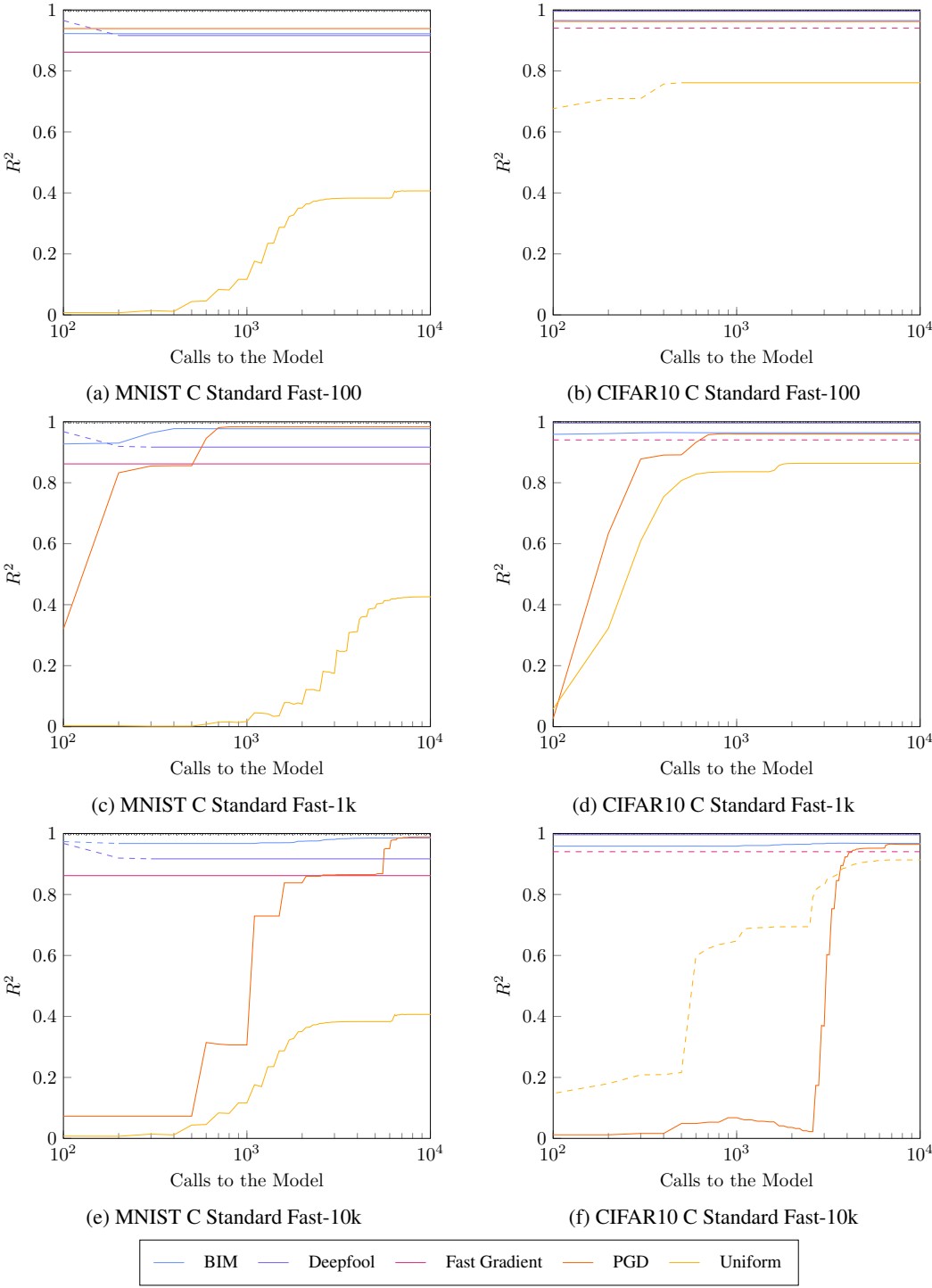

Figure 30: $R^2$ of linear model for the heuristic adversarial distances given the exact decision boundary distances for MNIST & CIFAR10 C Standard. A dashed line means that the attack found adversarial examples (of any distance) for only some inputs, while the absence of a line means that the attack did not find any adversarial examples. The loosely and densely dotted black lines respectively represent the balanced and strong attack pools. The x axis is logarithmic.

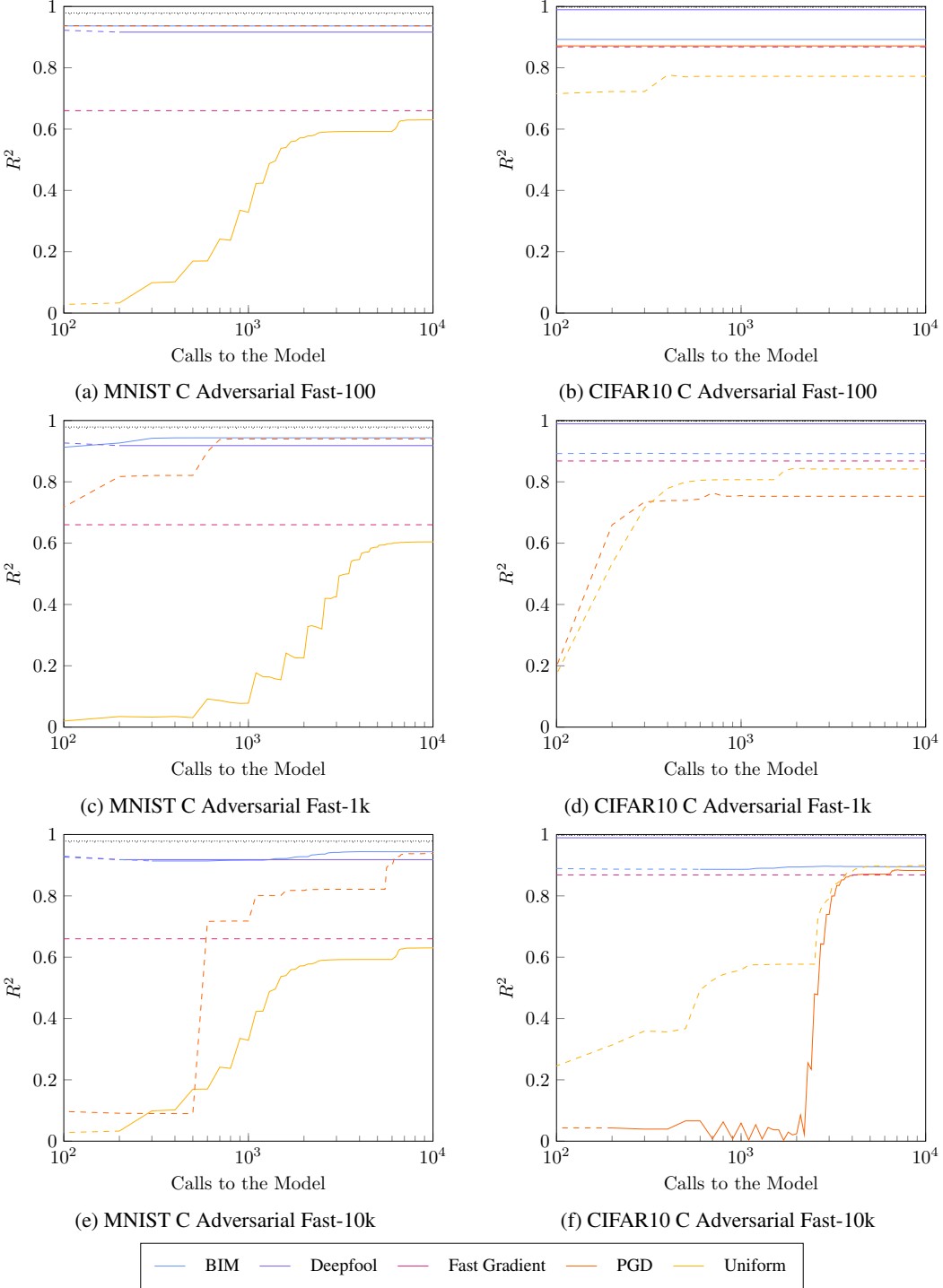

Figure 31: $R^2$ of linear model for the heuristic adversarial distances given the exact decision boundary distances for MNIST & CIFAR10 C Adversarial. A dashed line means that the attack found adversarial examples (of any distance) for only some inputs, while the absence of a line means that the attack did not find any adversarial examples. The loosely and densely dotted black lines respectively represent the balanced and strong attack pools. The x axis is logarithmic.

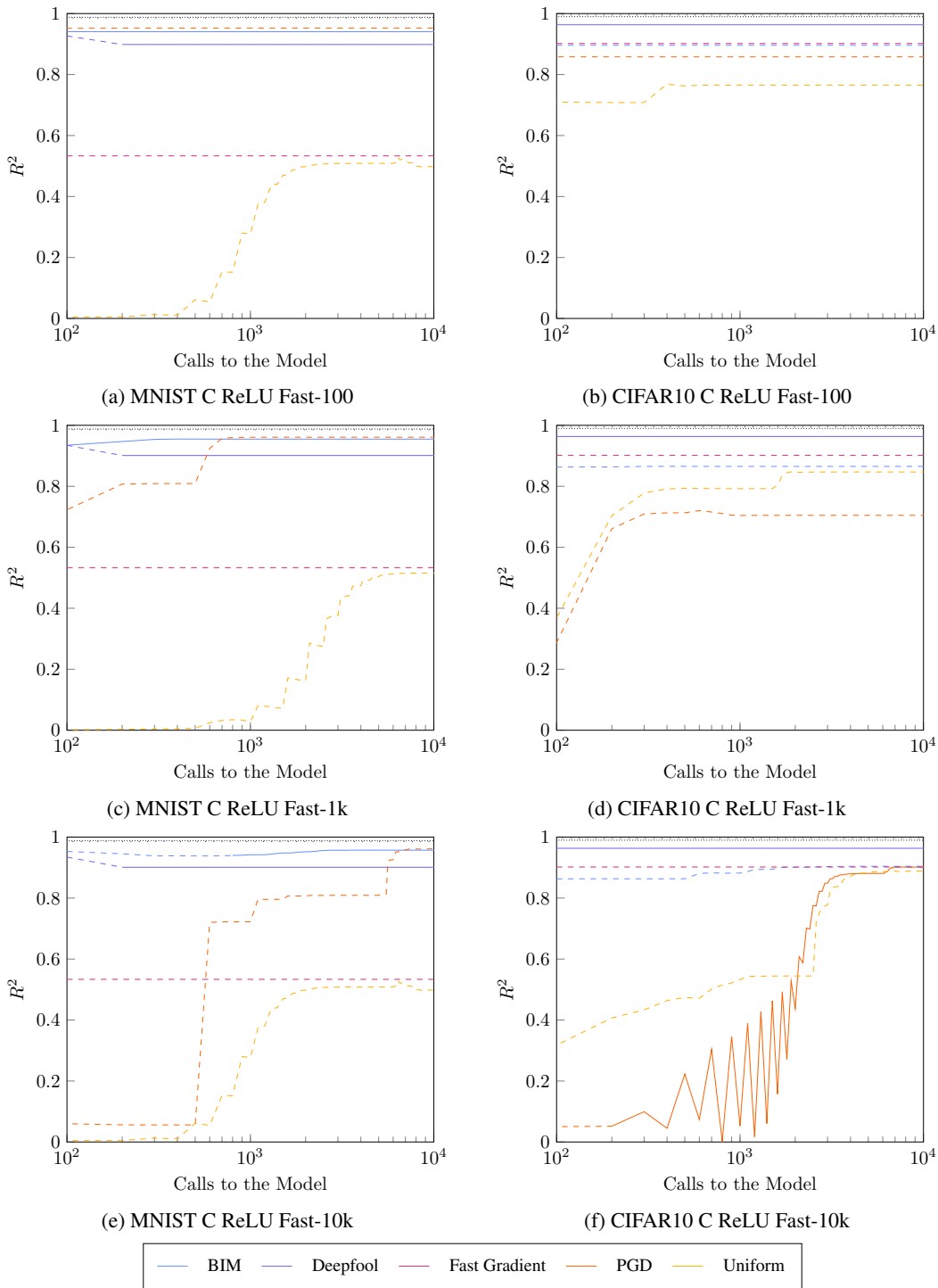

(a) MNIST C ReLU Fast-100

(b) CIFAR10 C ReLU Fast-100

(c) MNIST C ReLU Fast-1k

(d) CIFAR10 C ReLU Fast-1k

(e) MNIST C ReLU Fast-10k

(f) CIFAR10 C ReLU Fast-10k

BIM — Deepfool — Fast Gradient — PGD — Uniform

Figure 32: $R^2$ of linear model for the heuristic adversarial distances given the exact decision boundary distances for MNIST & CIFAR10 C ReLU. A dashed line means that the attack found adversarial examples (of any distance) for only some inputs, while the absence of a line means that the attack did not find any adversarial examples. The loosely and densely dotted black lines respectively represent the balanced and strong attack pools. The x axis is logarithmic.

# M    OVERVIEW OF CERTIFIED DEFENSES

In order to put our defense into context, we provide a slightly more in-depth overview of common approaches to certified robustness, as well as their strengths and weaknesses.

Initially, theoretical work focused on providing robustness bounds based on general properties. For example, Szegedy et al. (2013) computed robustness bounds against $L^2$-bounded perturbations by studying the upper Lipschitz constant of each layer, while Hein & Andriushchenko (2017) achieved similar results for $L^p$-bounded perturbations by focusing on local Lipschitzness. While these studies do not require any modifications to the network or distribution hypotheses, in practice the provided bounds are too loose to be used in practice. For this reason, Weng et al. (2018b) derived stronger bounds through a local Lipschitz constant estimation technique; however, finding this bound is computationally expensive, which is why the authors also provide a heuristic to estimate it.

Similarly, solver-based approaches provide tight bounds but require expensive computations. For example, Reluplex was used to verify networks of at most ∼300 ReLU nodes (Katz et al., 2017). Tjeng et al. (2019) was able to use a MIP-based formulation to significantly speed up verification, although large networks are still not feasible to verify. Solver-friendly training techniques can boost the performance of verifiers (such as in (Xiao et al., 2019)); however, this increase in speed often comes at the cost of accuracy (see Section 6).

Another solution to the trade-off between speed and bound tightness is to focus on specific (and more tractable) threat models. For example, Han et al. (2021) provide robustness guarantees against adversarial patches (Brown et al., 2017), while Jia et al. (2019) focus on adversarial word substitutions. In the same vein, Raghunathan et al. (2018) provide robustness bounds for specific architectures (i.e. 1-layer and 2-layer neural networks), while Zhang et al. (2021) introduce custom neurons that, if used in place of regular neurons, provide $L^\infty$ robustness guarantees. These techniques thus trade generality for speed.

The most common approach, however, consists in providing statistical guarantees. For example, Sinha et al. (2018) showed that using a custom loss can bound the adversarial risk. Similarly, Dan et al. (2020) proved adversarial risk bounds for Gaussian mixture models depending on the "adversarial Signal-to-Noise Ratio". Finally, Cohen et al. (2019) introduced a smoothing-based certified defense that, due to its high computational cost, is replaced by a Monte Carlo estimate with a given probability of being robust. This work was later expanded upon in (Salman et al., 2020) and (Carlini et al., 2022). The main drawback of these techniques is the fact that they cannot be used in contexts where statistical guarantees are not sufficient, such as safety-critical applications.

All of these certified defenses prioritize certain aspects (speed, strength, generality) over others. In the context of this (simplified) framework, CA in its exact form can be thus considered a defense that prioritizes strength and generality over speed, similarly to Katz et al. (2017) and Tjeng et al. (2019).

