# OpenReview forum: "Countering the Attack-Defense Complexity Gap for Robust Classifiers"
_ICLR.cc/2023/Conference — Submitted to ICLR 2023_

### Official Review · Reviewer_wxLx · 2022-10-25

**Confidence:** 3
**Correctness:** 4
**Technical Novelty And Significance:** 2
**Empirical Novelty And Significance:** 2
**Recommendation:** 6

**Clarity, Quality, Novelty And Reproducibility:**

The paper provides an interesting approach to bridging the attack-defense gap for robust classifiers. Some of the ideas and empirical results in the paper are new and interesting. The paper is also very clearly written and provides detailed proof of the theorems listed. The authors also provide all relevant details to reproduce the results. They also created a dataset UG100, recording some of the results for optimal bounds provided in the paper.

**Strength And Weaknesses:**

Strengths
- The complexity results for attacking polynomial models and training robust polynomial models provide a reasonable explanation for the hardness of doing these tasks on practical datasets.
- The empirical results showing that the attack radius achieved by a suit of weaker attack strongly correlates with the actual radius of the decision boundary around an example is very interesting and provides a good justification for the CA approach.

Weakness
- The underlying ideas presented in the paper are not very novel. Similar hardness results for ReLU networks do exist in previous papers. The flagging of specific examples provides a heuristic guarantee in contrast to the solid theoretical guarantees provided by other certification methods. The use case of Counter-Attack is not very clear to me.
-  Due to the computational limitations of the MIP solvers, the empirical justification for the method is only shown on small neural networks. It is not clear whether these results extend to general neural networks with more complicated architectures.

**Summary Of The Paper:**

To improve the attack-defense gap for robust classifiers, the paper suggests using a sample-specific attack-based flag, Counter-Attack(CA),  that tells the user if there are adversarial samples within a $\epsilon$-ball of the given sample. In the case of a perfect attack, the authors show that Counter-Attack also provides a computational robustness guarantee that it is hard to find unflagged adversarial examples beyond the threat models $\epsilon$ radius.

However, the paper also formally proves that, under broad assumptions, attacking a polynomial-time classifier is $NP$-complete, while training a polynomial-time model that is robust on even a single input is $\Sigma^P_2$-complete. They also extend these results to the case of general non-polynomial time classifiers.

The authors suggest using a heuristic attack to flag examples for practical purposes. To improve the validity of this approach, the authors recommend using a higher radius for the heuristic attacks to make up for their relative weakness to a perfect attack. To this end, the authors study the consistency between the average distance of an adversarial example found by the heuristic attacks vs. the perfect attack or MNISt and CIFAR10 datasets. The empirical results show that for a well-chosen suit of attacks, the gap is pretty consistent in the evaluated examples.

**Summary Of The Review:**

The idea of using the success of a suit of heuristic attacks to flag vulnerable examples is interesting, but it is not clear if there is a specific use case where it would help. Although the paper establishes some interesting theoretical results for the power of this method when used with a perfect attack, none of the guarantees extend to the setting of heuristic attacks. The empirical evidence provided to show the effectiveness is only limited to a very small set of architectures for shallow neural networks. It is not clear if these observations hold in general. As for the setting of shallow neural networks, some of the formal verification methods already provide good bounds.

I feel the paper provides some interesting results that might serve as an interesting starting point for further investigation. However, it does not provide enough justification to use the proposed Counter-Attack method.

---

> ### Author Response · Authors · 2022-11-12
> **Response to Reviewer wxLx**
>
> Dear reviewer wxLx,
>
> we thank you for your feedback. Your summary captured correctly all the key technical steps of our analysis, and your comment made us realize that our initial, technically focused, presentation was not effective at conveying some broader-scale elements of our contributions. We made an effort to clarify those in the general response comment (“General Comments for All Reviewers”) and we will of course do our best to outline them properly in the updated paper.
>
> **Complexity results**: We agree that, in the context of our paper, the generalization of existing NP-hardness results to new classes represents a minor contribution. However, it lays the theoretical groundwork for the novel  $\Sigma_2^P$ completeness result, which is one of the major contributions of the paper and one we believe might have far-reaching consequences. For instance, Theorem 3 and its corollaries represent a hard limit on what can be achieved with certified defenses: the existence of a poly-time technique to find robust parameters for a ReLU network would imply P=NP, while the existence of an NP-time technique to do the same would imply the truthfulness of the Polynomial Hierarchy Collapse conjecture (which is believed to be false).
>
> **Heuristic vs solid theoretical guarantees**: When used with an exact attack, CA provides solid theoretical guarantees. Using heuristic attacks is simply a way to reduce computational complexity at the cost of approximate robustness. Using an attack with a guaranteed approximation factor, or any other technique capable of providing bounds, would be another interesting alternative (and a topic for future research).
>
> **Use cases for CA**: There is now a growing body of literature (see the original review by 2Ucv) on abstaining from providing an answer, and CA could be used for the many of the same applications as such approaches. Those include human-in-the-loop schemes, safety critical contexts, decision support systems, content moderation; intuitively: every scenario where the goal is supporting a human decision maker rather than replacing them.
>
> **Small networks**: Using smaller-scale networks was basically a forced choice for us, since using techniques that focus on providing bounds was not sufficient for our evaluation. We explain the motivation in detail in the general response comment.
>
>
> **Conclusion**: We hope that our clarifications provided more insight regarding the strengths and limitations of our work. We will post in a few days a revised version, which reflects the points made in the responses and emphasizes the implications of our theoretical and empirical results. If you have any additional doubts regarding our responses, by all means do not hesitate to ask us. We hope that, in light of our contributions to the field’s understanding of adversarial defenses, you will find our work worth supporting in the updated version of the review. Thank you.
>
> Authors3153

---

### Official Review · Reviewer_mBWL · 2022-10-25

**Confidence:** 2
**Correctness:** 3
**Technical Novelty And Significance:** 3
**Empirical Novelty And Significance:** Not applicable
**Recommendation:** 6

**Clarity, Quality, Novelty And Reproducibility:**

In general, I think this paper is well-written and organized. The motivation and rationale of the proposed method are introduced clearly.

**Strength And Weaknesses:**

Strength:
The authors provided a thorough theoretical analysis of the complexity of neural network adversarial robustness. The proposed counter-attack method is very simple but is able to take advantage of existing defenses and provide robustness guarantees. The authors also showed detailed information about their experiments.
Weakness:
1. In the main text, it would be great if the authors can present a table for comparison between CA and existing defense methods on their certified robustness accuracy, as well as adversarial accuracy under different attacks.
2. Please refrain from only using color to distinguish curves in Figure 2, as it may not be friendly to readers with color blindness.

**Summary Of The Paper:**

This paper proves that attacking a polynomial-time classifier is NP-complete, and training a polynomial-time model that is robust on a
single input is Σ_2^P-complete. The authors proposed a method that evaluates on the fly if a model is robust on a specific point by running
an adversarial attack on the input. Based on this, the authors proposed a dataset of provably optimal adversarial examples found on
images of MNIST and CIFAR10 for 6 small-scale Neural Networks.

**Summary Of The Review:**

In general, I think the proposed method is well-motivated and the studies are solid. So I recommend a weak accept.

---

> ### Author Response · Authors · 2022-11-12
> **Response to Reviewer mBWL**
>
> Dear reviewer mBWL,
>
> we thank you for your feedback, both for the positive comments (in particular for realizing the benefits of using improvements in adversarial attacks to obtain better defenses) and those about limitations (which made us understand that some key points of our paper could be explained more clearly). Responses to some broader concerns expressed by multiple reviewers can be found in the general post (“General Comments for All Reviewers”). Here we will focus on your more specific comments.
>
> **Comparison between CA and existing defenses**: A comparison with other defenses would be certainly valuable, and for this reason we are adding a qualitative overview of the most common defense approaches and how they relate to CA. However, we really want to stress that the focus of our paper is different, since most of its content focuses on theoretical (but practically relevant) results.
>
> In particular, our goal is not mainly to present a better defense approach. Instead, we aim to:
> 1. Show that there is a structural problem that, under worst cases conditions, makes robust training harder than attacks. This is corroborated by an extensive body of empirical evidence available in the literature;
> 2. Point out that the issue we identified can be circumvented by tackling defense from a different perspective;
> 3. Make a sound investigation of how empirical adversarial attacks could replace exact attacks in order to reduce computational complexity at the cost of approximate robustness.
>
> Our main hope would be to orient future investigations and allow researchers to make their efforts more efficient, similarly to what worst-case complexity proofs have done for other fields.
>
> In hindsight, we realize we could have emphasized this key point better. Thankfully, this is something that can be done without major modifications to the paper structure.
>
> **Figures**: we will fix the figures in the main content in the revised version of the paper (which we will post in the next few days). Improving the plots in the supplementary content will take more time, as those are more content-dense.
>
> **Conclusion**: Overall, we thank you again for your review and we are glad that you found our paper solid and well-motivated. While we acknowledge that our work (as any other) has limitations, we hope we clarified what its main strengths are and how it could contribute not just one more defense approach, but insights that are useful for research on adversarial attacks in general. We thus hope that you will support our acceptance in the updated version of the review. Thank you.
>
> Authors3153

---

### Official Review · Reviewer_2UCv · 2022-10-27

**Confidence:** 5
**Correctness:** 3
**Technical Novelty And Significance:** 2
**Empirical Novelty And Significance:** 2
**Recommendation:** 5

**Clarity, Quality, Novelty And Reproducibility:**

The paper is written quite clearly. The approach is novel, but also flawed in some sense.


**Strength And Weaknesses:**

Strength: aiming for a computational understanding of the adversarial example phenomenon and explaining why defending is hard.

Mild weakness: the NP completeness results, as formulated, are rather trivial. For example, if we allow the model to compute SAT instances, is is clear that finding a "close point that is misclassified" could be NP complete, because one can choose the model and the input in such a way that doing so is the same as solving an arbitrary SAT instance.

Weakness: there are several severe limitations on how much one can interpret the results of this paper in relation with practice. The experimental part is extensive, but its guarantee is only heuristic while there is a line of work on provably certifying robustness of decisions.



**Summary Of The Paper:**

The paper aims to study the “computational complexity” of attacking versus that of defending machine learning models against adversarial attacks, which are test-time attacks that find so-called adversarial examples.

An adversarial example for an input x is a bounded perturbation of x that is misclassified.

The paper studies the worst-case computational complexity of finding adversarial examples (for a given perturbation threshold) vs. training a robust model. It is shown that in the *worst case* (even for basic ReLU models):
Finding adversarial examples is NP complete, while
Finding robust models is Sigma-2 complete. Sigma-2 is a bigger than (or equal to) complexity class than NP (and is conjectured to be strictly bigger).

The paper then defines a way of perhaps defending adversarial examples by trying to test robustness of the points one by one, and abstaining from answering when there is a non-robustness evidence. In particular, the paper applies a class S of attacks on each point, using a specific perturbation eps. Then, the thesis is that if we don’t find an example with eps perturbations, then perhaps it is relatively safe to assume that there is no such example for smaller eps’ < eps. The paper verifies this experimentally for small models for which the exact robustness can be computed.

Evaluation: I think the formulation of the problem this way has serious limitations that could prevent us from using these results for any real world insights. So I cannot support acceptance. See my comments below.

Considering NP completeness to reflect “easier task” than being Sigma-2 complete is both correct (theoretically) and wrong (practically). Theoretically, Sigma-2 is a “harder class” as it contains NP, but when it comes to running algorithms efficiently, there is no difference between them, as they both contain problems that we don’t know how to solve in polynomial time. So, there is a fundamental difference between the “computational approach” of this paper and that of e.g., Garg et al (cited work) where the distinction there is drawn between poly-time vs. non-poly-time algorithms. This makes the theoretical insight of this paper quite irrelevant to practice.

The paper does not pay attention that their notion of adversarial examples might *not* lead to misclassified inputs. For example, suppose we perturb x into x’ and the model changes the label for x’ while the true label is the changed label. (e.g., changing a cat image to a dog image while the model predicts the dog as dog). This should not constitute an adversarial example. E.g., see the work of “revisiting adversarial examples” by Suggala et al. Without this subtle point being fixed, it is hard to interpret the results of the paper as attacks. This might seem like a minor thing, but it becomes clear that it is a major issue when one notices that hardness of robust or non-robust learning of a task crucially depends on *what class of concept functions* we want to learn. E.g., one can show that simple function classes *can* be learned robustly and the literature is full of such results. This shows another flaw with the formulation of the problem in this paper.

The computational complexity of adversarially robust learning is an *average-case* phenomenon. Namely, we would like to know, what is adversary’s chance of making x misclassified, when we (say i.i.d) sample the training set, then train (perhaps in a randomized way) a model, and then pick x at random, and finally let adversary perturb x. So, even though the perturbation itself is studied in the worst-case, the whole problem is average-case. As an example, breaking crypto-systems is an average case task, even though one can find specific ciphertex whose complexity *in the worst case* is NP complete, yet finding NP complete encryption schemes is a major open question in cryptography.

By now, there is a rich literature on “certifying” robustness of decisions per instance. The paper seems to invent the wheel here and give a *weaker* certification than what those works offer, in terms of its guarantee. While other works on certification are provably right, here we only get a heuristic claim. This looks like a step back.

In addition, there is also a (by now rich) line of work on “abstaining” from answering queries for the sake of robustness. This paper’s approach is different, but still quite relevant, to that line of work. So it is good to do a discussion and compare. (As I understand those works try to abstain from answering the perturbed inputs, while this paper’s approach is to not answer even when x is not perturbed, but is just close to a dangerous perturbation). My understanding is that this approach hurts the correctness/accuracy much more, as it might abstain from answering normal instances as well.


**Summary Of The Review:**

Interesting direction for studying the complexity of adversarial examples vs defending against test time evasion attacks. The weakness is that the formulation of the problem does not give an actual insight on how much is the running time of those tasks in practice, as the formulation is in the worst case and there is no major difference between NP completeness and sigma-2 completeness in practice; they are both not feasible. See more details in the detailed review.

---

> ### Author Response · Authors · 2022-11-12
> **Response to Reviewer 2Ucv (1/2)**
>
> Dear reviewer 2Ucv,
>
> we thank you for your detailed analysis of our work. While we acknowledge some of the technical limitations that you highlight (which we will clarify in the revised version of the paper), we stress that our contributions are still significant, as they have the potential to better direct future efforts from the research community.
>
> In particular, our key contribution is providing a possible motivation for the empirical difficulties routinely encountered with conventional defense approaches, and pointing out an alternative research direction. We address this point in the general response comment (“General Comments for All Reviewers”), together with two broad concerns that many reviewers had.
>
> Here we provide detailed responses to your more specific comments.
>
> **NP vs $\Sigma_2^P$**: Non-poly-time problems are not all alike. Even within the same complexity class there can be huge differences, for example Set Covering Problems are much easier to solve compared to e.g. Resource Constrained Scheduling Problems, despite both being NP-complete (in their decision form). In the constrained optimization literature, the fact that two problems belong (in the worst case) to distinct classes is commonly accepted as a strong indication that one may be practically harder to solve than the other. For example, $\Sigma_2$ SAT, which is in $\Sigma_2^P$, is known to be empirically much more difficult to solve than classical SAT (as it is the case for most versions of quantified SAT). In our case, there is an extensive body of literature (e.g. [1, 2, 3, 4]) that shows how fooling defense approaches tends to be comparatively easy. Such an empirical asymmetry between attacks and defenses was one of the main motivations for conducting our investigation.
>
> **Definition of adversarial example**: In the adversarial attack literature, the problem of maintaining semantic consistency between the original and the perturbed sample is captured by using bounds on a distance metric. In principle, as you point out, such distance metric should be a reasonable approximation of human perception and interpretation: in practice, crafting such a metric is a very challenging (and very open) problem. Almost every work in the literature chooses to rely instead on simple distances (norms) and very small bounds: intuitively, changing a few pixels does not really change an image for a human observer. This simplification does not prevent such works from achieving important results, from both a theoretical and practical standpoint (see [5, 6, 7, 8]). Therefore, when doing our analysis, it made sense to choose a setting that captures the assumptions behind most current research. This is doubly true since our main goal is making sense of observed empirical trends in the literature and point out an alternative.
>
> **“Simple function classes can be learned robustly”**: This is true, and in fact it is mentioned in the paper (in Section 2 w.r.t a specific reference and in Section 4 when commenting about the worst-case nature of our result). There are, however, two important facts that should be kept in mind: 1) as mentioned in the previous comment, there is extensive empirical evidence that defending tends to be much harder than attacking; while this is not a proof, it hints at the fact that simple function classes might not be so common in practice. 2) As we discuss in the general response comment (“General Comments for All Reviewers”), research on tractable classes usually receives a strong motivation once a problem is proved to be intractable in the general case (this was the case for most well-known problems in combinatorial optimization, e.g. scheduling, knapsack, and routing).
>
> **Average vs worst-case complexity**: We agree that proving average case complexity would provide even greater insights and motivation. However, this fact does not imply that a worst-case complexity proof is not important. It should also be kept in mind that generic (ReLU) classifiers, as opposed to specific sub-cases such as those in [9], can be extremely complex mathematical objects; as such, stronger results can be reached only with long-term research. The field of adversarial attack research is still far from being able to tackle average complexity of general models and distributions. For example, the (very interesting) analysis by Garg et al. [10] deals with average-case complexity, but requires examples to be cryptographically signed (i.e. a very strong assumption on the input distribution and the system behavior).

---

> ### Author Response · Authors · 2022-11-12
> **Response to Reviewer 2Ucv (2/2)**
>
> **Heuristic vs exact guarantees**: When used with an exact attack, CA provides theoretically sound guarantees in the general case. This is in fact an improvement w.r.t. several approaches in the literature that, while being more scalable or able to provide statistically grounded guarantees, need to make restrictive assumptions on the architecture or on the ground truth distribution.
>
> Using heuristic attacks (as we suggest in Section 5.2) is simply one way to reduce computational complexity at the cost of approximate robustness. There is an interesting interaction here between algorithms capable of providing guaranteed approximations (or bounds) and CA, as they could be used as an alternative to heuristic attacks to improve scalability. This is also hinted to by a comment from reviewer mBWL, and it provides means for defenses to benefit from future research on adversarial attacks.
>
> No matter what the approach is, any defense that has attempted to be scalable has had to sacrifice something in terms of generality or level of guarantees. This is one more hint to the practical relevance of our complexity theorems: in particular Theorem 3 shows that, unless the polynomial complexity hierarchy collapses, a defense cannot at the same time be efficient and provide tight, non-probabilistic, bounds.
>
> **Abstaining from answering**: There might be multiple rationales for abstaining from answering; for example, some approaches in the mentioned field abstain from answering based on model confidence: this approach does not account explicitly for adversarial attacks and does not protect adequately towards them. Other approaches focus instead on the identification of dubious input, and those have been used as defenses: unfortunately it is not easy to distinguish perturbed from unperturbed inputs. Defense approaches that tried such a strategy have been fooled and proven unreliable in the past [11].
>
>
> **Conclusion**: Overall, your points have helped us to identify the key aspects of our work that needed clarification. We will post a revision where we will address those. If you have other doubts regarding our work, feel free to comment on OpenReview and we will do our best to respond in a timely manner.
>
> Your initial review agreed with our technical analysis, but expressed some doubts regarding the link between our results and the larger implications for the field (see the general comment). We hope that, if such doubts have been cleared, you will agree with reviewers mBWL and wxLx and provide your support for the updated version of our paper. Thank you.
>
>
> Authors3153
>
>
>
>
> [1] Nicholas Carlini and David Wagner. Defensive distillation is not robust to adversarial examples. arXiv preprint arXiv:1607.04311, 2016.
>
> [2] Warren He, James Wei, Xinyun Chen, Nicholas Carlini, and Dawn Song. Adversarial example defenses: ensembles of weak defenses are not strong. In Proceedings of the 11th USENIX Conference on Offensive Technologies, pp. 15–15, 2017.
>
> [3] Florian Tramer, Nicholas Carlini, Wieland Brendel, and Aleksander Madry. On adaptive attacks to adversarial example defenses. Advances in Neural Information Processing Systems, 33:1633–1645, 2020.
>
> [4] Francesco Croce, Sven Gowal, Thomas Brunner, Evan Shelhamer, Matthias Hein, and Taylan Cemgil. Evaluating the adversarial robustness of adaptive test-time defenses. arXiv preprint arXiv:2202.13711, 2022.
>
> [5] Christian Szegedy, Wojciech Zaremba, Ilya Sutskever, Joan Bruna, Dumitru Erhan, Ian Goodfellow, and Rob Fergus. Intriguing properties of neural networks. arXiv preprint arXiv:1312.6199, 2013.
>
> [6] Aleksander Madry, Aleksandar Makelov, Ludwig Schmidt, Dimitris Tsipras, and Adrian Vladu. Towards deep learning models resistant to adversarial attacks. In International Conference on Learning Representations, 2018.
>
> [7] Aditi Raghunathan, Jacob Steinhardt, and Percy Liang. Certified defenses against adversarial examples. arXiv preprint arXiv:1801.09344, 2018.
>
> [8] Hongyang Zhang, Yaodong Yu, Jiantao Jiao, Eric Xing, Laurent El Ghaoui, and Michael Jordan. Theoretically principled trade-off between robustness and accuracy. In International Conference on Machine Learning, pp. 7472–7482. PMLR, 2019
>
> [9] Pranjal Awasthi, Abhratanu Dutta, and Aravindan Vijayaraghavan. On robustness to adversarial examples and polynomial optimization. Advances in Neural Information Processing Systems, 32, 2019
>
> [10] Sanjam Garg, Somesh Jha, Saeed Mahloujifar, and Mahmoody Mohammad. Adversarially robust learning could leverage computational hardness. In Algorithmic Learning Theory, pp. 364–385. PMLR, 2020.
>
> [11] Nicholas Carlini and David Wagner. Adversarial examples are not easily detected: Bypassing ten detection methods. In Proceedings of the 10th ACM Workshop on Artificial Intelligence and Security, pp. 3–14, 2017a.

---

> > ### Comment · Reviewer_2UCv · 2022-11-27
> > **Confirming the update**
> >
> > dear author(s)
> >
> > thanks for your response and updating the draft.
> >
> > - it is certainly a step forward to clarify the limitations of your results. due to the updates, I can increase my score by one.
> >
> > - however, pointing out the limitations is not (in my opinion) enough to pass the bar, as I think: the complexity and/or NP completeness of attack (or defense) are fundamentally average case tasks. In particular, the learner's job is to output a model based on *sampled* datasets in such a way that it does well *on average* on a fresh point. So, the main result of the paper about the complexity does not really say whether robust learning is NP hard or not.
> >
> > - the heuristic defense of running many attacks and abstaining has limitations: (1) we need to add any new developed attack into the bag of these attacks to be sure. (2) even after that, it is merely a heuristic argument and does not come with any real guarantee on real large inputs/models.
> >
> > I thank you again for your detailed response.

---

> > > ### Author Response · Authors · 2022-11-29
> > > **Re: Confirming the update**
> > >
> > > Dear reviewer 2UCv,
> > >
> > > thank you for the response and the score update.
> > >
> > > A minor clarification: while a cautious defender might want to add every new attack to the bag (as you correctly point out), in practice a well-chosen bag of adversarial attacks can provide robustness that is virtually equivalent to the one provided by a bigger bag (see Appendix J). As new attacks are developed, it is possible that we will see a trend of diminishing returns, to the point where adding new attacks to the bag will provide marginal improvements. Such a phenomenon might represent a valid topic for future work (e.g. by using UG100 to estimate how well new attacks approximate the optimal distances).
> > >
> > > Overall, while we might disagree on some topics (e.g. the role of worst-case complexity in the study of adversarial attacks), we are thankful for this discussion and we believe that your feedback helped us to make significant improvements to our work.
> > >
> > > Authors3153

---

### Author Response · Authors · 2022-11-06
**Thank you for all your reviews of our initial submission**

First of all, we would like to thank all our reviewers for their feedback thus far.

We are aware that our work (as any scientific endeavor) has limitations: indeed one of the goals of the peer review process is to identify them so that progress can be made.

That said, we believe our paper brings solid and important contributions: 1) by means of complexity bounds, it highlights a structural problem in adversarial attacks and defenses; 2) it suggests ways to design defenses that are not affected by the computational asymmetry.

Our paper does not bring any closure to the problem of adversarial robustness (far from it). However, we think it can significantly help researchers understand the limits of what can be achieved with “conventional” adversarial defenses and to better direct their research efforts.

For this reason, we are convinced that not including the work in the program would ultimately result in a loss for the community.

We hope that you, through this discussion, will help us to wring out the remaining issues with our paper so that its core message can be delivered effectively.

In the next few days, we will provide our responses to each of your concerns raised in this first set of reviews, along with paper revisions.

We thank you in advance for all upcoming feedback and suggestions and look forward to a fruitful discussion period.

---

### Author Response · Authors · 2022-11-12
**General Comments for All Reviewers**

Dear reviewers,

we thank you all again for your feedback. We have prepared a detailed response to each of your comments, but since a few points of concern appeared relevant for many of you, we thought to discuss those in a general comment (i.e. this one).

**Core message of the paper**: The main intention of our paper is not to introduce a new defense technique. Rather we want to 1) highlight that there are theoretical reasons why conventional defenses are structurally challenging; 2) show that the issue can be circumvented by taking an alternative, rigorous, perspective on defenses (exemplified by CA); 3) discuss a possible heuristic version of CA that achieves reduced computation time at the cost of approximate robustness.

We acknowledge that, when writing the paper, we may have focused too much on technical aspects, making it difficult to see the big picture. However, this can be fixed without any major restructuring of the content.

**Worst case complexity proofs**: Reduction to problems having a known class is a tried and proved strategy for identifying the worst-case complexity of a new problem.

This approach has repeatedly provided dividends in fields such as numerical optimization, combinatorial optimization, hardware verification, scheduling and logistics. This type of proof highlights key sources of complexity (e.g. in the case of our work, the ability to model disjunctions is strongly connected to NP-hardness, while enumeration over disjunctions is linked to $\Sigma_2^p$ completeness). In the mentioned research fields, worst case complexity analyses have motivated observed empirical results, fostered research in the identification of tractable classes, and inspired effective heuristics. In fact, one of the main motivations for our work was observing how often (and quickly) defense approaches for NNs have been fooled: this is the same pattern that was noticed in the other research areas we mentioned.

Our hope is that our contribution can provide, for the field of adversarial attacks, the same benefits that similar proofs brought to other fields.

**Limits of the experimentation**: Since our main goal is pointing out a potential structural issue and a change of perspective, it made sense to focus our experimentation on evaluating the gap between theoretical properties and what can be achieved by using heuristic attacks. This required us to compute exact decision boundary distances to be used as a reference.

That said, the problem of performing NN verification is both very important and very hard. Many of the more scalable approaches only provide bounds, which are not enough for our evaluation: access to bounds rather than exact distances would prevent us from drawing any firm conclusion.

A few more scalable approaches that can provide exact distances exist, but they are very recent (e.g. [1]), and still not enough to target large networks. Our experimentation took 8 months with the computational resources available to us, which were not negligible (~24 HPC jobs). Further scaling up computational resources would have been unsustainable from both a financial and environmental point of view, and we would not have been able to tackle significantly larger models.

As NN verification techniques improve, it might be possible to verify empirically whether heuristic attacks remain high-quality approximators for the true decision boundary distance.


**Conclusions**: For these reasons, we believe our paper provides a significant contribution with the potential to open new research directions. We are at work on a revised version of the paper that we will post on OpenReview in the next few days, where we will 1) clarify the core message of the paper; 2) clarify both the limits and the strengths of our worst-case complexity proof; 3) explain both the limitations and the usefulness of our experimentation.

We invite you to express any remaining doubts you might have about the paper and we hope that you will choose to support our work.


[1] Calvin Tsay, Jan Kronqvist, Alexander Thebelt, Ruth Misener: Partition-Based Formulations for Mixed-Integer Optimization of Trained ReLU Neural Networks. NeurIPS 2021: 3068-3080

---

### Author Response · Authors · 2022-11-14
**Summary of Changes in the Revised Paper (1/3)**

Dear reviewers,

we have posted an updated version of our paper, modified according to your feedback.
As expected, the technical content is essentially untouched, but we made many smaller changes to how the results are presented, commented, and linked together.

As a result, we think the work is now more approachable and easier to read; the key contributions are better highlighted, together with their strengths, limitations, and their potential impact for future research. In synthesis, the same content is now presented under a new ( and more accurate) light.

This would not have been possible without your contribution: your comments forced us to think hard about what made (in our eyes) our work valuable, its limitations, and the value it can bring to the research community. In our first draft, we took many points for granted and we sometimes focused on specific technical results at the expense of conveying the “big picture”.

At least on our side, this exercise was a great example of how the peer review process, often viewed simply as a gating mechanism, can instead be a tool to advance research quality.

We thank you again and we hope you can share our sentiment in the final version of your reviews.

We list the changes we made to the paper in the reply to this post.

Authors3153

---

> ### Author Response · Authors · 2022-11-14
> **Summary of Changes in the Revised Paper (2/3)**
>
> Abstract:
> * Clarified that the proofs hold in the worst case;
> * Clarified that CA is an example of a different take on adversarial defenses;
> * Clarified that the heuristic version of CA is one of many potential approximations;
> * Clarified that our main goal is to help researcher better direct their efforts.
>
> Section 1 (Introduction):
> * Clarified that the proofs holds in the worst case;
> * Clarified that the complexity classes are distinct unless the Polynomial Hierarchy collapses;
> * Clarified that the computational asymmetry can be sidestepped by adopting an alternative take on adversarial defenses and that CA is an example of this;
> * Clarified that the heuristic version of CA is an example of a trade-off between run-time and guarantees;
> * Clarified that the main outcome of our experimentation is finding how closely and consistently heuristic attack approximate the exact decision boundary distances;
> * Clarified the main intention of the work (support future research by out a potential issue and a promising direction).
>
> Section 2 (Related Work):
> * Added a reference to Appendix M, which contains a comparative discussion of several defense approaches.
>
> Section 3 (Background and Formalization):
> * Recalled that there exists extensive literature suggesting that building robust classifiers is comparatively harder than attacking them;
> * Explained that our definitions capture the shared traits of most literature in the field;
> * Clarified that our definition of adversarial attack is a simplification, but a reasonable (and widely adopted) one.
>
> Section 4 (An Asymmetrical Setting):
> * Clarified that our theoretical results focus on the worst case;
> * Stated more explicitly the link between our results and the Polynomial Hierarchy collapse conjecture;
> * Clarified that Theorems 1 and 2 represent the theoretical groundwork for our main result, i.e. Theorem 3;
> * Provided an intuitive argument for Theorems 1, 2, and 3;
> * Clarified that robustness may be computationally viable in specific sub-cases (and raised interest in their identification);
> * Remarked that, in the general case, our result stands and this is backed by empirical evidence.
>
> Section 4.1 (Heuristic Relaxations):
> * Renamed to “Additional Sources of Asymmetry”;
> * Removed discussion redundant with the previous section;
> * Remarked that the other sources of asymmetry are complementary.
>
> Section 5 (Counter-Attack):
> * Renamed to “Side-Stepping the Computational Asymmetry”;
> * Clarified that CA is just an example of how a change in perspective can sidestep the limitations imposed by Theorem 3.
>
> Section 5.1 (Formal Properties):
> * Clarified that the formal properties hold when CA is used with exact attacks;
> * Simplified the final consideration in the “attacking with higher radius” paragraph.
>
> Section 5.2 (Relaxation):
> * Renamed to “Using Heuristic Attacks with CA”;
> * Clarified that using heuristic attacks is just one example of how to improve computational scalability at the cost of reduced guarantees;
> * Clarified that using heuristic attacks still leads to partial guarantees namely false positives are impossible, while false negatives can happen.
>
> Section 6 (Experimental Evaluation):
> * Renamed to “Empirical Investigation of Heuristic Attacks”;
> * Changed the first paragraph to highlight how investigating the accuracy of heuristic attacks is interesting per-se;
> * Stressed that using verification approaches that provide non-tight bound is not an option;
> * Emphasized the correlation analysis as independent from its use to calibrate a buffer model;
> * In Figure 2, distinguished the Balanced and Strong plots using line style as well as color (to improve accessibility for colorblind people).
>
> Appendices:
> * Replaced the color palette used in the graphs with the IBM Design colorblind-friendly palette;
> * Added Appendix M (which contains a comparative discussion of several defense approaches).
>
> Note that some parts of the main text were summarized or moved to the appendices to fit the 9 page limit.

---

> ### Author Response · Authors · 2022-11-14
> **Summary of Changes in the Revised Paper (3/3)**
>
> Additionally, since OpenReview does not allow the authors to edit the abstract, we report the updated abstract here:
>
> > We consider the decision version of defending and attacking Machine Learning classifiers. We provide a rationale for known difficulties in building robust models by proving that, under broad assumptions, attacking a polynomial-time classifier is $NP$-complete in the worst case; conversely, training a polynomial-time model that is robust on even a single input is $\Sigma_2^P$-complete, barring collapse of the Polynomial Hierarchy. We also provide more general bounds for non-polynomial classifiers.
> We point out an alternative take on adversarial defenses that can sidestep such a complexity gap, by introducing Counter-Attack (CA), a system that computes on-the-fly robustness certificates for a given input up to an arbitrary distance bound $\varepsilon$.
> Finally, we empirically investigate how heuristic attacks can approximate the true decision boundary distance, which has implications for a heuristic version of CA.
> As part of our work, we introduce UG100, a dataset obtained by applying both heuristic and provably optimal attacks to limited-scale networks for MNIST and for CIFAR10.
> We hope our contributions can provide guidance for future research.

---

### Decision · Program_Chairs · 2023-01-20

**Decision:**

Reject

**Justification For Why Not Higher Score:**

N/A

**Justification For Why Not Lower Score:**

N/A

**Metareview: Summary, Strengths And Weaknesses:**

This paper proves that attacking a polynomial-time classifier is NP-complete, and training a polynomial-time model that is robust on a single input is Σ_2^P-complete. The theory seems solid. But the reviewers find that the theory can not provide the insight on the running time in practice. The related work also shows the similar hardness results for ReLU networks. The experiments are also not extensive.